# Humans can use positive and negative spectrotemporal correlations to detect rising and falling pitch

**Parisa A. Vaziri[1], Samuel D. McDougle** [2,3,8] ✉ **& Damon A. Clark** [3,4,5,6,7,8] ✉

To discern speech or appreciate music, the human auditory system detects how pitch changes over time (pitch motion). Here, using psychophysics, computational modelling, functional neuroimaging and analysis of recorded speech, we ask whether humans can detect pitch motion using computations analogous to those used by the visual system. We adapted stimuli from studies of vision to create novel auditory correlated noise stimuli that elicited robust pitch motion percepts. In psychophysical experiments, we discovered that humans can judge pitch direction from spectrotemporal intensity correlations. Robust sensitivity to negative spectrotemporal correlations is a direct analogue of illusory 'reverse-phi' motion in vision, constituting a new auditory illusion. Functional MRI measurements in auditory cortex supported the hypothesis that human auditory processing may employ pitch direction opponency. Linking lab findings to real-world perception, we analysed recordings of English and Mandarin speech and found that pitch direction was signalled by both positive and negative spectrotemporal correlations, suggesting that sensitivity to both types confers ecological benefits. This work reveals how motion detection algorithms sensitive to local correlations are deployed by the central nervous system across disparate modalities (vision and audition) and dimensions (space and frequency).

From discriminating phonemes to being moved by Bach's *Partitas*, detecting relative changes in pitch over time is fundamental to human audition, allowing us to perceptually characterize sounds proceeding from low frequency to high frequency and vice versa (Fig. 1a). Indeed, in everyday speech we use both intonation and lexical tones—including complex rising and falling pitches—to signify meaning[1–3]. In English, for instance, rising pitch at the end of a sentence signifies a question. In Mandarin Chinese, changes of pitch within words convey differences in meaning. But how does the human auditory system detect changes in pitch?

Changes in relative pitch can, in principle, be detected in at least two broad ways. The most well-known computation centres on detecting a fundamental frequency (F0), which refers to the lowest frequency in the ladder of harmonics produced by most vibrating objects, including vocal cords. Once F0 is estimated (which is itself not a trivial task[4]), it may be tracked over time to detect changes in relative pitch[5]. However, relative pitch can also be perceived without detecting a fundamental frequency—psychophysical tests have demonstrated that changes in pitch can be perceived by detecting shifts in multiple constituent

[1]Yale College, Yale University, New Haven, CT, USA. [2]Department of Psychology, Yale University, New Haven, CT, USA. [3]Wu Tsai Institute, Yale University, New Haven, CT, USA. [4]Department of Molecular Cellular and Developmental Biology, Yale University, New Haven, CT, USA. [5]Department of Physics, Yale University, New Haven, CT, USA. [6]Department of Neuroscience, Yale University, New Haven, CT, USA. [7]Quantitative Biology Institute, Yale University, New Haven, CT, USA. [8]These authors jointly supervised this work: Samuel D. McDougle, Damon A. Clark. ✉e-mail: samuel.mcdougle@yale.edu; damon.clark@yale.edu

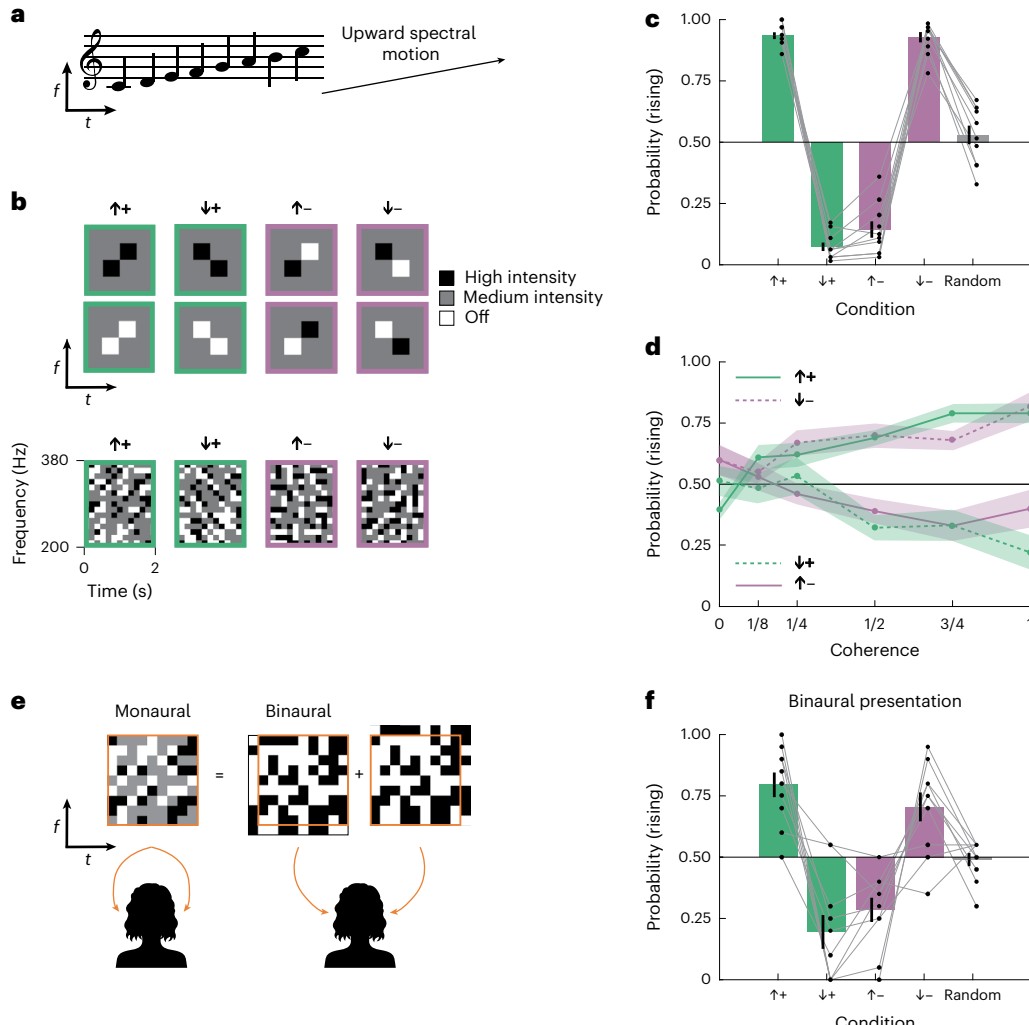

**Fig. 1 | Humans detect auditory motion in pairwise frequency–time correlations. a**, A rising sound written on a music staff and in frequency–time. **b**, Diagrams showing sample (top) and actual (bottom) stimuli. Frequency–time correlations can be directed either upward or downward and be either positively or negatively correlated. Stimulus intensities were updated every 1/6 s. **c**, Perceived direction of stimuli with varying direction and correlation. The bars and error bars show means ± s.e.m. over $N = 10$ participants. Two-sided, one-sample $t$-tests revealed significant deviations from chance (0.50) in pitch direction judgements in all four stimulus conditions (all $P < 0.001$). Pitch direction judgements in the random stimulus condition were not significantly different from chance ($P = 0.454$). **d**, Perceived direction of stimuli with varying degrees of correlation (coherence) in the stimulus. The upward-directed positive and downward-directed negative curves were not significantly different ($P = 0.164$ according to a two-way, repeated-measures ANOVA); similarly, the downward-directed positive and upward-directed negative curves were also not significantly different ($P = 0.500$, same test). Both ANOVAs revealed significant main effects of coherence on pitch direction judgements (all $P < 0.001$). The error shading represents ±s.e.m. ($N = 10$ participants). **e**, How binaural stimuli were presented to each ear. **f**, Perceived direction of stimuli with varying directions and correlations using binaural presentation. Two-sided, one-sample $t$-tests revealed significant deviations from chance (0.50) in pitch direction judgements in all four stimulus conditions (all $P < 0.001$). Pitch direction judgements in the random stimulus condition were not significantly different from chance ($P = 0.716$). The bars and error bars represent means ± s.e.m. ($N = 10$ participants).

frequencies of a sound without needing to compute F0 (ref. 6). These results indicate that a variety of cues in a sound, not just F0, can support the detection of pitch changes. Importantly, work on such spectral pattern cues typically employs sounds in which single tones or groups of tones shift in frequency[7–12]. These sorts of spectral intensity patterns could in principle be identified and tracked by auditory systems, akin to the detection of unique sound sources in noise (for example, the cocktail party effect[13,14]), or, more loosely, akin to the detection and tracking of objects in vision. Here, inspired by work in the visual system, we focus on novel auditory stimuli that possess no F0 information but contain local positive and negative spectrotemporal correlations in intensity.

Why examine spectrotemporal correlations? In vision, sensing local spatiotemporal correlations is the basis of canonical models for spatial motion detection[15,16]. These models generate sensitivity to

pairwise intensity correlations over time and space through a process of linear filtering and nonlinear interactions. Visual sensitivity to intensity correlations can be dramatically revealed by visual illusions, specifically those involving negative spatiotemporal correlations, as exemplified by 'reverse-phi' phenomena[15–17]. Thus, at least in vision, local intensity correlations serve as important cues for detecting motion in the environment. These cues feed into local motion detectors that complement parallel systems that detect motion by tracking visual objects over time[18–20]. Crucially, the sensitivity to negative correlations in reverse-phi phenomena is fundamentally inconsistent with purely object- or pattern-tracking-based models of motion detection.

Some existing evidence supports the possibility that local spectrotemporal correlations could aid in perceiving directional pitch changes without the need for computing F0 or tracking global spectral

patterns. For example, studies of cortical auditory neurons have characterized spectrotemporal receptive fields with shapes that can be oriented over frequency and time[21–23], which create signals that can in principle be used to reconstruct sounds, including changes in frequency[24]. Such receptive fields could support auditory feature tracking but could also sense spectrotemporal correlations[16]. Spectral correlations have been proposed to underlie pitch perception[25], while the joint spectrotemporal spectra of sounds support the perception of timbre, speech and music[26,27]. Moreover, a prior study presented a spectral intensity pattern that was repeatedly inverted and frequency-shifted in time, leading to perceptual reversals of pitch direction[28]. However, no previous study has isolated negative spectrotemporal correlations, which allow for definitive evaluations of the role of correlations in detecting pitch direction. Here, to evaluate whether and how spectrotemporal correlations contribute to pitch motion detection, we developed novel auditory stimuli that lack both long-range spectral features and a common F0 but contain specified local positive and—critically—negative spectrotemporal correlations.

## Results

### Spectral motion without features

We set out to test whether humans can detect auditory motion on the basis of local positive or negative spectrotemporal correlations. To do this, we adapted a stimulus used to study visual motion detection[29,30] to develop new correlated noise auditory stimuli that use increments and decrements in intensity to generate local correlations in intensity at specific offsets in frequency and time (Fig. 1b). The stimuli are generated by creating a random envelope that scales carrier frequencies over time and then adding the envelope back to itself after shifting it by an offset in frequency and time (Methods). This procedure generates correlations that are local because they are limited to a single offset in frequency and time. At that offset, the correlations in intensity can be set to be positive or negative, but outside of that offset all intensity correlations in the stimulus are zero (Extended Data Fig. 1a). We designed four stimuli with positive or negative correlations in intensity at a temporal offset of 1/6 s, with the frequency offset directed either upward or downward by 1/15 octave (Fig. 1b and Extended Data Fig. 1). These sounds were inharmonic, so that fundamental frequencies could not be used to judge pitch changes[6,31]. We presented these stimuli to participants for 2 s and asked them to report whether they perceived the sound as having a rising or falling pitch profile over time.

Participants reported that upward-directed positive correlations (↑+) rose in pitch over time and downward-directed positive correlations (↓+) fell in pitch over time (Fig. 1c and Supplementary Video 1). Being inharmonic, the stimulus did not carry F0 cues that could be leveraged to detect relative pitch. In these stimuli, the positive correlation over time reflected spectral features that partially persist for up to 1/3 s (Methods). Thus, while this result suggests that humans can use spectrotemporal correlations to judge rising versus falling pitch, it is also conceivable that humans could track the transient spectral patterns in this stimulus over these short timescales.

Remarkably, however, when we presented stimuli with negative correlations in frequency and time, participants reported opposing percepts (Fig. 1c and Supplementary Videos 1 and 2). That is, the upward-directed negative correlations (↑−) sounded like they were falling in pitch, while the downward-directed negative correlations (↓−) sounded like they were rising in pitch. Participants who consistently perceived rising or falling pitch in the stimuli with positive correlations also consistently perceived rising or falling pitch in the stimuli with negative correlations (Extended Data Fig. 1b). This striking illusion suggests that human audition is sensitive to negative spectrotemporal correlations. The inverted responses to negative spectrotemporal correlations are a direct analogue to illusory reverse-phi visual motion percepts, which have been reported across many species and phyla[15,17,32–34].

How does the strength of these spectrotemporal correlations relate to perception? To answer this question, we varied the coherence of the stimulus and again asked participants to judge whether tones were rising or falling in pitch (Fig. 1d). We titrated the coherence of the stimuli from 1 down to 0 by randomly replacing correlated frequency–time elements with random ones, such that the coherence represented the fraction of original correlations remaining (Methods). With high coherence, participants perceived rising and falling pitches in a pattern similar to the first experiment (Fig. 1c). As coherence decreased, however, the probability of judging a sound as rising tended towards chance (0.5). There were no significant differences between the psychophysical curves for (↑+) versus (↓−) or (↓+) versus (↑−) ($P = 0.164$ and 0.500, respectively, as measured by a two-way, repeated-measures analysis of variance (ANOVA)). This indicates that inverting the stimulus correlation and direction led to indistinguishable percepts. These results also reveal a clear monotonic relationship between the strength of spectrotemporal correlations and the strength of pitch change percepts, for both positive and negative correlations.

In vision, object tracking can integrate information between the two eyes, while local-correlation-based algorithms rely on correlations within each eye[20]. We next asked whether spectrotemporal correlations for pitch motion detection are computed monaurally or binaurally. The structure of our correlated noise stimulus is created by summing a random binary envelope with itself at a frequency–time offset (Fig. 1e and Methods). This allowed us to play one binary envelope to the left ear and an offset one to the right ear, such that neither ear alone was presented with any correlations. In this context, detecting spectrotemporal correlations can proceed only by integrating information across the two ears. We played all four types of binaural correlations to participants and asked them to judge whether they heard rising or falling tones. They reported the same pattern of percepts as in the monaural stimuli, though with average reported directions somewhat closer to chance. This demonstrates that the perception of rising or falling pitch uses information from both ears to integrate intensity information to compute spectrotemporal correlations. This is consistent with data showing that many cortical auditory neurons integrate signals from both ears[35] and represents a divergence from correlation detection in the human visual system, which is predominantly monocular[19].

### Tuning of human spectrotemporal correlation detectors

Our next step was to characterize the spectral and temporal tuning of the correlation sensitivity we had observed. To do this, we designed a different kind of stimulus, one inspired by random dot kinematograms in visual neuroscience[36]. In these stimuli, a medium-intensity sound that played at all frequencies was interrupted by brief pips at different frequencies, 50 ms in duration[37]. These pips either increased the intensity of a specific frequency or decreased it to zero (Fig. 2a and Methods). After an initial set of pips were placed randomly in frequency and time, we added a second set of pips with a specific delay in time and change in frequency, yielding correlated 'pip pairs'. These pairs had positive correlations when both pips were high intensity or both were silent, and negative correlations when one was high intensity and one was silent. This allowed us to create auditory stimuli with upward- and downward-directed pairs of pips with positive or negative correlations (Fig. 2b). Like the stimuli used in Fig. 1, these stimuli had no auditory objects that persisted for more than two pips in time or frequency, but, crucially, they allowed us to continuously vary the delay between correlated pips. These stimuli helped us better isolate perceptual sensitivity to delays between tones and frequency displacements.

We first used these stimuli to map out the sensitivity to different delays between individuated tones. We kept the frequency change at 1/15 octave and swept values of the delay between correlated pips while asking participants to judge whether the pitch was rising or falling over time (Fig. 2c). For both negative and positive correlations and upward- and downward-directed displacements, we found that peak directional

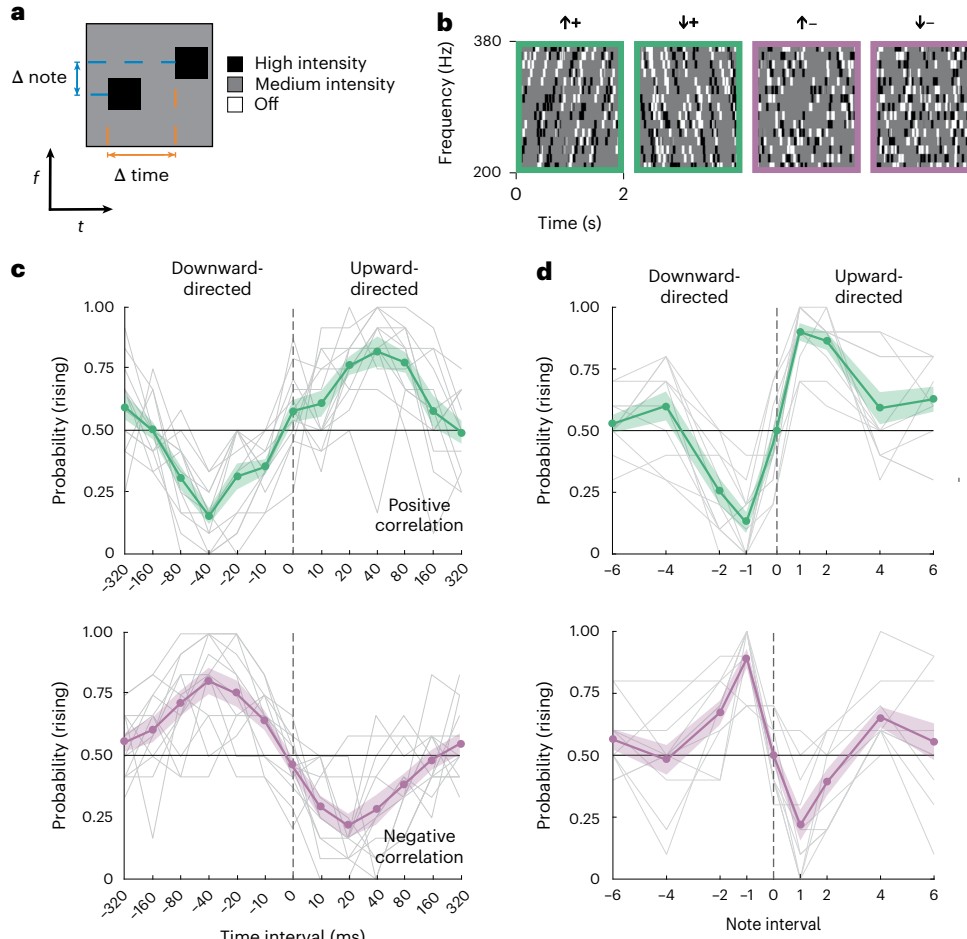

**Fig. 2 | Correlation detection is tuned to small frequency changes and short delays in time. a**, A correlated pip pair with a frequency displacement (note) and a delay between pips. **b**, Spectrotemporal diagrams of four different correlated pip stimuli directed upward and downward with positive and negative correlations. Pip duration in these experiments was 50 ms. Stimulus envelopes had a resolution of 1 ms, and pip pairs were not synchronized across frequencies. **c**, Perceived direction of stimuli with note = +1 and varying pip delays; results are shown for positive pip correlations (top) and negative pip correlations (bottom). One-way, repeated-measures ANOVAs for the positive and negative correlation curves revealed significantly different responses across pip delays (all $P < 0.001$). The grey lines are individual participant curves. The error shading represents ±s.e.m. ($N = 13$ participants). **d**, Perceived direction of stimuli using varying note intervals and 40-ms pip delays; results are shown for positive pip correlations (top) and negative pip correlations (bottom). One-way, repeated-measures ANOVAs for the positive and negative correlation curves revealed significantly different responses across note intervals (all $P < 0.001$). The grey lines are individual participant curves. The error shading represents ±s.e.m. ($N = 11$ participants).

sensitivity occurred at a delay of around 40 ms. This peak was measured for pips with a duration of 50 ms but did not change appreciably when the pip duration was shortened to 20 ms (Extended Data Fig. 2a,b). According to models for visual motion estimation, this peak sensitivity value can reflect the characteristic delay timescale in the circuits detecting local motion signals[29]. The peak delay seen here is slightly longer than peak delays measured by similar experiments in human and fly visual systems[29,38]. The timescale of sensitivity is typical of timescale differences over which asynchrony of tone presentation can be detected[39] but also over which stream tracking can occur[40]. Importantly, the timescales are similar for both positive and negative correlations, while it is difficult to reason how a high–low- or low–high-intensity pair of pips could be tracked as an auditory object.

We then measured sensitivity to the magnitude of displacements in frequency space. Using a similar method, we set the delay to 40 ms and varied the frequency displacement within each pip pair (Fig. 2d). We found that peak sensitivity occurred for tone displacements of 1/15 octave, though there was still significant direction selectivity at 2/15-octave displacements ($P = 0.001$ for both positive and negative correlations according to a paired $t$-test). This result shows that motion

detectors in the human auditory system are most sensitive to small shifts in frequency in the vicinity of 1/15 of an octave (4.7% changes in frequency) or less, for both positive and negative correlation stimuli. This result is consistent with peak sensitivity for changes in complex sounds[41] and with the smaller values of frequency discrimination thresholds in humans[7].

These results build on our previous findings and are inconsistent with alternative explanations, such as motion detection tracking heuristics that, for instance, rely only on comparing the net high–high-intensity pairs of tones[28]. Such short-timescale tracking heuristics could conceivably explain the results in Fig. 1, where the presence of negative correlations with one displacement implies the absence of positive correlations with the same displacement, so that monitoring high–high-intensity pip pairs could explain responses to negative correlation stimuli (Extended Data Fig. 2c). However, the sparse correlations in correlated pip stimuli mean that detecting negative correlations using tracking heuristics of high–high-intensity pip pairs is substantially more difficult than just detecting the negative correlation patterns directly (Extended Data Fig. 2d). The inverted and proportional percepts in negative correlation patterns are thus

parsimoniously explained by the use of local spectrotemporal correlations to detect rising and falling pitch.

## Sensitivity to spectrotemporal intensity patterns

Our positive and negative correlation stimuli each consist of multiple patterns in intensity over frequency and time. Upward-directed positive correlation (↑+) stimuli consist of both high–high and low–low intensity combinations, whereas the negative versions (↑−) consist of both high–low and low–high intensity combinations. Prior work using long-lasting spectrotemporal correlations in auditory stimuli has posited that humans are selectively sensitive to the high–high combinations[28]. Are humans sensitive to all four pairwise combinations or to just a subset of them? To address this question, we generated new correlated pip auditory stimuli (Fig. 3a) where each stimulus possessed correlated paired pips of only one of the four types: high–high, low–low, high–low or low–high intensity (Methods). We asked participants to judge whether these different stimuli were rising or falling and recorded their responses (Fig. 3b). Participants were sensitive to all four different pairings, with both upward and downward displacements generating significant direction-selective responses to each pairing (Fig. 3c), and with the familiar inversion of the percept when the pip correlation was reversed. Like the other sparse correlated pip stimuli, these percepts are more plausibly accounted for with spectrotemporal correlation detection than with a tracking heuristic (Extended Data Fig. 3a).

In visual motion detection, one generalization beyond pairwise correlations involves so-called triplet correlations[42,43]. In vision, triplet correlation stimuli are patterns that contain spatiotemporal correlations in intensity between three points in space and time but no pairwise correlations, and can elicit visual motion percepts in humans[42,44], flies[44,45] and fish[46]. Those results show that visual motion detection algorithms are sensitive to this higher-order correlative structure; but is the same true in audition? To answer this question, we presented participants with auditory analogues of visual triplet correlation stimuli (Extended Data Fig. 3a and Methods). They did perceive auditory motion in these stimuli (Extended Data Fig. 3c,d), and the perceived direction reversed when the correlation reversed, similar to the inversion in the percepts of the pairwise correlations (Figs. 1–3). This result demonstrates that human perception of pitch direction exploits third-order correlations (or possibly even higher-order ones), as well as the second-order correlations that make up the bulk of this study. Interestingly, humans perceived motion in higher-order auditory correlations in a pattern very similar to the pattern found in fly and fish visual perception (Extended Data Fig. 3e). This correspondence across both species and modalities points to notable similarities in the neural algorithms used by animals in processing auditory (spectral) and visual (spatial) motion.

## Psychophysical and cortical signatures of opponent subtraction of spectral motion signals

When we presented positively and negatively correlated stimuli, we discovered a striking symmetry: the tuning of percepts of negative correlation stimuli matched the tuning of percepts of positive correlation stimuli that were displaced in the opposite direction (Fig. 2c,d). This symmetry is suggestive of an opponent architecture. To investigate this, we first built a toy model of a motion energy model unit to describe a hypothetical directionally tuned auditory unit (Fig. 4a). The model unit temporally filtered and summed sound intensity at two adjacent frequencies in a pattern that enhanced upward-directed spectral motion, similar to prior suggestions[47], before sending the filtered signal through a quadratic nonlinearity[16]. (We note that this model is not intended to be a realistic model of neural processing, but rather a tractable simplification of spectrotemporal correlation detection algorithms[16].)

When we presented this model with correlated pip stimuli (Fig. 2), it responded at an elevated baseline level, with deviations

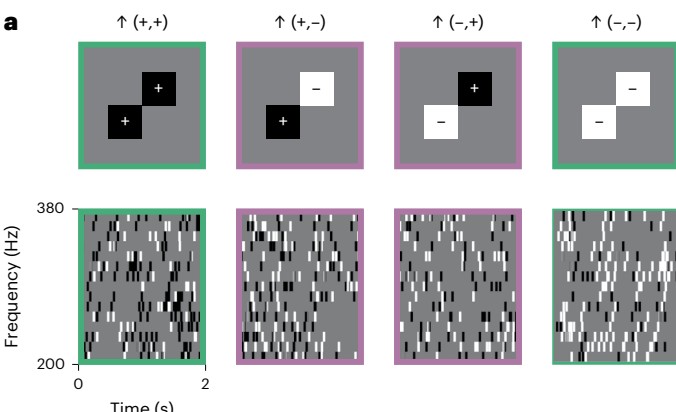

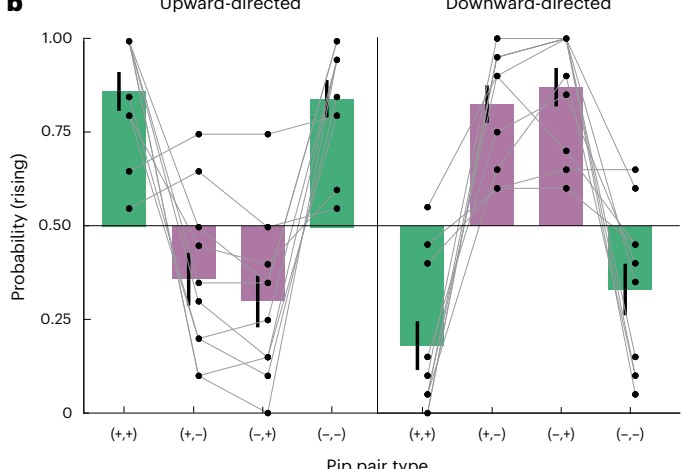

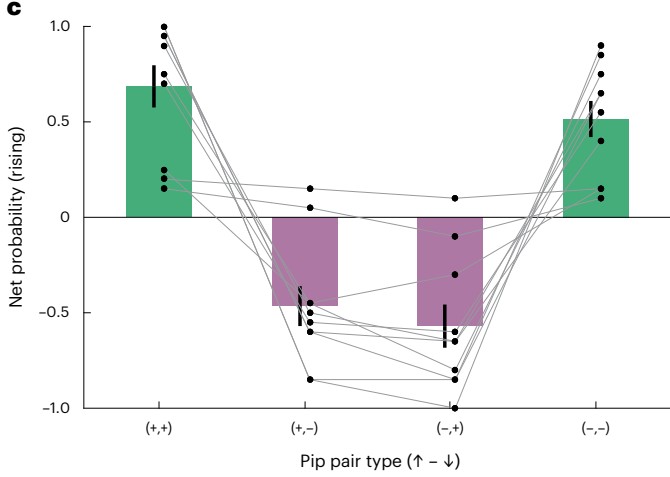

**Fig. 3 | Sensitivity to all four pairwise intensity combinations contributes to rising and falling pitch perception. a,** Frequency–time diagrams of four different 50-ms-long pip combinations, presented with 40-ms delays. Stimulus envelopes had a resolution of 1 ms, and pip pairs were not synchronized across frequencies. **b,** Probability of perceiving rising pitch for each of the four intensity combinations directed upward (left) and downward (right). Two-sided, one-sample $t$-tests comparing upward- versus downward-directed stimuli for each matched pair revealed significant direction selectivity across all pitch direction judgements (all $P < 0.001$). The bars and error bars represent means ± s.e.m. ($N = 10$ participants). **c,** Net perceived direction of pip pair stimuli. The net probability rising is computed using $P$(rising|upward-directed) − $P$(rising|downward-directed) in **b** to obtain a net probability distinguishing the upward- and downward-directed pip patterns. The bars and error bars represent means ± s.e.m. ($N = 10$ participants).

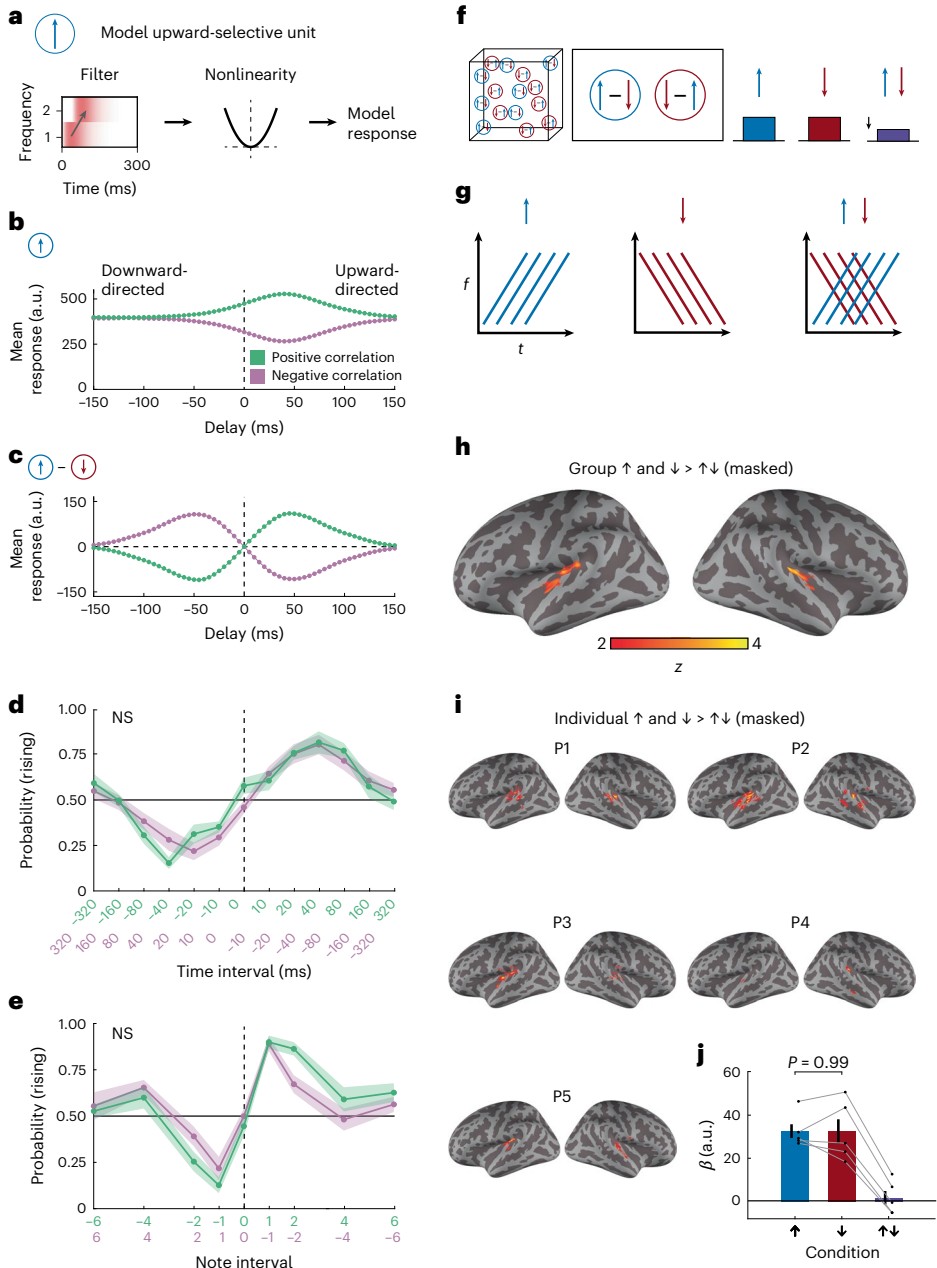

**Fig. 4 | Bilateral regions of human auditory cortex show signatures of opponency. a**, A simple model auditory unit that responds more to upward-directed spectral motion than to downward-directed spectral motion. The stimulus spectrogram is convolved with an upward-oriented spectrotemporal filter before the result is squared, as in a motion energy model[16]. **b**, Mean response of the unit to correlated pip stimuli with different delays and correlation signs, corresponding to upward- and downward-directed positive and negative correlations. **c**, As in **b**, but for an opponent signal, consisting of an upwardly tuned unit response minus an identical unit tuned to downward motion. **d**, Comparison of P(rising) for positive and negative correlation stimuli sweeping the time interval, aligning upward-directed positive correlation stimuli with downward-directed negative correlation stimuli. The data are replotted from Fig. 2. The curves were not significantly different ($P = 0.655$ according to a two-way, repeated-measures ANOVA). The error shading represents ±s.e.m. ($N = 13$ participants). **e**, As in **d** but for sweeping the tone difference. The curves were not significantly different ($P = 0.965$ according to a two-way, repeated-measures ANOVA). The error shading represents ±s.e.m. ($N = 11$ participants). **f**, Conceptual schematic of opponency in brain regions. An opponent voxel/region would respond strongly to rising and falling tones but be suppressed by the sum of the two stimuli. **g**, Stimulus design. Stimuli were rising, falling, or

summed rising and falling. **h**, Group-level analysis. Surface maps were masked to include only auditory regions (Methods and Extended Data Fig. 4). A weighted linear average comparison looked for voxels where the mean activation to rising stimuli and to falling stimuli (weighted equally) exceeded the activation to the simultaneously presented rising and falling stimulus. A bilateral region within auditory cortex responded less to summed stimuli than to non-summed stimuli. Thresholded at $P < 0.05$ (two-sided) with a cluster-forming constraint of 20 voxels. See Extended Data Fig. 4 for a cluster-corrected group map using non-parametric permutation-based inference. **i**, Individual-level analysis. Surface maps were masked to include only auditory regions (Methods and Extended Data Fig. 4). Regions in auditory cortex across participants (P1–P5) responded less to summed stimuli than to non-summed stimuli, using the same analysis as in **h**. Thresholded at $P < 0.05$ (two-sided) with a cluster-forming constraint of 20 voxels. See Extended Data Fig. 4 for individual cluster-corrected maps using non-parametric permutation-based inference. **j**, Control analysis showing symmetric $\beta$ values in response to rising and falling stimuli in individually defined opponent regions of interest (ROIs) ($P = 0.986$ via a two-sided, one-sample $t$-test). (Note that all $\beta$ values are relative to an implicit baseline that includes responses to ambient scanner noise.) The bars and error bars represent means ± s.e.m. ($N = 5$ participants). a.u., arbitrary units.

that depended on the direction and sign of the stimulus correlation (Fig. 4b). As designed, it responded more to upward-directed positive correlations than to downward-directed ones. Since this model relies solely on pairwise correlations, it was also expected that negative correlation stimuli elicited equal and opposite deviations from baseline to positive correlation stimuli (Fig. 4b). These opposite deviations result in the symmetry of the positive and negative correlation curves about a response of ~400 arbitrary units (Fig. 4b). Crucially, however, in this model, negatively correlated stimuli exhibit a different tuning from oppositely directed positive stimuli; inverting the correlation is not equivalent to inverting the correlated pip direction (that is, the temporal delay in the pips). In other words, at a given delay, inverting the correlation results in a different response versus simply inverting the delay while retaining the correlation. This model thus does not predict the symmetry we observed in the psychophysical experiments, where negatively correlated stimuli were perceived similarly to positively correlated stimuli but with the opposite direction (Fig. 2c,d).

We next created an opponent signal by subtracting signals from two model units with opposite directional tuning (Fig. 4c). This opponent signal responded to positively correlated stimuli with positive and negative values when they were directed upward and downward (Fig. 4c, green). Critically, this opponent signal possesses the key symmetry: responses to negatively correlated stimuli have identical tuning to positively correlated stimuli in the opposite direction. Upward-directed negative correlation stimuli thus yield the same responses as downward-directed positive correlation stimuli. We also derived this result analytically: when motion energy signals are opponently subtracted, negative correlation stimuli elicit mean responses that match oppositely directed positive correlation stimuli (see Methods for derivation).

To more directly test whether our data contained this symmetry, we compared percepts of negative correlation stimuli to percepts of positive correlation stimuli in the opposite direction, for both frequency change and delay time tuning (Fig. 4d,e, replotting data from Fig. 2c,d). The curves appeared to fully superimpose, and ANOVA tests confirmed that there was no measurable difference between the positive correlation curves and the flipped negative correlation curves (see figure legends for statistics). This robust symmetry between positive and negative correlation stimuli has also been found in visual motion detection in fruit flies[29] and in humans[38].

In primate vision, opponent subtraction occurs in visual area V5, also called MT[48,49], which is causally involved in visual motion perception[50]. Similarly, fly visual systems subtract motion signals with opposing preferred directions[51]. Motivated by our psychophysical results, by analogies with vision, by proposals for opponent subtraction in spectral direction[8] and by spectral direction opponent auditory cells found in bats[52], we reasoned that human auditory cortex might possess signatures of opponent processing. Indeed, human auditory cortex shows signatures of opponent coding in another domain of auditory processing, azimuthal location coding[53].

We followed the logic of previous functional magnetic resonance imaging (fMRI) studies that identified opponent signals in human cortical area MT by using visual stimuli that summed motion in opposite directions[54]. To start, we assumed that cortical voxels involved in detecting spectral motion contain units that respond preferentially to rising tones and units that respond preferentially to falling tones, but not units that respond to both (Fig. 4f)[55]. Such a voxel should thus respond reliably to stimuli containing either rising or falling tones. The key distinction here between systems with and without opponency lies in their responses to a summed stimulus that contains superimposed rising and falling tones: if units are opponent, then the summed stimulus should cause a decrease in voxel activity due to a net suppression of signals in units with opponent responses[54]. Related prior work has examined overlapping rising and falling tones as forms of acoustic textures[56] but to our knowledge has not directly examined this form

of opponency. We therefore designed simple stimuli consisting of rising tones, falling tones or their sum (Fig. 4g, Extended Data Fig. 4d and Supplementary Video 3) and presented them to participants while measuring blood-oxygen-level-dependent (BOLD) signals via fMRI. (We note that we did not use the correlated noise stimuli from the earlier experiments because pilot studies found that pitch motion in those stimuli was difficult to discern when competing with scanner noise.)

We searched within a broad a priori anatomical auditory cortex mask (Extended Data Fig. 4g) for voxels that responded more to the non-summed (rising or falling) stimuli than to the summed (opponent) stimulus (Methods). At both the group and individual levels, a bilateral region within superior temporal cortex was significantly more activated by the non-summed stimuli than by the summed stimulus (Fig. 4h,i), consistent with opponency (see Extended Data Fig. 4 for additional statistical analyses). The group map extended over several bilateral functional subregions of the primary and non-primary auditory cortex[57], including core regions A1 and RI, Area 52, and some lateral and medial belt regions (Extended Data Fig. 4f), though we note that given the modest sample we cannot make strong anatomical inferences on the basis of these results. According to the opponency hypothesis, activity in opponent voxels should be similar in magnitude for rising and falling stimuli and suppressed for the summed stimulus. We thus wanted to ensure that our results followed this symmetry and were not biased by either the rising or falling stimulus alone (Methods). Activity in putative opponent regions was indeed comparable for rising and falling tones (Fig. 4j). Overall, these suggestive fMRI findings (though in a small sample of participants) echo a key result from our behavioural studies—symmetry between positive and negative correlation percepts. To our knowledge, these regions of human auditory cortex have not previously been identified as potential loci for opponent spectral motion signals, though they are broadly consistent with regions of auditory cortex sensitive to spectral processing[58-61].

## Positive and negative correlation spectrotemporal cues signal tone modulation in speech

Is there an ecological advantage in detecting both positive and negative spectrotemporal correlations in intensity? To address this question, we asked how spectrotemporal correlations could help predict rising and falling frequencies in naturalistic sounds. This approach follows literature in visual motion detection, in which different spatiotemporal cues have been analysed to understand how they can be used to infer motion in scenes[43,62-64]. To examine rising and falling pitch in naturalistic sound, we chose to examine human speech, where tone modulation contains critical semantic information in both tonal and non-tonal languages[1-3]. Since humans are sensitive to both positive and negative pairwise correlations in frequency and time, we hypothesized that these correlations could convey information about the direction and speed of tone modulation in human speech, in addition to the F0 cues also associated with human speech[65]. Following in the tradition of relating auditory processing to natural sounds[66-69], we analysed corpora of spoken English and Mandarin and examined how rising and falling tone modulation is related to underlying positive and negative pairwise spectrotemporal correlations in intensity (Fig. 5 and Methods).

Our analysis took several steps, which were intended simply to analyse the structure of spectrotemporal correlations in speech, not reproduce any processing in the ear or downstream auditory system. First, we computed spectrograms for each of the speech recordings (Fig. 5a, top). We then used an optical flow algorithm to estimate how the tones in the sound changed at each point in time, a quantity we termed the tone change. The tone change represents the degree to which the sound was rising or falling in frequency at each time point (Fig. 5a, bottom, and Methods). Next, we binarized the spectrogram and looked for specific patterns of intensity in frequency and time, examining all four patterns of high and low intensity combinations: high–high, low–low, high–low and low–high intensity (Fig. 5b), which

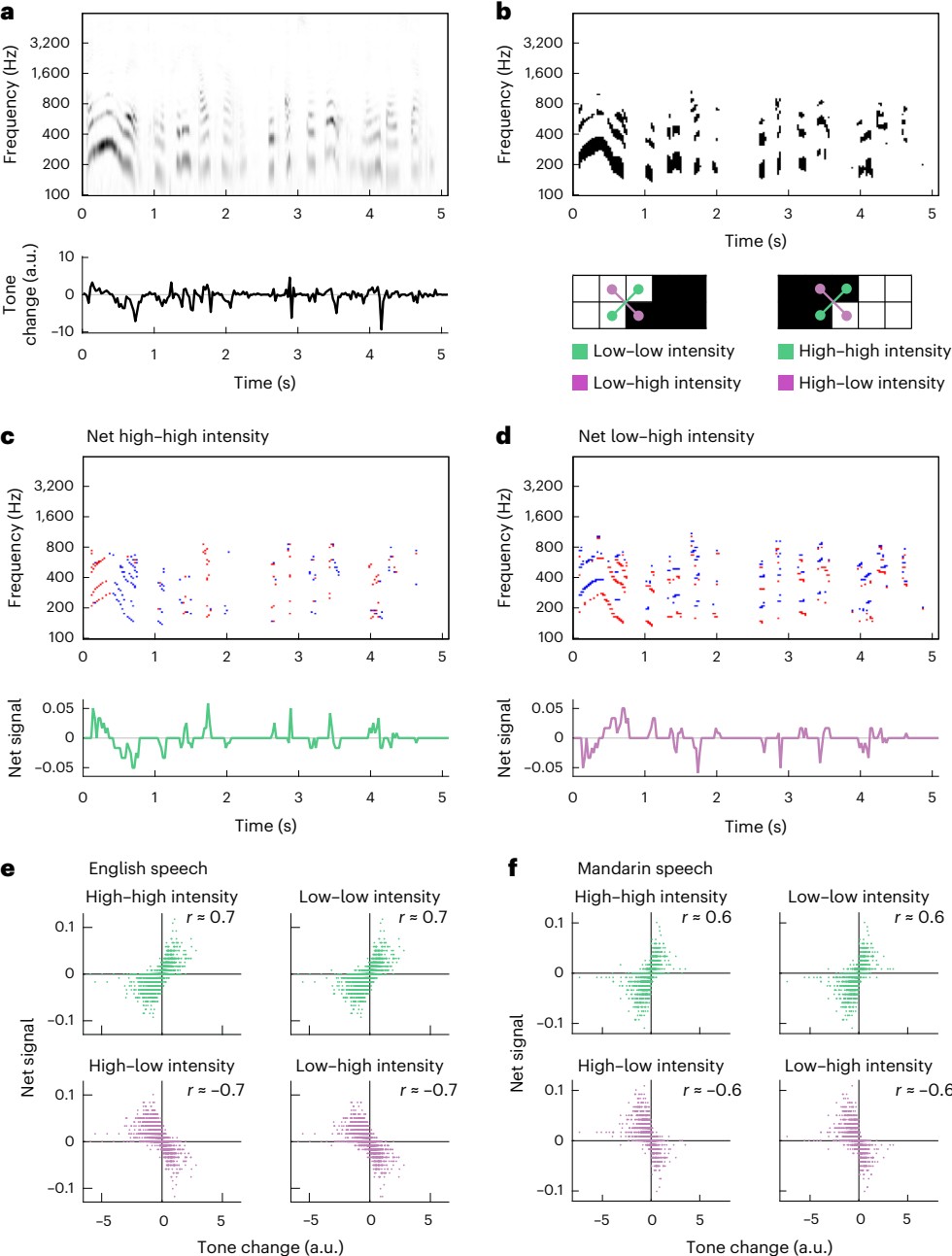

**Fig. 5 | Rising and falling tone in spoken language can be detected through both positive and negative pairwise correlations. a**, Spectrogram of a voice saying, "Anyone lived in a pretty how town (with up so falling many bells down)" (top), and intonation velocity estimate from the spectrogram (bottom; Methods). Positive tone changes correspond to rising frequencies in the sound. **b**, Binarized spectrogram from **a** (top) and four distinct high- and low-intensity frequency–time combinations in the binarized spectrogram (bottom). **c**, Net high-high intensity instances at each frequency and time in the binarized spectrogram in **b** (top; red indicates +1, blue indicates −1 and white indicates 0)

and frequency-averaged net high–high-intensity signal (bottom). **d**, Net low–high intensity patterns at each frequency and time in the binarized spectrogram in **b** (top; red indicates +1, blue indicates −1 and white indicates 0) and frequency-averaged net low–high intensity signal (bottom). **e**, Correlations between the tone change estimate at each time and the frequency-averaged net signals for high–high, low–low, high–low and low–high intensity patterns. The data are from the English speech corpus[102]. **f**, As in **e** but for the Mandarin speech corpus[103]. a.u., arbitrary units.

constitute spectrotemporal patterns that contain positive correlations (high–high and low–low intensity) and negative correlations (high–low and low–high intensity). We next computed the local net signal for each pattern at each frequency and time by subtracting the downward-directed patterns from the upward-directed ones for each of the four patterns (Fig. 5c,d). Finally, we averaged these local net signals over all frequencies to obtain a net pattern signal at each time point (Fig. 5c,d). Computing the net pattern signals is an operation

consistent with the opponency we observed psychophysically and in fMRI (Fig. 4). For the high–high intensity patterns, there was a positive correlation between the time trace of the net pattern signal and the tone change. For the high–low intensity patterns, the correlation was negative. This correspondence suggests that both positive and negative spectrotemporal correlations contain information about tone changes that could be useful to listeners in detecting rising and falling tones in speech.

To see whether this result generalized, we analysed hundreds of speech snippets that totalled over 90 min in English and 40 min in Mandarin Chinese (Fig. 5e,f and Methods). In English, the tone changes should be dominated by intonation, while in Mandarin Chinese, the tone changes should reflect both intonational and within-syllable changes in tone[1–3]. We reproduced the analysis of the different intensity patterns and then correlated the net signal for each pattern with the computed change in tone. In both English and Mandarin Chinese, the two positive correlation patterns (high–high and low–low intensity) produced a strong positive correlation ($r > 0.5$) with the tone change, whereas the two negative correlation patterns (high–low and low–high intensity) produced a strong negative correlation ($r < -0.5$) (Fig. 5e,f). These results show that all four patterns could be useful in estimating tone changes in speech. The negative stimulus correlation produced an anti-correlation with tone changes, which explains why they elicit percepts in the opposite direction: upward-directed negative spectrotemporal correlations indicate downward-directed tone changes. We obtained similar results when we processed the speech recordings with continuous rather than digital operations to obtain positive and negative spectrotemporal correlations (Methods and Extended Data Fig. 5). It is not surprising that correlation-based algorithms robustly detect pitch direction, since in vision these algorithms work with a variety of stimuli[16,63,70] and it is clear how correlations arise at object edges (Fig. 5b). Still, the contributions of negative correlations in naturalistic stimuli have remained underappreciated. This analysis provides an ecological explanation of the observed inverted percepts to negative auditory correlations: negative spectrotemporal correlations provide useful information for distinguishing between sounds with rising versus falling frequencies. The reversed perceptual direction for the negative correlation stimuli matches the relationship of negative correlations with rising and falling frequencies in sounds.

## Discussion

In the studies reported here, we have demonstrated that humans are sensitive to positive and negative spectrotemporal correlations in intensity over frequency and time as they discern whether a sound is rising or falling in pitch (Figs. 1–3). These correlations can be sensed over durations of less than 100 ms (Fig. 2). The perception of negative spectrotemporal correlations in sparse stimuli (Figs. 2 and 3) argues against explanations based on spectral pattern tracking. The perception of negative spectrotemporal intensity correlations also mirrors a powerful visual phenomenon, the reverse-phi illusion, in a different modality (audition) and over a different dimension of motion (frequency). Inspired by our behavioural results showing symmetry between inverting correlation and inverting direction, we hypothesized that the human auditory system might implement opponent subtraction, echoing a similar operation in visual motion detection. Using fMRI, we found suggestive evidence that regions within human auditory cortex show signatures consistent with opponency (Fig. 4). Finally, we demonstrated that both positive and negative spectrotemporal correlations can act as reliable cues to assess tone changes in speech (Fig. 5).

The stimuli we developed here (Figs. 1–3) in some ways resemble Shepard tones[71], which were designed to sound like they are unceasingly rising or falling. However, Shepard tones consist of periodic auditory features that persist over frequency and time (similar to the stimuli in Fig. 4f). The rising or falling of a Shepard tone could thus be assessed by simply tracking auditory features over time. The auditory stimuli we developed and investigated here, however, have no such persistent features—a rising or falling percept must instead depend on detecting local positive and negative pairwise spectrotemporal correlations within the stimulus. Thus, the strong percepts of rising and falling tones, which depended on the sign of the correlation, reflect an authentic auditory illusion in which there is no true rising or falling tone but only the imposition of specific local spectrotemporal correlations in intensity.

Sensitivity to spectrotemporal correlations in judging pitch direction could act as a mechanism for frequency shift detectors inferred from previous psychophysical studies[8,11]. The sensitivity to correlations at different frequency offsets measured here (Fig. 2d) shares tuning with the previously inferred shift detectors[41], and arrays of local correlation detectors would also naturally solve the problem of binding of subsequent tones used in frequency shift measurements[11]. The present experiments differ from prior ones, however, by emphasizing correlation sensitivity, especially to negative correlations. However, if frequency shift detectors employ sensitivity to pairwise intensity correlations, that sensitivity would need to act in coordination with other algorithms for judging changes in relative pitch. In particular, changes in frequency can be judged over gaps of up to a second[41], much longer than the correlation sensitivity we measured (Fig. 2c); this points to a different system for such judgements. Judgements about relative pitch tend to be made using fundamental frequencies in harmonic sounds, but also via the tracking of broader spectral patterns[6,31,72]. These examples suggest that auditory spectral motion processing is similar to visual motion processing in that there are a variety of probably distinct algorithms at play: changes can be detected by both local correlational algorithms and slower, longer-range object-tracking algorithms[19].

Our psychophysical findings inspired us to test the idea that opponent computations may be performed during spectrotemporal processing. We observed activity in regions in both primary and non-primary auditory cortex across Heschl's gyrus and somewhat less so in the superior temporal gyrus (STG) that may reflect opponent computations to resolve net pitch direction (Fig. 4h,i). The regions of primary and non-primary auditory cortex that we suggest are candidates for showing opponent-related activity were broadly consistent with areas seen in fMRI literature on spectral and pitch processing (Extended Data Fig. 4)[60,61,73]. We speculate that this result reflects the intriguing possibility that direction opponency is a basic feature of primary or non-primary auditory cortical regions that process relative spectral information over time—this possibility should be tested further in larger samples and with a wider variety of stimuli. How might our neural results relate to the neural underpinnings of speech perception? Human auditory cortex displays regional specialization, with areas that selectively encode different aspects of speech, primarily in the STG[74–77]. Our results are broadly consistent with other findings showing that regions within the STG encode variability in speaker intonation and lexical tone[77,78]; however, we may have observed lower STG involvement in our task because of the pure tone stimuli we used. Overall, our fMRI results suggest that opponency may be a signature of pitch direction processing in circuits involved in simple pitch computations and perhaps in more complex perceptual tasks such as speech processing. Future work can further test this hypothesis, perhaps using invasive methods that more directly characterize neural activity[78].

The sensitivity to spectrotemporal correlations in sound intensity can be accommodated by existing measurements and by established theories of auditory processing. Canonical algorithmic models for motion detection are explicitly sensitive to negative correlations[15,16], and more neurophysiologically inspired models for motion detection are similarly sensitive to both positive and negative correlations[79,80]. At the single-neuron level, units in rodent[81], bat[82] and primate[37] auditory cortex display spectrotemporally oriented receptive fields, which should confer sensitivity to pitch motion direction[24] and to both positive and negative spectrotemporal correlations (Fig. 4 and Extended Data Fig. 4)[16]. Our results suggest that neurons with this type of sensitivity—an oriented spectrotemporal receptive field followed by a nonlinearity—could underlie spectrotemporal correlation detection in humans. Such concise models of direction selectivity could be aided in their tuning to specific correlational features in complex sounds by peripheral nonlinearities found in the auditory system[63,83]. Meanwhile, our psychophysical and fMRI results also suggest that units in multiple regions across auditory cortex might exhibit directional

opponency, a property observed in bat auditory neurons[52]. Indeed, such opponent codes are also observed in auditory spatial localization in mammals[53,84–87]. Pitch direction opponency could arise in primary motion detectors, depending on their linear and nonlinear processing motifs[88], or could result from subtracting opposing cortical or subcortical motion signals[89,90]. Importantly, while these proposed mechanisms can account for the measurements in this research, our work here proposes that it is the tuning of these physiological properties to correlational structures in sound that is critical to detecting pitch direction[16,43].

There are well-established similarities in the processing of visual motion between invertebrates and vertebrates[91–93], phyla that diverged hundreds of millions of years ago. Our study shows that local correlational algorithms for motion detection also span modalities, since human audition and vision appear to employ similar computational motifs. Audition thus joins olfaction[94] as a non-visual sense where pairwise, local correlations can generate directional motion percepts. A critical aspect of our results is that sensitivity to pairwise stimulus correlations also implies sensitivity to negative correlations. This sensitivity to negative correlations is due in part to the mathematics of computing correlations (Methods)[16,43,95], providing a conceptual framework for understanding the neural detection of motion that spans modality and species.

Lastly, negative correlations sensed in audition probably act as useful cues to infer real-world changes in the frequency domain (Fig. 5), just as they may aid in visual motion detection[63,96]. Auditory processing appears to be tuned to the statistics of natural sounds[66–69], so direction selectivity in the auditory system could also be tuned to specific correlational structures in natural sounds, including higher-order structures (Extended Data Fig. 3). Thus, the illusory pitch motion percepts to negative spectrotemporal intensity correlations described here are not just an interesting laboratory epiphenomenon. Rather, they reflect neural sensitivity to the statistics of the auditory world, with direct implications for everyday speech and music perception.

## Methods

### Psychophysical measurements

All participants ($N = 33$; 12 female; mean age, 23.3 years; range, 18 to 32 years) provided informed, written consent in accordance with procedures approved by the Yale University Institutional Review Board, and participated for credit or for US$20 per hour. To measure human psychophysical curves (Figs. 1–3), we recruited participants with self-reported normal hearing from within the university population. The participants were seated in a quiet room, wearing headphones (Model DT 770 PRO, Beyerdynamic) to listen to various sound stimuli and make perceptual judgements. The sounds were created in MATLAB v.2021b and presented using Psychtoolbox v.3.0.18 (refs. 97–99) on a MacBook Pro, using its native soundcard and the command PsychPortAudio. The participants adjusted the intensity to a comfortable level. Each sound was played for 2 s, after which the participants were cued to judge, to the best of their ability, whether it sounded like a rising or falling tone. To ensure they understood the task, before the experiment began, we had the participants listen to the sounds and told them to judge whether overall it was rising or falling, but we did not provide feedback about their answers. The following is the approximate script used by the experimenter at the beginning of each experiment: "In this experiment, your task is to decide whether a tone is rising or falling in pitch over time. Press the 'up' arrow if you think that the tone is rising and press the 'down' arrow if you think that the tone is falling. Please do not react too quickly or too slowly. You should wait for the tone to finish before you make your decision, but you should not take more than a couple of seconds after the conclusion of the tone to make your decision. You will receive a warning on the screen if your reaction times are too fast or slow. I will now play a couple of example tones, so that you can practice deciding whether the tone is rising or falling. Please

adjust the headphone volume, so that it is at a comfortable level, and make sure that you can hear the tones clearly. After we listen to a few example tones, let me know when you feel ready to begin the task."

Participants usually completed two experiments lasting approximately 15 min each. Three participants were excluded: one who reported being tone-deaf and unable to discern any rising or falling pitch in stimuli and two who reported they had not followed the instructions to report the perceived pitch direction. The data were analysed using custom code written in MATLAB. The code to produce the sounds, all anonymized data, and the code used to analyse the data and produce Figs. 1–3 are all publicly available at https://github.com/ClarkLabCode/humanAuditoryCorrelations and https://doi.org/10.5061/dryad.hmgqnk9w8.

### Creating correlated sounds

We created complex sounds containing multiple frequencies, following the design of visual stimuli that have been informative in that field. To do this, we created a comb of constant carrier frequencies, with frequencies ranging over six octaves from 200 Hz to 6,400 Hz, with 15 frequencies per octave, equally spaced in log-frequency. The sampling frequency was chosen to be 20 kHz for all experiments. Each carrier frequency waveform was then multiplied by a slower, time-varying envelope, before the frequencies were summed to make the overall waveform for that sound. Mathematically, the sound waveform, $w(t)$, looks like:

$$w(t) = \sum_{i=1}^{N} \theta_i m_i(t) \sin(2\pi f_i t)$$

where $f_i$ is the indexed carrier frequency, $t$ is sampled at 20 kHz and the value $\theta_i$ was chosen to roughly equalize the perceptual salience of the different frequencies, using the ISO standard 226 at 60 dB. (We note that in various tests in the lab, this perceptual salience scaling was not critical for the percepts we measured; since we included it in the initial experiments, we included it for all stimuli in this study.) It remains to compute the suite of $m_i(t)$ envelope functions to create each sound. The envelope functions were computed as outlined below. All envelope functions were computed to have non-negative binary or ternary values and were filtered with a 25-ms low-pass filter in the ternary stimuli (Fig. 1) and a 0.5-ms low-pass filter in the pip stimuli (Figs. 2 and 3) to eliminate sharp transitions.

**Ternary pairwise correlations.** To create sounds with only local, pairwise correlations between specific frequency and time offsets (Fig. 1b–d), we followed a protocol used in prior visual experiments[29,30,100]. On the basis of informal experiments attempting to optimize our own percepts, we discretized frequencies into 15 notes per octave and time into 1/6-s frames. This change in frequency is similar to the most salient change in frequency in a prior study[41]. We then created an initial binary mask in this coarse-time representation, $B_{i,j}$, where $i$ indexed the frequency and $j$ the time step in 1/6-s intervals. In each trial, each element of $B$ was chosen from a Bernoulli distribution with probability 0.5 and then centred to have values of ±1/2 instead of 0 and 1. A ternary mask, $M$, was created by the following formula:

$$M_{i,j} = B_{i,j} + QB_{i+d,j+1}$$

The mask is thus the binary matrix added back to itself with a displacement in frequency of $d = \pm 1$ for upward and downward-directed correlations. The mask is ternary, with values of 0, 1 and −1. The correlation parity is chosen by $Q = \pm 1$, so that the offset matrices are added to create positive correlations and subtracted to create negative correlations. The discrete autocorrelation function of this mask $M$ is equal to:

$$C_{m,n} = \frac{1}{2}\delta_{m,0}\delta_{n,0} + \frac{1}{4}Q\left(\delta_{m,d}\delta_{n,1} + \delta_{m,-d}\delta_{n,-1}\right)$$

where the $\delta_{i,j}$ terms are Kronicker delta functions (Extended Data Fig. 1). Importantly, the elements in the mask are not deterministically the same or different at the spectrotemporal offset of the correlated displacement, so that spectral patterns vary substantially at each temporal update of the stimulus.

A continuous time expression for the autocorrelation function is available in a prior work describing similar stimuli in vision[30]. This stimulus construction is reminiscent of iterated rippled noise (IRN) used in prior psychophysical experiments[101], but it is different in three important respects: the correlations here are generated in the envelope of the noise, not the waveform itself (as in IRN); the temporal offsets are larger than in IRN; and the offsets in time are coupled with offsets in frequency to generate spectrotemporal correlations, while IRN generates correlations in time.

The coarse-time matrix $M$ was recentred to have values of 0, 0.5 and 1, then up-sampled to the sampling frequency $F_s$ to create $m_i(t)$ at each frequency, acting as envelopes on the amplitude of the carrier waveform at each frequency. These envelopes were filtered with a 25-ms low-pass filter to eliminate sharp transitions.

To create the stimuli with varying coherence, we replaced a fraction of mask elements with random ternary stimuli, drawn from the values (0, 0.5, 1) with probabilities (0.25, 0.5, 0.25). The fraction replaced was equal to $(1 - C)$, where $C$ is the coherence value.

**Binaural pairwise correlations.** To play sounds such that correlations existed only by integrating across the ears (Fig. 1d,e), we simply played $B_{i,j}$ in one ear and $QB_{i+d,j+1}$ in the other ear for the correlations as described above to create the ternary pairwise correlations. To play these binary masks, we created two masks $M_{i,j} = B_{i,j}$ and $M_{i,j} = QB_{i\pm d,j+1}$ to play to the two ears. The matrices were recentred to have values of 0 and 1 and then up-sampled to the sampling frequency. The masks were filtered with a 0.5-ms low-pass filter to eliminate sharp transitions.

**Correlated pips with time and frequency offsets.** We wanted to measure perceptual tuning to different time and frequency scales of correlation (Fig. 2). When we sped up the ternary pairwise correlation stimulus, so that notes lasted for less than 50 ms, we were unable to discern any rising or falling pitches. We therefore devised the correlated pip stimulus, an alternative stimulus with sparse but well-defined spectrotemporal correlations, in which we could smoothly vary the correlation delay and still perceive rising or falling pitch in the sounds.

To create the correlated pip stimulus, we discretized frequency space into 15 tones per octave. We first initialized our masks $m_i(t)$ to be 0 for all times, sampled at the sampling frequency $F_s$. We then placed initial delta-function pips in a Poisson distribution across all frequencies and times in our sound, at a rate of four pips per frequency per second. Positive and negative pips were equally probable, represented by mask values of ±1. We then created a second set of pips offset by the selected change in frequency and delay time, according to the two different correlation types. After imposing the correlations, the overall pip rate became eight pips per frequency per second. We then convolved this event trace with a boxcar function with the length of the pip duration to create the mask at $F_s$. Pips had a duration of 40 ms in Fig. 2 and 20 ms in Extended Data Fig. 2. Last, the masks were linearly transformed to be between 0 and 1 and filtered with a 0.5-ms low-pass filter to eliminate sharp transitions. The high-intensity values corresponded to values of 1 in the mask, the low intensity to values of 0 and the background to values of 0.5.

**Correlations between high- and low-intensity pips.** These stimuli (Fig. 3) were generated similarly to the correlated pips stimulus above. However, only two thirds of all pips were in correlated pairs of high–high, low–low, high–low or low–high. In the case of the high–high correlated pips, the remaining third of pips consisted of randomly placed low-intensity pips. In the case of low–low correlated pips, the remaining third of pips consisted of randomly placed high-intensity pips. And in the cases of low–high and high–low, the remaining third were equally distributed between low- and high-intensity pips. The four types thus had equal numbers of correlated pairs in each stimulus. The overall rate of pips for all stimuli was six pips per frequency per second.

**Triplet correlations.** We made triplet correlation binary masks, discretized in frequency and time, following prior procedures (Extended Data Fig. 3)[42,44]. The frequency was discretized into 15 tones per octave, and time was discretized into 1/6-s frames. The frequencies began at 200 Hz and ranged over five octaves. The masks $m_i(t)$ were linearly transformed to have values of 0 and 1 and were filtered in time with a 0.5-ms low-pass filter to eliminate sharp transitions.

**Rising, falling and opponent tones.** To create the rising, falling and opponent tones used in our fMRI experiment (Fig. 4), we used frequencies discretized into 1/16-octave steps and time discretized into 1/6-s steps. Ascending tones were created from a binary mask equal to an ascending line of time–frequency elements in this discretized space (Extended Data Fig. 4), and descending tones consisted of a descending line of time–frequency elements. The summed ascending plus descending was the sum of the two masks. All masks were filtered in time with a 0.5-ms low-pass filter to eliminate sharp transitions. We switched to 16 steps per octave for this experiment so that the ascending and descending stimuli never played the same frequency simultaneously, making the addition of the stimuli more straightforward.

Code to generate the sounds used in these experiments is available at https://github.com/ClarkLabCode/humanAuditoryCorrelations.

## Tracking heuristic analysis

An alternative explanation to sensitivity to intensity correlations over frequency and time is a tracking heuristic, in which the auditory system for instance tracks high-intensity frequencies over time[28]. Most basically, this would for instance detect patterns of high–high intensity at neighbouring frequencies and subsequent times. To analyse how such a heuristic might analyse the different stimuli in our study, we computed the envelopes for the ternary correlated noise stimulus in Fig. 1 and the correlated pip stimuli in Figs. 2 and 3 with 1-ms resolution. For each time step in the envelopes, we analysed how often there occurred patterns of high–high, low–low, high–low and low–high intensity pairs at neighbouring frequencies with an offset of 40 ms in time. These patterns are represented in Extended Data Figs. 2c,d and 3a by HH, LL, HL and LH, respectively We examined these patterns in the direction of the imposed correlation ('with') and in the opposite direction ('against'). The occurrence was measured as the fraction of the time where each pattern was observed. The net occurrence was measured as the 'with' occurrence fraction minus the 'against' occurrence fraction and is shown in the lower axes in these plots. In the correlated ternary noise stimulus, for instance, the net HH occurrence was positive for the positive stimuli and negative for the negative stimuli, indicating that this tracking heuristic inverts when the correlation sign is reversed. However, in the correlated pip stimuli, the net HH occurrence was much smaller for the negative correlation stimulus than for the positive correlation stimulus, while the negatively correlated patterns (HL and LH) carried strong signals. In this case, the heuristic of tracking HH alone cannot easily explain the reversed percepts, while sensitivity to intensity correlations is a parsimonious explanation for all the psychophysical results. Overall, this feature-counting analysis is similar to the analysis of sounds in Fig. 5.

## Model motion energy unit

We created a toy model motion energy unit by convolving a linear filter with a sound spectrogram and then squaring the result (Fig. 4). This model was simply intended to provide intuition about how spectrotemporal intensity signals could be processed—it is not intended to

mimic the spectral processing of the cochlea and subsequent processing steps. That is:

$$r(t) = ((f_1 * S_1)(t) + (f_2 * S_2)(t))^2$$

The filters were chosen to be:

$$f_1(t) = \frac{t^2}{\tau^3} e^{-t/\tau}$$

$$f_2(t) = f_1(t - T)\,\Theta(t - T)$$

where $f_2$ is just a time-shifted version of $f_1$ with a time shift of $T = 40$ ms. The timescale of the filter, $\tau$, was chosen to be 20 ms so that it reached its peak at 40 ms. The function $\Theta$ is a Heaviside step function. The two filters are applied to adjacent frequencies in the spectrogram, $S_1(t)$ and $S_2(t)$, so that the filter enhances signals directed upward over time.

We computed the mean of $r(t)$ over time to get the mean response for a given stimulus. Stimuli were created to match the correlated pip-style stimuli in Fig. 2. The opponent response was computed as

$$r_{\mathrm{opp}}(t) = ((f_1 * S_1)(t) + (f_2 * S_2)(t))^2 - ((f_2 * S_1)(t) + (f_1 * S_2)(t))^2$$

The second, negative term is the same as the first term but with the filter flipped in frequency space, so that it corresponds to a downward selective unit. This response was likewise averaged over time to produce the plots in Fig. 4.

The MATLAB code to create Fig. 4b,c is available at https://github.com/ClarkLabCode/humanAuditoryCorrelations.

## Speech analysis

Spoken language databases were analysed to ask how spectrotemporal correlations could act as indicators for rising and falling tones in speech. The analysis simply examines the spectrotemporal correlations in the sounds, rather than attempting to mimic any realistic auditory processing. Using MATLAB, we first loaded short snippets of speech from two databases: 438 snippets constituting a total of 91 min of data from Librispeech, a corpus of read English[102]; and 749 snippets constituting a total of 52 min of data from MagicData Mandarin Chinese Read Speech Corpus[103], a corpus of read Mandarin. We computed a spectrogram for each snippet of speech using the MATLAB command `spectrogram`; we extracted the spectral amplitude at a resolution of 40 samples per second with no overlap between samples, at 20 evenly spaced frequencies per octave from 100 Hz to 6,400 Hz (Fig. 5a). We estimated the rising/falling intonation change of the sound at each point using the MATLAB command `opticalFlowHS`, which uses the Horn–Schunck method[104] to estimate directional local flow (typically optic flow) between frames. We averaged the calculated flow over frequencies to compute an estimate of the frequency 'flow' with arbitrary units, which we termed tone change (Fig. 5a). This method does not make strong assumptions about how changes in speech tone or frequency should be computed. It should work to extract tone changes from most complex sounds. We then examined estimators of this tone change as follows:

(1) To compute binary correlations in frequency and time, we first binarized the spectrogram using Otsu's method (MATLAB command `imbinarize`)[105], which maximizes the variance between the binarized time–frequency element amplitudes while minimizing variance within each of the two categories (Fig. 5b). We made eight new binary frequency–time data arrays, containing Boolean values at each point in time and frequency, $V_{t,f,\uparrow,\pm,\pm} := (\{A_{t,f}\,A_{t+1,f+1}\} = \{\pm1, \pm1\})$, and an equivalent one for downward-directed intensity patterns. These matrices are records of the existence of each pattern of sound intensity at each time and frequency. From these, we computed the net signal of each pattern at each frequency by subtracting the downward-directed matrix from the upward-directed one. We last found the mean net signal over all frequencies for each pattern (Fig. 5c,d). We computed the correlation between these mean net signals at each time point with the calculated upward or downward flow velocity (Fig. 5e,f). Note that these net pattern signals sum to 0 over the four different patterns (±,±), so that the four signals are not independent.

(2) To generate non-binarized correlation plots, we first linearly filtered the spectrogram amplitudes, $A_{t,f}$, to take temporal derivatives: $F_{t,f} = A_{t,f} - A_{t-1,f}$. We then used these derivatives, $F_{t,f}$, which have positive and negative values, as inputs to a Hassenstein–Reichardt correlator model[15,63]. We then computed the net (+,+) correlations, for instance, as $N_{t,f,+,+} = [F_{t,f}]_+ [F_{t+1,f+1}]_+ - [F_{t+1,f}]_+ [F_{t,f+1}]_+$, where $[x]_+ = x$ when $x > 0$ and $[x]_+ = 0$ otherwise. We used a similar process to compute the net (−,−), (+,−) and (−,+) correlations. We averaged these signals over frequency to obtain a single indicator of velocity at each point in time. These indicators were then correlated with the estimated tone change of the sound snippet at that point in time (Extended Data Fig. 5).

Code to analyse the spoken language databases and produce the panels in Fig. 5 is available at https://github.com/ClarkLabCode/humanAuditoryCorrelations.

## fMRI recordings and analysis

Whole-brain imaging was performed at the Brain Imaging Center at Yale University, on a Siemens 3T Prisma MRI scanner using a 32-channel head coil. Functional data were acquired with a gradient-echo echoplanar pulse sequence (TR = 0.80 s; TE = 30 ms; flip angle, 52°; voxel size, 2.4 mm × 2.4 mm × 2.4 mm; MB acc. factor, 6). T1-weighted MP-RAGE anatomical images were collected as well (TR = 2.5 s; TE = 2.0 ms; flip angle, 8°; 208 slices; voxel size, 1.0 mm isotropic). Functional imaging in our sample (N = 5; 1 female; mean age, 26.2 years; range, 20 to 36 years; P.A.V. and S.D.M. were participants in the fMRI study) was performed in ~5-min runs, with the total number of functional runs per participant ranging from three to five. The participants provided informed, written consent in accordance with procedures approved by the Yale University Institutional Review Board and participated either for class credit or for US$20 per hour. Fifteen auditory stimuli were presented per run in an event-related design (five each of three stimulus types: rising, falling and summed). (We found that the correlated intensity sounds played in Fig. 1, for instance, were not easily distinguishable by participants in the scanner, so we did not examine neural responses to them in this study.) Each stimulus lasted for 13.33 s, separated by an inter-trial interval of 4 s. The order of the three stimulus types was randomized in each run. The participants passively listened to the tones and were not required to render any responses. MRI-optimized noise-cancelling headphones (Optoacoustics OptoACTIVE III) were used to limit the effects of background scanner noise, and the noise-cancelling software was trained on the EPI sequence sound features before each session using a brief calibration run.

The fMRI-Prep toolbox was used for preprocessing[106]. The anatomical image was corrected for intensity non-uniformity with N4BiasFieldCorrection[107] and used as the T1w-reference. The T1w-reference was then skull-stripped with a Nipype implementation of the antsBrainExtraction.sh workflow in ANTs, and tissue segmentation of cerebrospinal fluid, white matter and grey matter was performed on the brain-extracted T1w using FFAST (FSL v.6.0.5)[108]. Volume-based spatial normalization to standard (MNI) space was performed through nonlinear registration with antsRegistration (ANTs v.2.3.3). For each of the BOLD runs, a reference volume and its skull-stripped version were generated using a custom methodology of fMRIPrep. Head-motion

parameters were estimated using MCFLIRT (FSL v.6.0.5)[109], and BOLD time series were resampled into native space by applying the transforms to correct for head motion. The BOLD reference was co-registered to the anatomical reference using mri_coreg (FreeSurfer) followed by FLIRT. Co-registration was configured with 6 DOF. Several confounding time series were calculated on the basis of the preprocessed BOLD: framewise displacement, DVARS and three region-wise global signals. The BOLD time series were resampled into standard space, and volumetric resamplings were performed using ANTs.

Our main analyses involved constructing general linear models to quantify the effects of the three stimulus types within auditory cortex. General linear model analyses were performed using Nilearn v.0.10.1 (ref. [110]). Confound regressors of no interest (generated using fMRIPrep; see above) were entered into each general linear model. These included six standard motion regressors, the framewise displacement time course, and white matter and global signal time courses. Each stimulus type (rising, falling and summed) was modelled using boxcar regressors over the entire stimulus presentation phase (13.33 s) of the relevant trials and was convolved with the canonical double-gamma haemodynamic response function. The main contrast of interest at the group and individual levels compared BOLD responses to the non-summed directional stimuli (that is, rising and falling) with responses to the summed stimuli (that is, superimposed rising + falling). The contrast was designed to highlight deviations from a null hypothesis of equivalent responses between directional and opponent stimuli. Individual participant runs were combined and then brought to the group level for second-level analyses. In our visualizations for Fig. 4, we controlled the false positive rate (thresholded) at $P < 0.05$ (uncorrected), with a cluster-forming threshold of 20 voxels. Individual-level results for all participants were also analysed and are displayed in Fig. 4, using the same thresholding parameters. All contrasts were performed within an a priori anatomical mask that included both the STG and Heschel's gyrus, created by combining all voxels that crossed the 50% probability threshold for either area within a combined bilateral probabilistic atlas (Harvard–Oxford). The individual and group results were projected onto the standard (MNI) cortical surface (FreeSurfer) for visualization.

In addition to creating these uncorrected maps with only a cluster size constraint to avoid false positives, we performed more conservative non-parametric inference on group and individual brain maps to further control the false discovery rate. At the group level, we performed non-parametric inference (using the non_parametric_inference second-level analysis function in Nilearn) by generating a null distribution of maximum cluster sizes at the group level using permutations and then computing an empirical $P$ value for the true, observed maximum cluster size. We ran this analysis with 10,000 permutations and a cluster-forming $\alpha$ of 0.05. In a more stringent test, we also performed non-parametric inference at the individual level, permuting condition labels for each individual stimulus in each run to generate an empirical null distribution of maximum cluster sizes for each participant[111]. We ran this analysis with 100 permutations per participant and a cluster-forming threshold of $z > 3$. The resulting non-parametric cluster-corrected group and individual maps are shown in Extended Data Fig. 4, and the (one-sided) $P$ value of the true maximum cluster size compared with the null distribution is reported in Extended Data Fig. 4.

A simple control analysis was also performed to ensure that the non-summed > summed results were not driven by a single non-summed stimulus (for example, rising or falling) having a proportionally larger response, but rather by symmetric responses to the rising and falling stimuli. To perform this control analysis, we first extracted individualized ROIs from the non-summed > summed contrast (using the threshold described above) and then extracted average $\beta$ values within that ROI for each stimulus type. We note that while this was of course not an unbiased ROI relative to the hypothesis

that non-summed stimuli would on average show stronger activity than summed, it was unbiased relative to the hypothesis of symmetric responses to rising versus falling tones.

## Opponency implies a symmetry in responses with opposite correlations in opposite directions

The motion energy model uses pairwise correlations to extract motion information from input stimuli and seems to accurately represent important aspects of cellular physiology[16]. In the motion energy model, stimuli over space and time, $S(x,t)$, are convolved with a space-time-oriented linear filter, $H(x,t)$. (In this section, we derive results in space, but a frequency variable $f$ could substitute for $x$, and this approach would apply sound intensity over frequency rather than light intensity over space.) The result of the convolution is squared to obtain a response:

$$r(x,t) = \left( \iint dx' dt' H(x',t') S(x-x', t-t') \right)^2$$

This response is stronger, on average, to stimuli with motion in the preferred direction than in the null direction. The preferred direction corresponds to the orientation of the filter $H$ in space-time, which amplifies signals when the motion direction aligns with the filter orientation. When the response is averaged over time and space, it yields a pleasing form in Fourier space, such that the mean response is the dot product of the stimulus power with a weighting function[16]:

$$\langle r \rangle = \iint dk d\omega |\tilde{H}(k,\omega)|^2 |\tilde{S}(k,\omega)|^2$$

where $\tilde{H}$ and $\tilde{S}$ are the Fourier transforms of $H$ and $S$. Therefore, to understand the responses of this model, it is useful to compute the power spectrum of the stimulus.

For a random dot kinematogram in which the dots are displaced by $\Delta x$ in space and $\Delta t$ in time, the covariance density, $C$, of the stimulus is a function of the offsets in time and space, $x$ and $t$:

$$C(x,t) = \beta \delta(x,t) + \alpha \delta(x - \Delta x, t - \Delta t) + \alpha \delta(x + \Delta x, t + \Delta t)$$

where the first term is the stimulus autocovariance and the remaining two terms correspond to correlations in the stimulus at offsets of $(\Delta x, \Delta t)$ and $(-\Delta x, -\Delta t)$. For random dot kinematograms, $\beta < 1$ and $\alpha$ can take on positive or negative values for positively and negative correlated random dot kinematograms. This derivation is in continuous space, using Dirac delta function correlations; a similar result with discrete time and frequencies was found earlier in the methods for the ternary stimuli. The power spectrum of the stimulus is the Fourier transform of this covariance function:

$$|\tilde{S}(k,\omega)|^2 = \iint dx dt e^{ikx} e^{i\omega t} C(x,t) = \beta + \alpha \cos(\omega \Delta t + k \Delta x)$$

The power is highest/lowest along lines of constant phase in cosine, or when $\omega \Delta t + k \Delta x = n\pi$. When the $\alpha$ is negative, for negative correlation stimuli, this effectively changes the phase of the cosine by 180 degrees. The motion energy model says the mean response to such a stimulus, for a unit with filter $H$, is:

$$\langle r \rangle = \iint dk d\omega |\tilde{H}(k,\omega)|^2 (\beta + \alpha \cos(\omega \Delta t + k \Delta x))$$

This is the type of curve shown in Fig. 4b, in which there is a baseline response determined by $\beta$ and the integral of $|\tilde{H}(k,\omega)|^2$. There is a modulatory term that depends on $\alpha$ and the dot product of $|\tilde{H}(k,\omega)|^2$ with $\cos(\omega \Delta t + k \Delta x)$, which gives the modulation a directional tuning.

This form means that the modulation inverts when the sign of the correlation (sign of $\alpha$) inverts. If there is a peak response to a stimulus with correlation $\alpha$ at a specific $\Delta t$ and $\Delta x$, then the peak will be equal and opposite when $\alpha$ is inverted. Importantly, however, the peak is not the same when the direction of the stimulus is inverted—that is, when $\Delta x \rightarrow -\Delta x$.

However, if we compute an opponent response, in which we subtract the response with one filter orientation from the response with the opposite filter orientation (inverting the $k$ in the Fourier domain), then we find:

$$\langle r_{\mathrm{opp}} \rangle = \iint \mathrm{d}k\mathrm{d}\omega (|\bar{H}(k,\omega)|^2 - |\bar{H}(-k,\omega)|^2)(\beta + \alpha \cos(\omega \Delta t + k \Delta x))$$

$$\langle r_{\mathrm{opp}} \rangle = \alpha \iint \mathrm{d}k\mathrm{d}\omega (|\tilde{H}(k,\omega)|^2 - |\tilde{H}(-k,\omega)|^2)(\cos(\omega \Delta t + k \Delta x))$$

Here we see that the opponent subtraction causes the $\beta$ term to drop out entirely so that the remaining term is just proportional to $\alpha$, the correlation in the stimulus. The mean opponent response can be computed for correlation stimuli with parameters $\alpha$, $\Delta t$ and $\Delta x$: $\langle r_{\mathrm{opp}}(\alpha, \Delta t, \Delta x)\rangle$. Because of the directional opponency, the response inverts when the stimulus is reversed in space:

$$\langle r_{\mathrm{opp}}(\alpha, \Delta t, \Delta x) \rangle = -\langle r_{\mathrm{opp}}(\alpha, \Delta t, -\Delta x) \rangle$$

And because of the proportionality with the correlation, the response inverts when the stimulus correlation is inverted:

$$\langle r_{\mathrm{opp}}(\alpha, \Delta t, \Delta x) \rangle = -\langle r_{\mathrm{opp}}(-\alpha, \Delta t, \Delta x) \rangle$$

Therefore, for an opponent signal, inverting the correlation is equivalent to inverting the direction of the signal:

$$\langle r_{\mathrm{opp}}(-\alpha, \Delta t, \Delta x) \rangle = \langle r_{\mathrm{opp}}(\alpha, \Delta t, -\Delta x) \rangle$$

For any set of filters, as long as they are opponently subtracted, inverting the sign of the correlation is identical to inverting the direction of the stimulus, when computing the spatiotemporal average response. So when stimuli are generated with autocovariance structures like those in the ternary scintillator (Fig. 1) or in a random dot kinematogram (Fig. 2), if the computation is based on pairwise correlations and is opponent, the equations above show that the response will always be inverted when the stimulus correlation is inverted and will always be equivalent to inverting the direction of the stimulus. Opponency therefore implies the sort of inversion symmetries we observed in our data, where inverting the correlation sign generates percepts with the same tuning as inverting the direction of the stimulus (Fig. 4d,e, but also visible in Figs. 1–3). Opponency also implies the sort of consistent symmetries between positive and negative correlation stimuli observed in human motion perception[38].

### Reporting summary
Further information on research design is available in the Nature Portfolio Reporting Summary linked to this article.

## Data availability
The raw data from the experiments as well as the data displayed in each figure are available via Dryad at https://doi.org/10.5061/dryad.hmgqnk9w8 (ref. 112).

## Code availability
The code for running all experiments and analyses and for generating the figures is available via GitHub at https://github.com/ClarkLabCode/humanAuditoryCorrelations.

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

## Acknowledgements

This work was funded by a grant from the Wu Tsai Institute at Yale University. D.A.C. and this work were funded by NIH R01 EY026555. We thank R. Aslin, E. V. Clark, H. H. Clark, L. Maisuradze, I. Yildirim and J. Zavatone-Veth for helpful discussions and comments on this project.

## Author contributions

P.A.V. and D.A.C. designed the auditory stimuli. P.A.V. and S.D.M. acquired the data. P.A.V., S.D.M. and D.A.C. analysed and interpreted the data. P.A.V., S.D.M. and D.A.C. wrote the paper.

## Competing interests

The authors declare no competing interests.

## Additional information

**Extended data** is available for this paper at https://doi.org/10.1038/s41562-025-02371-7.

**Correspondence and requests for materials** should be addressed to Samuel D. McDougle or Damon A. Clark.

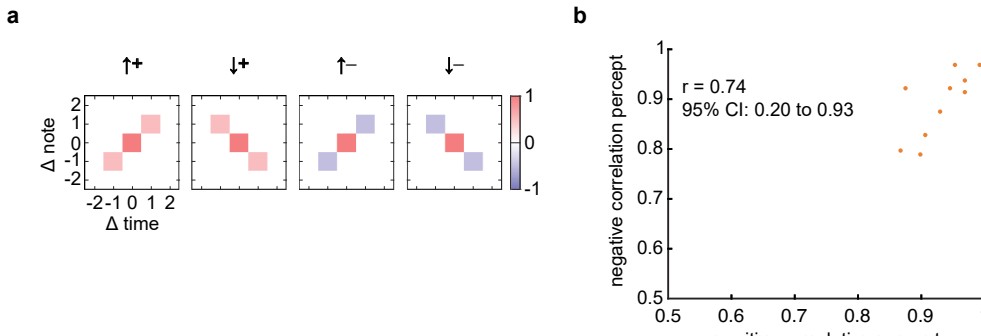

Extended Data Fig. 1 | Stimulus autocorrelation plots and correlation between perception of positively and negatively correlated stimuli. a) Stimulus autocorrelation plots at different note and time offsets for the stimuli in Fig. 1b. The stimuli have positive or negative intensity correlations at a single spectrotemporal offset, directed either upward or downward in frequency over time. These plots are normalized so that the origin has correlation of 1. b) Correlation between perception of positively correlated and negatively correlated stimuli. To obtain the positive correlation percept values, we averaged P(rising) for the upward-directed, positive correlation stimuli with 1-P(rising) for the downward-directed, positive correlation stimuli. To obtain the negative correlation percept values, we averaged P(rising) for the downward-directed, negative correlation stimuli with 1-P(rising) for the upward-directed, negative correlation stimuli. Correlation coefficient is the Pearson correlation, and a 95% confidence interval is noted.

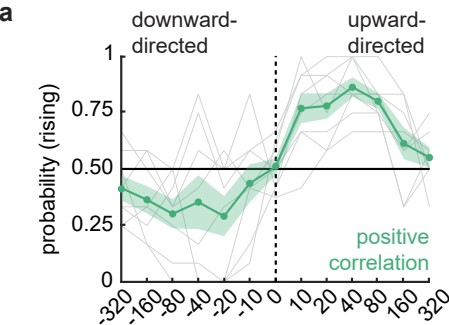

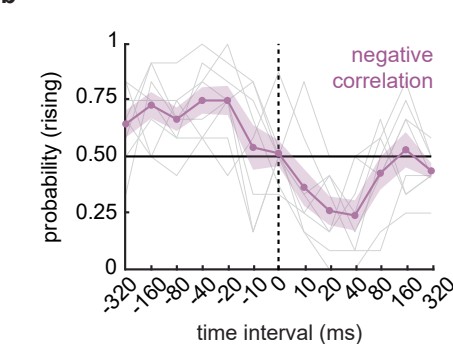

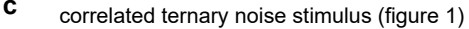

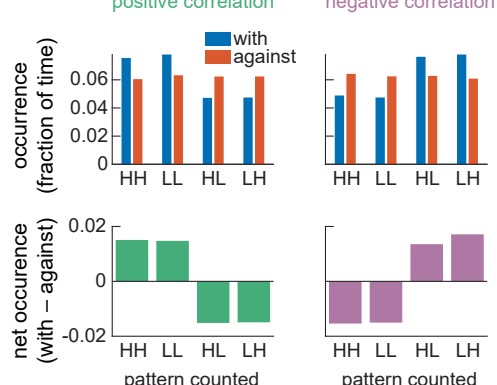

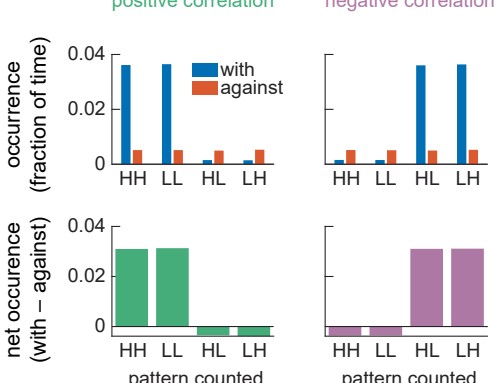

**Extended Data Fig. 2 | Interval sweep with a different pip duration. a**) Perceived direction of positively correlated stimuli with varying pip delays and 20 ms pips. Sensitivity tends to peak around 40 ms delays, similar to the data in Fig. 2c. A one-way, repeated measures ANOVA revealed significantly different responses across pip delays (p < 0.001). Error shading represents ± SEM (N = 8 participants). **b**) Perceived direction of negatively correlated stimuli with varying pip delays and 20 ms pips. Sensitivity tends to peak around 40 ms delays, similar to the data in Fig. 2c. A one-way, repeated measures ANOVA revealed significantly different responses across pip delays (p < 0.001). Error shading represents ± SEM (N = 8 participants). **c**) To assess a short-timescale feature tracking heuristic, we examined how often specific patterns of high and low intensity sounds occurred in the correlated noise stimulus used in Fig. 1b–d. At a frequency offset of 1 note and time offset of 40 ms, we measured how often there appeared patterns with high-high (HH), low-low (LL), high-low (HL), and low-high (LH) intensity

patterns oriented either in the direction of the imposed correlation ('with') or in the opposite direction ('against') (see **Methods**). Below, we measured the net occurrence by subtracting the 'against' occurrence value from the 'with' occurrence value. The high-high tracking heuristic could be used to predict the positive correlation percept, since the net occurrence of HH is positive. It can also be used to predict the negative correlation percept, since the net occurrence of HH is negative for the negative correlation stimuli. Thus, though the correlation is negative, a heuristic that (somehow) tracked high-high intensity pairs across frequency and time could in principle explain the perceptual inversion. **d**) As in (**c**) but for the correlated pip stimuli in Fig. 2. Here, in the negative correlation stimulus, the high-high (HH) intensity pairs show about 1/10 the signal of the two negative correlation pairs. This means that the high-high intensity tracking heuristic has difficulty explaining the perceived direction, while the direct detection of intensity correlations is a parsimonious explanation of the data.

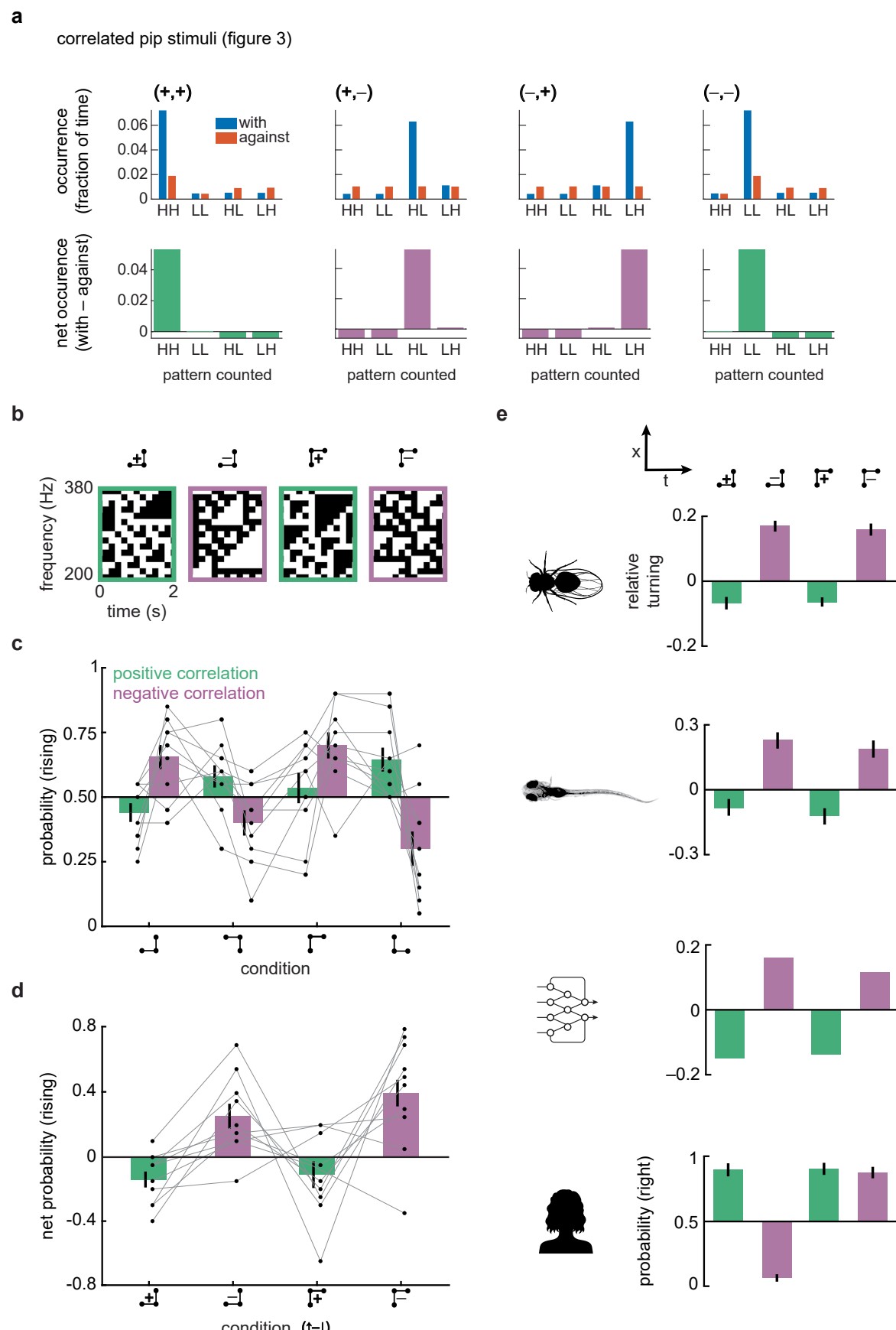

**Extended Data Fig. 3 | See next page for caption.**

**Extended Data Fig. 3 | Human auditory sensitivity to 3-point glider stimuli resembles visual sensitivity in different species. a**) Measurement of specific tracking patterns as in Extended Data Fig. 2c, d, but for the four correlated pip stimuli presented in Fig. 3. The tracking heuristics, which could look for specific patterns of stimuli, like high-high ('HH') intensity pairs offset in frequency and time, cannot easily be used to assess the negative correlation stimuli. This is clear in the occurrence of these patterns in the direction of the correlation ('with') and in the opposite direction ('against'), as well as in the difference between the with and against occurrences, measured in the net occurrence below each pattern. These net occurrences are about ten times larger for the pattern of enforced correlation than for the high-high pairs (or other non-enforced pairs). This suggests that such tracking heuristics would have difficulty explaining the measured percepts, while correlation sensitivity is a more parsimonious explanation. **b**) Diagram of 3-point glider stimuli in frequency and time[42]. 3-point glider stimuli contain correlations in intensity between triplets of points in frequency and time as denoted by the barbell diagrams, and contain no pairwise correlations in intensity at any time or frequency offsets. Thus, any motion percepts with these stimuli would have to rely on correlations beyond pairwise ones. All four examples are upward directed, since the hypotenuse

of the correlation pattern is directed upward. The left two are 'diverging', since a single early point correlates with two subsequent ones; the right two are 'converging', since two early points correlate with one subsequent one. **c**) Perceived direction of 3-point glider stimuli. Participants heard rising and falling tones in these triplet correlation stimuli. Error bars represent mean ± SEM (N = 10 participants). **d**) Net perceived direction of 3-point glider stimuli with positive and negative correlations. The net probability rising is computed by subtracting the downward-directed P(rising) from the upward-directed P(rising) in panel (**c**), where the direction of the correlation is the direction of each triangle's hypotenuse. Positively correlated stimuli were perceived as net falling, while negatively correlated stimuli were perceived as net rising. Paired t-tests revealed significantly different responses to positively and negatively correlated diverging gliders, and to positively and negatively correlated converging gliders (all $p$s < 0.001). Error bars represent mean ± SEM (N = 10 participants). **e**) Net perceived direction of 3-point glider stimuli across various visual systems. Data is replotted from prior publications for fruit flies[44], larval zebrafish[46], a machine learning algorithm[63], and human visual psychophysics[44]. Human auditory percepts resemble fruit fly and zebrafish visual percepts and machine learning responses, but not the human visual percepts.

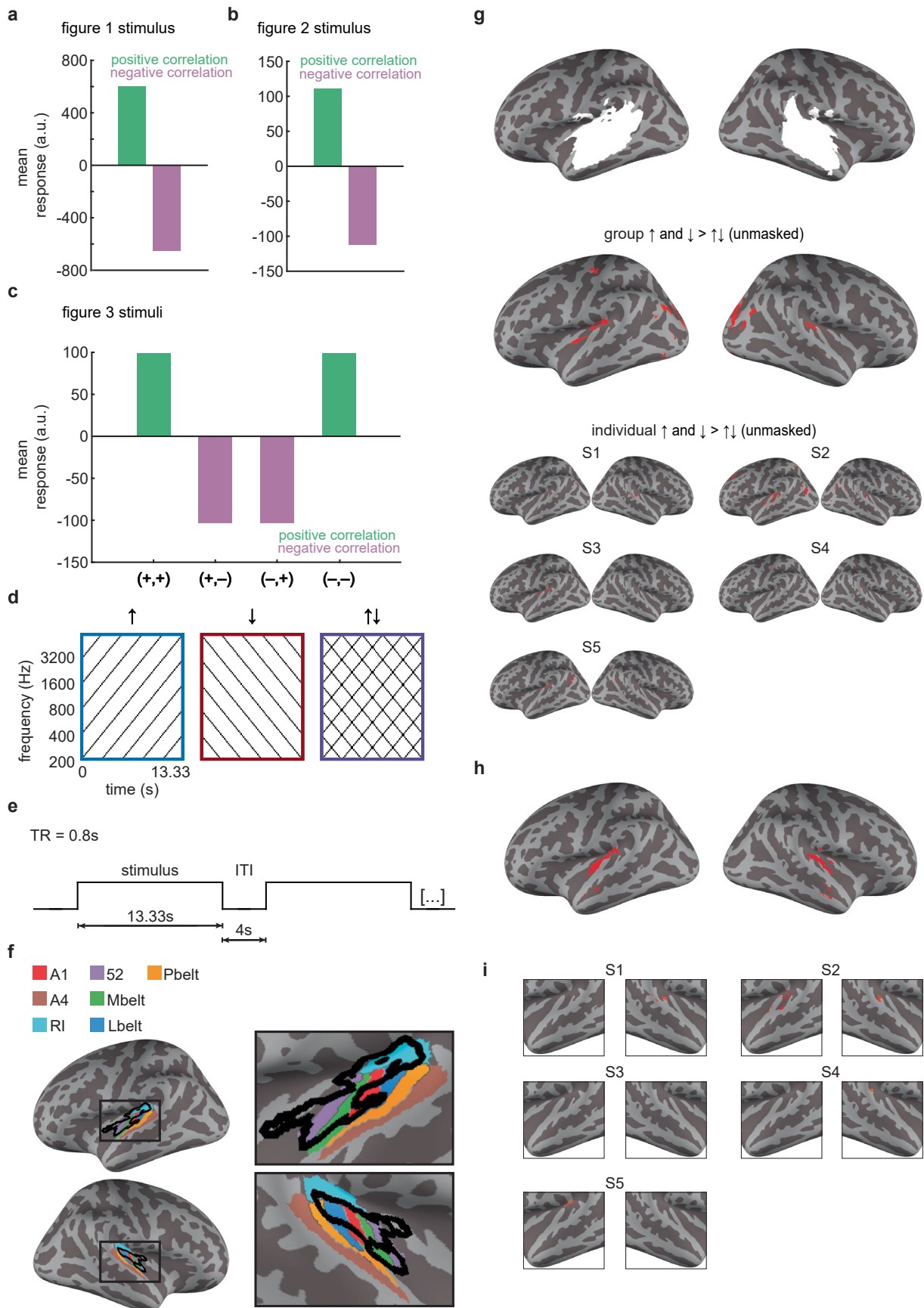

**Extended Data Fig. 4 | See next page for caption.**

**Extended Data Fig. 4 | Further method and result details from fMRI opponency experiment. a)** Mean responses to stimuli for the opponent motion energy model in Fig. 4c. When the model is presented with the ternary stimuli in Fig. 1, it responds positively (negatively) to the positive (negative) correlation stimuli. **b)** When the model is presented with the correlated pip stimuli in Fig. 2, it responds positively (negatively) to the positive (negative) correlation stimuli. **c)** When the model is presented with the correlated pip stimuli in Fig. 3, it responds positively to both the high-high and low-low positive correlation stimuli, and negatively to the high-low and low-high negative correlation stimuli. **d)** Depiction of actual stimuli used for the opponency experiment. **e)** Time course of fMRI trial structure. **f)** Group level analysis showing bilateral regions within auditory cortex that demonstrate significant opponent properties. Black outline reflects significant clusters from Fig. 4h. Colored patches show cortical regions in accordance with[57]. *RI* = retroinsular cortex; *Mbelt* = medial belt of auditory cortex; *Lbelt* = lateral belt of auditory cortex; *Pbelt* = parabelt region. **g)** *A priori* anatomical mask of auditory cortex (primary, non-primary, and STG regions) used in Fig. 4h, i (*top*). This liberal mask includes all voxels passing a 50% probability threshold of residing in Heschl's gyrus or any region of the STG according to the Harvard-Oxford probabilistic atlas. Unmasked group analysis of data from Fig. 4h (*middle*). Thresholded at $p < 0.05$ (two-sided) with a cluster-forming constraint of 20 voxels. Unmasked individual analysis of data from Fig. 4i (*bottom*). Thresholded at $p < 0.05$ (two-sided) with a cluster-forming constraint of 20 voxels. **h)** Group results of a non-parametric permutation-based analysis (see **Methods**). Shown clusters are larger than a permutation-based null distribution of cluster sizes with $p < 0.05$. The largest cluster had $p = 0.03$ by this analysis. **i)** Individual results of a non-parametric permutation-based analysis (see **Methods**). For each individual, shown clusters are those that are greater than or equal to the 90th percentile of the largest clusters in a null distribution generated by permuting stimulus labels (that is, $p < 0.1$). For the five individuals, the overall p-values of the largest cluster by this analysis were (in order): 0.02, 0.01, 0.30, 0.08, 0.10.

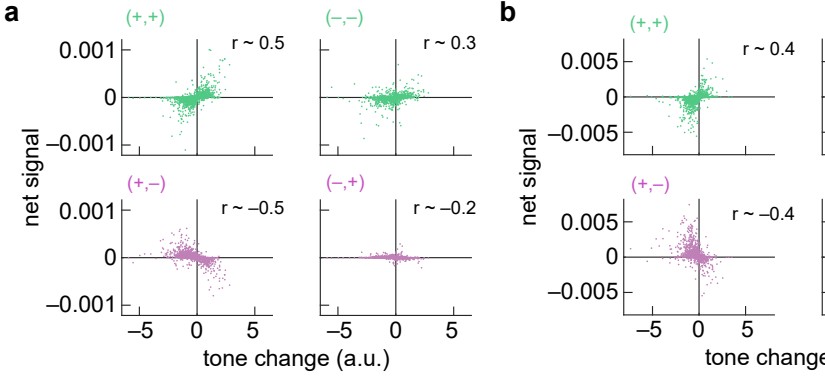

**Extended Data Fig. 5 | Multiplicative interactions of amplitude derivatives are informative about intonation direction. a**) Correlations between the tone change estimate at each time and a continuous correlator model using only positive signals (+,+), only negative signals (−,−), and mixtures of the two (+,− and −,+) (see **Methods**). The correlations comprising the net signal were obtained by taking the derivative of the spectrogram amplitude in time, then multiplying derivatives of neighboring frequencies with a time-step delay and subtracting a mirror image product. Signals were rectified before multiplication to obtain the four pairs of multiplied signals, which together add up to a full correlator model. The net signals computed from (+,+) and (−,−) pairs correlated positively with tone change, while the net signals from (+,−) and (−,+) pairs correlated negatively with tone change. Data from English speech corpus[102]. **b**) As in (**a**) but with data from Mandarin speech corpus[103].

# Reporting Summary

## Statistics

For all statistical analyses, confirm that the following items are present in the figure legend, table legend, main text, or Methods section.

| n/a | Confirmed | |
|---|---|---|
| ☐ | ☒ | The exact sample size ($n$) for each experimental group/condition, given as a discrete number and unit of measurement |
| ☒ | ☐ | A statement on whether measurements were taken from distinct samples or whether the same sample was measured repeatedly |
| ☐ | ☒ | The statistical test(s) used AND whether they are one- or two-sided <br> *Only common tests should be described solely by name; describe more complex techniques in the Methods section.* |
| ☐ | ☒ | A description of all covariates tested |
| ☒ | ☐ | A description of any assumptions or corrections, such as tests of normality and adjustment for multiple comparisons |
| ☐ | ☒ | A full description of the statistical parameters including central tendency (e.g. means) or other basic estimates (e.g. regression coefficient) AND variation (e.g. standard deviation) or associated estimates of uncertainty (e.g. confidence intervals) |
| ☒ | ☐ | For null hypothesis testing, the test statistic (e.g. $F$, $t$, $r$) with confidence intervals, effect sizes, degrees of freedom and $P$ value noted <br> *Give P values as exact values whenever suitable.* |
| ☒ | ☐ | For Bayesian analysis, information on the choice of priors and Markov chain Monte Carlo settings |
| ☒ | ☐ | For hierarchical and complex designs, identification of the appropriate level for tests and full reporting of outcomes |
| ☒ | ☐ | Estimates of effect sizes (e.g. Cohen's $d$, Pearson's $r$), indicating how they were calculated |

*Our web collection on statistics for biologists contains articles on many of the points above.*

## Software and code

Policy information about availability of computer code

| | |
|---|---|
| Data collection | Custom code was written to collect data in PsychToolbox 3.0.18 and run in Matlab 2021b. It is available here: https://github.com/ClarkLabCode/humanAuditoryCorrelations and here: https://doi.org/10.5061/dryad.hmgqnk9w8. |
| Data analysis | Custom code was written to analyzed psychophysics data. It is available here: https://github.com/ClarkLabCode/humanAuditoryCorrelations and here: https://doi.org/10.5061/dryad.hmgqnk9w8. fMRI code was analyzed using JupyterLab and Nilearn packages. It is available here: https://doi.org/10.5061/dryad.hmgqnk9w8. |

For manuscripts utilizing custom algorithms or software that are central to the research but not yet described in published literature, software must be made available to editors and reviewers. We strongly encourage code deposition in a community repository (e.g. GitHub). See the Nature Portfolio guidelines for submitting code & software for further information.

## Data

Policy information about availability of data

All manuscripts must include a data availability statement. This statement should provide the following information, where applicable:
- Accession codes, unique identifiers, or web links for publicly available datasets
- A description of any restrictions on data availability
- For clinical datasets or third party data, please ensure that the statement adheres to our policy

All psychophysics data is freely available through the links in the paper. Raw fMRI data is available from the Lead Authors on request.

# Research involving human participants, their data, or biological material

Policy information about studies with human participants or human data. See also policy information about sex, gender (identity/presentation), and sexual orientation and race, ethnicity and racism.

| | |
|---|---|
| Reporting on sex and gender | Participant sex is reported. |
| Reporting on race, ethnicity, or other socially relevant groupings | na |
| Population characteristics | Age statistics are reported. |
| Recruitment | Participants were recruited by the first author from among students at the University. |
| Ethics oversight | The study was run under a protocol approved by the Yale Institutional Review Board. |

Note that full information on the approval of the study protocol must also be provided in the manuscript.

# Field-specific reporting

Please select the one below that is the best fit for your research. If you are not sure, read the appropriate sections before making your selection.

☐ Life sciences  ☒ Behavioural & social sciences  ☐ Ecological, evolutionary & environmental sciences

For a reference copy of the document with all sections, see nature.com/documents/nr-reporting-summary-flat.pdf

# Behavioural & social sciences study design

All studies must disclose on these points even when the disclosure is negative.

| | |
|---|---|
| Study description | Data are quantitative. |
| Research sample | Yale University students. |
| Sampling strategy | Sampling was not random; participants were recruited by the authors. |
| Data collection | A laptop computer recorded psychophysical decisions. A 3T fMRI machine collected fMRI data. |
| Timing | Spring and Fall 2022, and Spring 2023 |
| Data exclusions | Data were excluded as noted in the methods. |
| Non-participation | No participants began but did not finish the experiments. |
| Randomization | There was no assignment to experimental and control groups. |

# Reporting for specific materials, systems and methods

We require information from authors about some types of materials, experimental systems and methods used in many studies. Here, indicate whether each material, system or method listed is relevant to your study. If you are not sure if a list item applies to your research, read the appropriate section before selecting a response.

## Materials & experimental systems

| n/a | Involved in the study |
|---|---|
| ☒ | ☐ Antibodies |
| ☒ | ☐ Eukaryotic cell lines |
| ☒ | ☐ Palaeontology and archaeology |
| ☒ | ☐ Animals and other organisms |
| ☒ | ☐ Clinical data |
| ☒ | ☐ Dual use research of concern |
| ☒ | ☐ Plants |

## Methods

| n/a | Involved in the study |
|---|---|
| ☒ | ☐ ChIP-seq |
| ☒ | ☐ Flow cytometry |
| ☐ | ☒ MRI-based neuroimaging |

# Plants

| | |
|---|---|
| Seed stocks | na |
| Novel plant genotypes | na |
| Authentication | na |

# Magnetic resonance imaging

## Experimental design

| | |
|---|---|
| Design type | No active tasks were assigned. Data was collected to measure passive responses to specific auditory stimuli, presented as described in the methods. |
| Design specifications | slow event related design; Fifteen auditory stimuli were presented per run in an event-related design (5 each of three stimulus types: rising, falling, and summed). Each stimulus lasted for 13.33 s, separated by an inter-trial interval (ITI) of 4 s. The order of the three stimulus types was randomized in each run. Participants passively listened to the tones and were not required to render any responses. |
| Behavioral performance measures | None during fMRI. |

## Acquisition

| | |
|---|---|
| Imaging type(s) | Functional and antomical |
| Field strength | 3T |
| Sequence & imaging parameters | Whole-brain imaging was performed at the Brain Imaging Center at Yale University, on a Siemens 3 T Prisma MRI scanner using a 32-channel head coil. Functional data were acquired with a gradient-echo echoplanar pulse sequence (TR = 0.80 s, TE = 30 ms, flip angle = 52°, voxel size = 2.4 mm × 2.4 mm × 2.4 mm, MB acc. factor = 6). T1-weighted MP-RAGE anatomical images were collected as well (TR = 2.5 s, TE = 2.0 ms, flip angle = 8°, 208 slices, voxel size = 1.0 mm isotropic). |
| Area of acquisition | Whole brain scan. |

Diffusion MRI  ☐ Used  ☒ Not used

## Preprocessing

| | |
|---|---|
| Preprocessing software | The fMRI-Prep toolbox was used for preprocessing (Esteban et al. 2019). The anatomical image was corrected for intensity non-uniformity (INU) with N4BiasFieldCorrection (Tustison et al. 2010) and used as T1w-reference. The T1w-reference was then skull-stripped with a Nipype implementation of the antsBrainExtraction.sh workflow in ANTs, and tissue segmentation of cerebrospinal fluid (CSF), white-matter (WM), and gray-matter (GM) was performed on the brain-extracted T1w using FFAST (FSL 6.0.5) (Zhang et al. 2001). Volume-based spatial normalization to standard (MNI) space was performed through nonlinear registration with antsRegistration (ANTs 2.3.3). For each of the BOLD runs, a reference volume and its skull-stripped version were generated using a custom methodology of fMRIPrep. Head-motion parameters were estimated using MCFLIRT (FSL 6.0.5) (Jenkinson et al. 2002) and BOLD time-series were resampled into native space by applying the transforms to correct for head-motion, and the BOLD reference was co-registered to the anatomical reference using mri_coreg (FreeSurfer) followed by FLIRT. Co-registration was configured with 6 DOF. Several confounding time-series were calculated based on the preprocessed BOLD: framewise displacement (FD), DVARS and three region-wise global signals. The BOLD time-series were resampled into standard space, and volumetric resamplings were performed using ANTs. |
| Normalization | as above: The anatomical image was corrected for intensity non-uniformity (INU) with N4BiasFieldCorrection (Tustison et al. 2010) and used as T1w-reference. The T1w-reference was then skull-stripped with a Nipype implementation of the antsBrainExtraction.sh workflow in ANTs, and tissue segmentation of cerebrospinal fluid (CSF), white-matter (WM), and gray-matter (GM) was performed on the brain-extracted T1w using FFAST (FSL 6.0.5) (Zhang et al. 2001). Volume-based spatial normalization to standard (MNI) space was performed through nonlinear registration with antsRegistration (ANTs 2.3.3). |
| Normalization template | MNI |
| Noise and artifact removal | Confound regressors of no interest (generated using fMRIPrep, see above) were entered into each GLM. These included six standard motion regressors, the framewise displacement time course, and white matter and global signal time courses. |

| Volume censoring | na |
|---|---|

## Statistical modeling & inference

| Model type and settings | Our main analyses involved constructing general linear models (GLMs) to quantify the effects of the three stimulus types within auditory cortex. Univariate GLM analyses were performed using Nilearn (Abraham et al. 2014). Confound regressors of no interest (generated using fMRIPrep, see above) were entered into each GLM. These included six standard motion regressors, the framewise displacement time course, and white matter and global signal time courses. Each stimulus type (rising, falling, and summed) was modeled using boxcar regressors over the entire stimulus presentation phase (13.33 s) of the relevant trials, and was convolved with the canonical double-gamma hemodynamic response function. The main contrast of interest at the group and individual levels compared BOLD responses to the non-summed directional stimuli (i.e., rising and falling) to the summed stimuli (i.e., superimposed rising + falling). The contrast was designed to highlight deviations from a null hypothesis of equivalent responses between directional and opponent stimuli. |
|---|---|
| Effect(s) tested | Univariate contrast of directional pitch stimuli trials versus combined opponent stimuli |

Specify type of analysis:  ☐ Whole brain  ☒ ROI-based  ☐ Both

| Anatomical location(s) | We used a probabilistic atlas to restrict analysis to broad temporal auditory areas, as described in the methods. |
|---|---|

Statistic type for inference

(See Eklund et al. 2016)

| | cluster-wise |
|---|---|

| Correction | non-parametric/permutation tests |
|---|---|

## Models & analysis

| n/a | Involved in the study |
|---|---|
| ☒ | ☐ Functional and/or effective connectivity |
| ☒ | ☐ Graph analysis |
| ☐ | ☒ Multivariate modeling or predictive analysis |

| Multivariate modeling and predictive analysis | standard univariate GLMs |
|---|---|

