## [Peer Review File · Nature Human Behaviour]

Humans can use positive and negative spectrotemporal correlations to detect rising and falling pitch

Corresponding Author: Professor Damon Clark

Version 0:

Decision Letter:

22nd January 2025

Dear Prof Clark,

Thank you once again for your manuscript, entitled "Humans can use positive and negative spectrotemporal correlations to detect rising and falling pitch", and for your patience during the peer review process.

Your Article has now been evaluated by 3 referees. You will see from their comments copied below that, although they find your work of potential interest, they have raised quite substantial concerns. In light of these comments, we cannot accept the manuscript for publication, but would be interested in considering a revised version if you are willing and able to fully address reviewer and editorial concerns.

We hope you will find the referees' comments useful as you decide how to proceed. If you wish to submit a substantially revised manuscript, please bear in mind that we will be reluctant to approach the referees again in the absence of major revisions. We are committed to providing a fair and constructive peer-review process. Do not hesitate to contact us if there are specific requests from the reviewers that you believe are technically impossible or unlikely to yield a meaningful outcome.

In particular, the referees raise concerns about the advance of your model compared to prior work, and remark on the importance of situating these findings in the context of the broader literature. We ask that you address these issues in full.

If you wish to submit a suitably revised manuscript, we would hope to receive it within 4 months. I would be grateful if you could contact us as soon as possible if you foresee difficulties with meeting this target resubmission date.

- Include a "Response to the editors and reviewers" document detailing, point-by-point, how you addressed each editor and referee comment. If no action was taken to address a point, you must provide a compelling argument. When formatting this document, please respond to each reviewer comment individually, including the full text of the reviewer comment verbatim followed by your response to the individual point. This response will be used by the editors to evaluate your revision and sent back to the reviewers along with the revised manuscript.
- Highlight all changes made to your manuscript or provide us with a version that tracks changes.

Link Redacted

Thank you for the opportunity to review your work. Please do not hesitate to contact me if you have any questions or would like to discuss the required revisions further.

Sincerely,

Reviewer expertise:

Reviewer #1: Auditory processing, neuroimaging, modeling

Reviewer #2: Auditory processing, neuroimaging, modeling

Reviewer #3: Auditory processing, neuroimaging, modeling

REVIEWER COMMENTS:

Reviewer #1 (Remarks to the Author):

This paper explores the cues that humans use to perceive pitch changes. The study uses psychophysics, computational modeling, functional neuroimaging, and speech analysis to investigate if humans detect pitch motion similarly to visual motion. The authors find that humans can judge pitch direction based on positive or negative spectrotemporal intensity correlations. This ability is analogous to the visual "reverse-phi" motion illusion, suggesting a new auditory illusion. fMRI results support the hypothesis that auditory processing involves pitch direction opponency, and an analysis of English and Mandarin speech shows that pitch direction is signaled by both types of correlations, indicating ecological benefits. Overall, the research suggests that the central nervous system uses "motion detection" algorithms sensitive to local correlations for processing pitch across different sensory modalities.

This paper is well-written, the experiments flow well from one another, and it touches on a fundamental question in hearing that will be interesting to a broad audience. I appreciated the integration of the visual and auditory approaches.

My main concern with this paper is that the authors have not fully engaged with the existing pitch literature, therefore, it is unclear how novel these results are. There is scope to connect these findings to the existing pitch literature to situate them more convincingly in the current understanding of pitch.

- The authors cite several examples of spectrotemporal modulations being used for pitch processing; however, their citations are outdated. New work from Mounya Elhilali (some summarized in her 2019 chapter in Timbre) and Shihab Shamma (Shamma and Dutta, 2019, for example) should be integrated. The current paper's spectrotemporal stimuli appear to be an extension of existing work, rather than a novel application of this approach.
- Older work from Loeb and colleagues also shows how spatial cross-correlation on the cochlea could account for pitch processing – this could conceivably explain the current results (Loeb, G. E., White, M. W., & Merzenich, M. M. (1983). Spatial cross-correlation: A proposed mechanism for acoustic pitch perception. Biological cybernetics).
- Likewise, recent work from Laurel Carney demonstrating that neural fluctuation contrast may help in the decoding of complex sound is germane.
- It is unclear how a pitch model relating to intensity correlations connect with the existing understanding of pitch computations, which appear to be limited by the coding strategies of the periphery and influenced by the statistics of natural sounds. For example, recent work from Mark Saddler (Saddler, Gonzalez, and McDermott, "Deep neural network models reveal the interplay of peripheral coding and stimulus statistics in pitch perception") offers a review.
 - o Again, I believe the findings in this paper align well with previous work that properties of pitch computations reflect adaptations to natural sound statistics. However, this could be better connected to existing work in the discussion.
- It would be worth discussing how these results relate to findings with iterated ripple noise, particularly findings that cortical responses to IRN are driven by slowly varying spectro-temporal modulations instead of pitch. Is it possible you are picking up on neural signals that are not pitch-related and instead just relate to spectro-temporal modulations?
 - o Iterated ripple noise is, admittedly, a strange type of stimulus, but so are the stimuli used in this experiment. They bear some superficial similarities and I would have appreciated a deeper engagement with that literature.
- How do the locations where you observed your opponency effects relate to classic pitch-selective voxels in the auditory cortex? (See work of Allen and Oxenham (<https://www.jneurosci.org/content/jneuro/42/3/416.full.pdf>), or Haignere, Kanwisher & McDermott - <https://pubmed.ncbi.nlm.nih.gov/24336712/>). Likewise, work from Claire Tang and Eddie Chang using ECoG has shown voice pitch-selective neurons that, if the voxels you identified are related to speech, might be co-located. Connections to previous fMRI work on pitch was, in general, lacking.

In general, the discussions of the neural basis of pitch, and the pitch literature in general, struck me as outdated and incomplete. Hopefully the references here will help.

Other concern:

- fMRI safe headphones are generally less linear than regular headphones, causing distortions (<https://pmc.ncbi.nlm.nih.gov/articles/PMC4803580/>), could this have influenced your results? Particularly with noise cancellation? You said that participants could not distinguish the correlated intensity stimuli – how was this determined, as the participants had no responses? Were the other stimuli equally distinguishable for participants? Could this have influenced the results?

Smaller comments:

- What are the lines on the bar graphs? There are not enough to be individual participants? If this was signaled, I missed it, so it should be more clearly marked.

- What were the examples that participants were shown before the experiment? The stimuli do not really sound like 'tones', so asking the participants to identify whether they sound like they are rising or falling tones seems like odd instructions. (The proof that the instructions made sense is demonstrated in the interpretable pattern of results), but I am wondering why the authors chose this set of instructions.
- For all experiments, were the participants musicians? Were there differences between musicians and non-musicians?

Reviewer #2 (Remarks to the Author):

The manuscript describes a novel kind of auditory stimulus designed to introduce local "spectro-temporal correlations" within a sound's spectra. The idea is directly inspired by visual stimuli that elicit spatial motion percepts, which the aim of uncovering similar functional mechanisms in the two modalities. In a series of psychophysical experiments, it is shown that listeners report upward- or downward- pitch changes associated with the nature and strength of the spectro-temporal correlations. A computational model suggests that frequency motion direction opponency can account for the behavioral results. This conclusion is further strengthened by an fMRI study, with a different set of stimuli (upward frequency sweeps, downward sweeps, and their combination). Finally, acoustic analyses of speech databases show that frequency changes induce spectro-temporal correlations, which is taken as an argument for an ecological role of such correlations.

I found the study fascinating and thought-provoking. The translation of a visual phenomenon to audition can certainly be illuminating and has the potential to highlight deep parallels across sensory modalities. The diversity of the evidence provided in the manuscript is also impressive, with psychophysical results complemented by a computational model, brain imaging data, and acoustic analyses of natural sounds.

Now, for the issues I'd like to see addressed: I would like to stress that I am writing this review with the perspective of an auditory scientist. As such, I must admit it took me a few re-reads to try and recast the features of the stimuli in a language that I am more familiar with. This included putting the sound examples from the Supplemental movies (very helpful, thanks) through various kinds of auditory models, including the STRFs cited in the manuscript. Which led to the main point I'd like to raise: can the behavioral and neural results be explained by existing models of auditory processing, or do they absolutely require the novel framework developed in the manuscript?

I will detail this general question below, so the authors have a chance to refute all specific points if possible. My perspective is really that I'd like to be convinced that the novel framework is warranted, but I'd like to be reassured that it is not a rephrasing of well-established auditory models. Or if it is, that this is made clear to highlight the other contributions of this work.

Specific comments

- My main question is whether the stimulus design procedure used here introduces local frequency-shifts between spectral components, which could then be used to predict the behavioral results. An option to answer this question would be to try to fit the results with existing auditory models. For instance, cochleograms could be computed and provided to complement the schematic illustrations of the stimuli; then local frequency shifts could be extracted; then the amount of 'up' versus 'down' shifts compared. This is more or less Allik et al. 1989's original suggestion, and also the idea behind the frequency-shift detector model (both appropriately cited in the manuscript). The extraction of frequency shifts could be achieved by STRF models, as also cited in the manuscript. I realize that this suggestion is a big ask, and that there is no strong incentive for the authors to find an alternative, less exciting model to account for their results. But I think it would really clarify the novelty of the findings.

- L21. "Importantly, work on such spectral pattern cues typically employs sounds with persistent groupings of tones". It is not clear to me what is meant by persistent, here and elsewhere (the word is certainly not common in the auditory jargon). How long would qualify as persistent? In the work cited in the manuscript (Demany and Ramos 2005 plus follow-ups), there was a single short random chord followed by a single short tone. How is this different in terms of 'persistence' to the frame-by-frame correlation introduced here, at least for the positive correlation cases?

- L67. "stimuli that use increments and decrements in intensity to generate local correlations in intensity at specific offsets in frequency and time (see Methods)." This could be expanded, as unpacking the meaning of the sentence is critical to appreciate the rest of the manuscript. For instance, it could help to further specify which vector/2D matrix is correlated with what, what are the numerical ranges of correlations obtained with the stimuli used in the experiment, etc...

- L93. "We emphasize that in these negative correlation stimuli, there do not exist spectral patterns that persist between frames.". This is, I believe, the critical claim for the whole manuscript. But I am not sure I understand why this is a priori obvious. The schematics include 'white spots' suggesting absence of sound, but wouldn't the relevant correlations for perception be between the remaining 'black spots' or 'gray spots'? Could it be that a side effect of the negative correlation design is to introduce local frequency-shifts between the remaining spectral components, in the opposite direction and consistent with the subjective judgments? Eyeballing the cochleograms of the stimuli and computing a few STRFs suggested that it is not out of the question. Eyeballing is no strong argument (thankfully!) but a proper quantification of such putative shifts would help tremendously to strengthen your point, I think.

- The binaural experiment is both elegant and interesting. I understand the results as showing that both the FSD/correlation detectors and the putative opponency operation must be post binaural convergence. Would you agree? In this case, it is still possible that each happen at different stages of processing. For instance, there is plenty of neural evidence for FM detectors subcortically (in the inferior colliculus for instance), but as far as I know, no evidence for opponency. Would a two-stage model still be consistent with your findings?

- The next experiment, involving tone pips, is used to measure tuning characteristics. I am not sure I understand why it was necessary to change the stimulus design procedure for that purpose. Couldn't you adjust the parameters of the stimuli of Experiment 1?

- L207: The sensitivity for the correlation-detection mechanism peaks at very similar parameters than what was measured for FSDs in the literature. Isn't this a tantalizing hint that both mechanisms could in fact be one and the same thing?

- L234. The triplet correlation experiment, again inspired by visual studies, is very succinctly described. I am not sure I understand what it brings to the core argument.

- L265. "Crucially, however, in this model, negatively correlated stimuli exhibit a different tuning from oppositely directed positive stimuli; [...]". I agree this is crucial, but I am not sure I see it in Fig 4b, which looks quite symmetrical. It could be useful to provide the detailed predictions of a model without opponency, so the benefit of adding the second stage becomes obvious.

- fMRI experiment. The nice novelty of this experiment is to contrast responses of upward or downward frequency glides to responses of the same but superimposed up and down glides. While this has never been analyzed in terms of an opponency model, there has been at least one previous study using similar stimuli: Overath et al., J Neurosci, 2010. I think it could be relevant to cite, and if possible at all (given that their task was very different), compare with your results.

- Fig 4g, stimuli. I agree that, if one assumes an opponency model, your predictions are fully warranted. However, I wonder if the same predictions could be made without any need for an opponency mechanism. For example, current neural models of FM detection involve delayed inhibition to select for one FM direction (following the work of Suga 1965, admittedly mostly in bats). Wouldn't such a delayed inhibition mechanism also produce a reduced response for the superimposed stimulus?

- L382. The use of an optical flow algorithm to evaluate the amount of 'tone change' over time is yet another example of the potential benefit of getting inspiration from visual science, very clever. Would it be interesting to apply the algorithm to the stimuli used in your experiment, and not only for the acoustic database analysis? Could this quantify the amount of local frequency-shifts in the stimuli?

- The amount of 'tone change' is found to be correlated to the presence of spectro-temporal correlations. Is this really surprising? I wonder if it is not an inescapable consequence of having harmonic sounds, as changes in f_0 will necessarily translate to many structured changes in correlations all over the spectrum. If so, then focusing on speech databases for the analysis may be slightly misleading. The important prerequisite for the observed correlation is not that ecological sounds are used, but rather that harmonic sounds are included.

- L633. Several sound examples were given before experiment to train participants. I can imagine that the task was tricky to explain, as most pitch-shift tasks usually are. Was there any exclusion criterion? Did all recruited participants manage to pass the informal training phase? If they did, that'd be quite surprising, and worth reporting.

- L659. Waveforms were "scaled to have a minimum value of -1 and maximum value of $+1$." In addition to being slightly unusual (one would usually scale rms values and not max/min amplitude), this does not sound right: such a procedure implies that all waveforms had symmetrical maximum/minimum values, or that they were scaled by introducing a non-zero DC. It is likely completely inconsequential to the results, of course, so just noted here for precision.

L838: the sample size for the fMRI experiment is only clearly stated here: $N=5$, including two of the authors. I am no fMRI expert, but it seems unusual from what I see in the literature. Is this acceptable by current standards?

Reviewer #3 (Remarks to the Author):

Vaziri et al. use novel correlation stimuli in the auditory domain combined with fMRI and acoustic analysis of natural speech to make two main claims:

The first main claim is that humans can use local spectrotemporal correlations to judge whether an acoustic stimulus ("pitch" is used) is rising or falling, even if listeners are prevented from tracking spectral patterns across longer timescales (i.e. successively rising or falling local components). The use of stimuli with negative local spectrotemporal correlations and the fact that listeners perceived these stimuli in the opposing direction as their positively correlated counterparts is the main evidence used for the central claim.

The second main claim, based on fMRI experiments with upwards, downwards, and combined upwards-and-downwards spectral motion, is that spectral motion uses an opponent channel code in auditory cortex.

If true, these results would indeed be novel and interesting and potentially initiate a new line of inquiry in (human) auditory processing. However, I have several concerns that should be addressed by the authors before I can endorse the findings:

1. It's not clear from the data and analyses shown how different the down-negative stimuli (right-most panels of Fig 1B) are from the up-positive stimuli (left-most panels of Fig 1B). I understand how they are different theoretically, but practically speaking, the resulting down-negative stimuli appear to have several spectrotemporal sub-regions with upward spectral motion that could potentially be tracked across time as an auditory stream. Similarly, the up-negative stimuli (3rd column of Fig 1B) seem to have spectrotemporally localized downward spectral motion (like the the down-positive stimuli do). Given how critical this is to the authors' first main claim, I think more needs to be done to convince the reader that these two pairs of classes of stimuli (pair 1: up-positive and down-negative; pair 2: down-positive and up-negative) are substantially different from each other, especially at the level of individual trials where such localized spectrotemporal motion should be characterized.

a. One suggestion would be autocorrelation types of pitch models.

b. Or maybe something here could be useful? <https://amtoolbox.org/>

2. I couldn't find in the manuscript whether feedback was given to listeners during the main psychoacoustic experiment. This should be reported, as it could potentially have a big impact on the behavioral findings.
3. Does the time interval scale of Fig 2C match the offset parameter mentioned in the first paragraph of the Results? That is, does 1/6 of second as mentioned in the first paragraph of the Results align with 166.67 ms on the x-axis of Fig 2C? If so, then aren't there some behavioral inconsistencies between the data shown in Figs 2 and 3? Some clarification here would be appreciated.
4. Related to (3), it seems that the maximum effect size happens for time intervals of ~40 ms. This is largely consistent with other lines of work showing figure popout amongst otherwise random sequences of chords (pop-out was strongest around 25 Hz or 40 ms per chord). See Teki et al. 2013 and related work from Maria Chait's group. This again makes me wonder about auditory stream tracking across time of localized spectrotemporal motion as a possible explanation for the results.
5. The contrast shown in Fig 4H and Fig 4I, which I believe is the average of the response to up-motion and down-motion vs the response to stimuli with simultaneous up- and down-motion should be more clearly indicated on the figure itself.
6. The mask that was used to search for significant clusters should be shown somewhere, at least in the supplement. Additionally, it would help convince the readers regarding the significance of the clusters shown to also include a whole-brain analysis (i.e., without using a mask).
7. The rationale for using different stimuli for the fMRI experiment isn't clear to me. Was it mainly for simplicity of testing the opponency idea in fMRI? In any case, it would be useful for the reader to be able to hear those stimuli for themselves, as the authors have done for the stimuli used in the psychophysical and modeling studies.
8. Related to (7), and looking at Fig S4a, pitch saliency seems like it could be a potential confound in that both the single upward-sweeping complex tone and its downward-sweeping counterpart may generate pitch percepts that are more salient than the combined upwards- and downwards-sweeping tones.

There are other auditory features – spatial localization, in particular - for which there is good evidence for opponency codes. This work should be cited. For example:

- Stecker et al. 2005 PLoS Biology
- Magezi and Krumbholz 2010 J Neurophys
- Ortiz-Rios et al. 2017 Neuron
- Day and Delgutte 2013 J Neurosci
- Dery et al. 2015 Cerebral Cortex
- Werner-Reiss and Groh 2008 J Neurosci
- Brand et al. 2002 Nature
- Briley et al. 2013 J Assoc Res Oto
- Stecker et al. 2015 Neuroimage
- McLaughlin et al. 2016 J Assoc Res Oto

Version 1:

Decision Letter:

1st July 2025

Dear Prof Clark,

Thank you once again for your revised manuscript, entitled "Humans can use positive and negative spectrotemporal correlations to detect rising and falling pitch," and for your patience during the re-review process.

Your manuscript has now been evaluated by the same reviewers who evaluated your original manuscript. All reviewer feedback is included at the end of this letter. Although the reviewers found your manuscript to have improved during revision, Reviewer #2 notes that the heuristic model cannot be completely ruled out and Reviewer #3 maintains some important concerns about the strength of the evidence. We remain interested in the possibility of publishing your study in Nature Human Behaviour, but would like to consider your response to these outstanding concerns in the form of a revised manuscript before we make a decision on publication.

In sum, we invite you to revise your manuscript taking into account all reviewer and editor comments. We are committed to providing a fair and constructive peer-review process. Do not hesitate to contact us if there are specific requests from the reviewers that you believe are technically impossible or unlikely to yield a meaningful outcome.

We hope to receive your revised manuscript within 4-8 weeks. I would be grateful if you could contact us as soon as possible if you foresee difficulties with meeting this target resubmission date.

- Include a "Response to the editors and reviewers" document detailing, point-by-point, how you addressed each editor and referee comment. If no action was taken to address a point, you must provide a compelling argument. This response will be used by the editors and reviewers to evaluate your revision.
- Highlight all changes made to your manuscript or provide us with a version that tracks changes.

Link Redacted

We look forward to seeing the revised manuscript and thank you for the opportunity to review your work. Please do not hesitate to contact me if you have any questions or would like to discuss these revisions further.

Sincerely,

[Redacted Signature]

Nature Human Behaviour

Reviewer expertise:

Reviewer #1: Auditory processing, neuroimaging, modeling

Reviewer #2: Auditory processing, neuroimaging, modeling

Reviewer #3: Auditory processing, neuroimaging, modeling

REVIEWER COMMENTS:

Reviewer #1 (Remarks to the Author):

Thank you for thoroughly addressing my comments and engaging more thoroughly with the existing pitch perception literature.

Reviewer #2 (Remarks to the Author):

The authors have extensively engaged with all of my previous comments, and I would first like to acknowledge the amount of work put in the revision in general and in the rebuttal in particular. The detailed responses to each point were very interesting to read, and led to further insights for me. For instance, I did enjoy how you highlighted generic algorithmic principles. So, thank you for this asynchronous but fruitful scientific exchange!

Overall, I am now reassured that the results cannot be easily accounted for by existing auditory models, which was my main concern. Moreover, I understand better how they could in fact contribute to such models - a point which is well made in the Discussion. There would be still a few things I'd be curious to clarify, but this is quite subjective and I don't feel it would contribute constructively to the review process, so I will keep this new review very brief.

The first thing that may deserve minor adjustments is what is concluded from the new figure panels S2c,d and S3a, b (I found these new simulations very interesting, thanks). If I understand correctly, the heuristic model always predicts correctly the subjective direction of pitch shift. However, the size of the "signal" from the heuristic model is much smaller for negative than positive correlations. I agree that this observation strengthens your alternative model based on spectrotemporal correlations, from a parsimony argument, but I don't think it rules out completely the heuristic model: one could imagine all kinds of transforms/thresholding to map the heuristic model's output to perception. So, maybe soften the claim that this rules out completely the heuristic model?

The second thing concerns the acoustic analyses. In the rebuttal, you mention that "However, the correlations we observe will exist

even with a single pure tone moving to higher or lower frequency." I completely agree. But doesn't this go in the direction of my argument, which was that the main conclusion from the acoustic analysis may have been drawn from all sorts of stimuli, artificial or natural? Speech sounds, at least the voiced bits, can be thought of as a coherent bunch of pure tones (harmonic as it turns out, but you are right, it is only the coherence part that matters). Isn't this fact alone enough to predict the outcome, for English, Mandarin, or any language that has a fair proportion of voiced sounds? All this to say that I agree that it was useful to check that the prediction holds for natural sounds, but I still believe that the outcome may have been predicted from first principles. If the authors disagree, it could be useful to spell out what alternative outcomes could have been predicted. Otherwise, maybe mention the generic interpretation somewhere?

Again, congratulations for an original, impressive, and thought-provoking piece of work.

Reviewer #3 (Remarks to the Author):

The authors did a good job responding to all my earlier comments. However, I still have substantial concerns.

1. I wouldn't interpret the fMRI results to be as robustly in favor of pitch direction opponency as the authors do. First, it's only 5 subjects. Second, not all subjects show significant voxels in the *fdr*-corrected maps. Third, the mask used is atypical insofar as it isn't a concatenation of predefined anatomical regions based on an atlas. I could be more convinced if the authors would also show an unmasked version of Fig 4H and 4I, that is, the same cluster-based correction method without the mask that is now included in Fig S4.

Smaller points:

- Yes, please include the whole-brain FDR-corrected single-subject contrast maps in the supplement.
- In Figs 1-3, please make more explicit the frame duration being used (e.g. 166.67 versus 40 ms).
- The rationale for using different stimuli (and not the correlated stimuli as used in the psychophysics) should be included in the Results section (the first time the new stimuli/expt are mentioned) in addition to the Methods section.
- The idea for opponency coding in the auditory domain should also be briefly mentioned in the motivation for the fMRI experiment.

Version 2:

Decision Letter:

Our ref: NATHUMBEHAV-24114772B

19th September 2025

Dear Dr Clark,

Thank you for submitting your revised manuscript "Humans can use positive and negative spectrotemporal correlations to detect rising and falling pitch" (NATHUMBEHAV-24114772B). It has now been seen by two of the original referees and their comments are below. As you can see, the reviewers find that the paper has improved in revision. We will therefore be happy in principle to publish it in *Nature Human Behaviour*, pending revisions to satisfy the referees' final requests and to comply with our editorial and formatting guidelines.

We are now performing detailed checks on your paper and will send you a checklist detailing our editorial and formatting requirements within two weeks. Please do not upload the final materials and make any revisions until you receive this additional information from us.

Sincerely,

Nature Human Behaviour

Reviewer #2 (Remarks to the Author):

The new revision has addressed the remaining minor qualifications I suggested in the last round. I cannot comment on the fMRI concerns of reviewer 3, but at least for my part, I am happy to congratulate again the authors for their work.

Reviewer #3 (Remarks to the Author):

I thank the authors for their substantial additional effort re-analyzing their fMRI data. While I myself am still not convinced by those results, I leave it to the authors and the editors to determine whether or not they should be included in the manuscript. If the fMRI data remain, I would strongly suggest including unmasked (with regard to ROI) versions of all surface activity plots as supplements to their masked surface activity plots. This will ultimately allow the reader to gauge how strongly they think the fMRI data support the authors main claim of opponency coding for pitch motion.

Version 3:

Decision Letter:

Dear Prof Clark,

We are pleased to inform you that your Article "Humans can use positive and negative spectrotemporal correlations to detect rising and falling pitch", has now been accepted for publication in Nature Human Behaviour.

Authors may need to take specific actions to achieve compliance with funder and institutional open access mandates. If your research is supported by a funder that requires immediate open access (e.g. according to [Plan S principles](https://www.springernature.com/gp/open-science/plan-s-compliance) or the [NIH public access policy](https://www.springernature.com/gp/open-science/us-federal-agency-compliance)) then you should select the gold OA route, and we will direct you to the compliant route where possible. Because authors warrant under our subscription licensing terms that they haven't committed to licensing any version of their article under a licence inconsistent with the terms of our agreement – including the applicable embargo period – publication under the subscription model isn't suitable for authors whose funders require no embargo.

With best regards,

P.S. Click on the following link if you would like to recommend Nature Human Behaviour to your librarian
<http://www.nature.com/subscriptions/recommend.html#forms>

** Visit the Springer Nature Editorial and Publishing website at http://editorial-jobs.springernature.com?utm_source=ejp_NHumB_email&utm_medium=ejp_NHumB_email&utm_campaign=ejp_NHumB for more information about our career opportunities. If you have any questions please click [here](mailto:editorial.publishing.jobs@springernature.com).

Dear reviewers,

Thank you for taking the time to evaluate and comment on our manuscript. Below, we have addressed all major and minor comments, as noted in blue. Based on these comments, we have made the following key changes to the manuscript:

- 1) Throughout we have addressed holes in our original survey of related literature, in both the introduction and in the discussion.
- 2) We have added two new sets of simulations to the paper to clarify our results:
 - a. We added simulations to show that a simple motion energy style model accounts for the phenomenology we see. This in part demonstrates how we believe existing models of auditory processing can accommodate our results.
 - b. We have added simulations showing that tracking pairs of high-intensity tones offset in time and frequency *cannot* account for our results, especially in Figures 2 and 3.

We think that these changes and the other smaller changes we made in response to the reviewer comments have clarified the ideas and presentation of data in manuscript. We hope you agree and find it suitable for publication.

Best regards,
Parisa Vaziri, Damon Clark, & Sam McDougle

Reviewer #1 (Remarks to the Author):

This paper explores the cues that humans use to perceive pitch changes. The study uses psychophysics, computational modeling, functional neuroimaging, and speech analysis to investigate if humans detect pitch motion similarly to visual motion. The authors find that humans can judge pitch direction based on positive or negative spectrotemporal intensity correlations. This ability is analogous to the visual "reverse-phi" motion illusion, suggesting a new auditory illusion. fMRI results support the hypothesis that auditory processing involves pitch direction opponency, and an analysis of English and Mandarin speech shows that pitch direction is signaled by both types of correlations, indicating ecological benefits. Overall, the research suggests that the central nervous system uses "motion detection" algorithms sensitive to local correlations for processing pitch across different sensory modalities.

This paper is well-written, the experiments flow well from one another, and it touches on a fundamental question in hearing that will be interesting to a broad audience. I appreciated the integration of the visual and auditory approaches.

We are glad to hear these positive assessments of the paper.

My main concern with this paper is that the authors have not fully engaged with the existing pitch literature, therefore, it is unclear how novel these results are. There is scope to connect these findings to the existing pitch literature to situate them more convincingly in the current understanding of pitch.

Thank you for pointing out these areas where we were not providing enough context or situating our results clearly. As we note in more detail below, we have now tried to do this, incorporating broader frameworks of pitch perception and more relevant citations into the manuscript.

- The authors cite several examples of spectrotemporal modulations being used for pitch processing; however, their citations are outdated. New work from Mounya Elhilali (some summarized in her 2019 chapter in Timbre) and Shihab Shamma (Shamma and Dutta, 2019, for example) should be integrated. The current paper's spectrotemporal stimuli appear to be an extension of existing work, rather than a novel application of this approach.

Thank you for pointing us towards these papers. Indeed, we agree that these papers examined questions related to spectrotemporal correlations. However, there are critical differences in the theoretical questions and stimuli, namely that they did not employ stimuli with the tightly controlled correlations we used to study pitch movement perception. We now cite these relevant papers in the introduction as prior work that also examines spectrotemporal correlations (Elhilali 2019 and Choi 2018), in the context of spectrotemporal power for investigating timbre and music perception. Thank you as well for pointing us to Shamma and Dutta 2019, which is specifically interested in how spectrotemporal filtering (not oriented in frequency-time) could be applied to unresolved harmonics to estimate F0. Based on this, we looked at other work from Shamma and found a different paper from that group (Chi, Ru,

and Shamma 2005) that deals specifically with oriented spectrotemporal receptive fields and directionality of signals, and it is a perfect citation to situate our work better in the field.

- Older work from Loeb and colleagues also shows how spatial cross-correlation on the cochlea could account for pitch processing – this could conceivably explain the current results (Loeb, G. E., White, M. W., & Merzenich, M. M. (1983). Spatial cross-correlation: A proposed mechanism for acoustic pitch perception. *Biological cybernetics*).

Thank you for pointing us to this paper. Our reading of it is that it presents a model for spectral cross-correlations – assessing pitch by looking at correlations in power between different frequencies, in some ways similar to more recent theories of matched filters for assessing pitch of harmonic ladders. Since it does not examine temporal correlation *per se*, it doesn't appear able to explain the changes in pitch we focus on in this study. In the revised manuscript, we have included this citation in the introductory background covering how spectral and spectrotemporal correlation detection processes have been proposed to support audition.

- Likewise, recent work from Laurel Carney demonstrating that neural fluctuation contrast may help in the decoding of complex sound is germane.

We believe the reviewer is referring to the paper with a very similar title to this comment: Carney, *Hearing Research* 2024, “Neural Fluctuation Contrast as a Code for Complex Sounds: The Role and Control of Peripheral Nonlinearities”. This paper lays out the case that early nonlinearities in auditory circuits are not necessarily an obstacle to future decoding, but rather can make certain decoding tasks easier, especially with complex sounds; i.e., these nonlinearities can be viewed as features rather than bugs in the system. Our paper focuses on simple pairwise correlations in sound intensity over frequency and time. Extracting these correlations from the sound will certainly be affected by the sorts of nonlinearities described in this paper, and could potentially be enhanced by some of these nonlinearities, including those involved in gain control.

In the revised manuscript, we now cite this paper as an example of investigating more complex processing properties where we discuss mechanisms underlying the correlation detection process.

- It is unclear how a pitch model relating to intensity correlations connect with the existing understanding of pitch computations, which appear to be limited by the coding strategies of the periphery and influenced by the statistics of natural sounds. For example, recent work from Mark Saddler (Saddler, Gonzalez, and McDermott, “Deep neural network models reveal the interplay of peripheral coding and stimulus statistics in pitch perception”) offers a review. Again, I believe the findings in this paper align well with previous work that properties of pitch computations reflect adaptations to natural sound statistics. However, this could be better connected to existing work in the discussion.

Thank you for pointing out that we did not do enough to tie this aspect of our paper to the literature. We agree that our results mesh nicely with the idea that auditory computations like

these may be tuned to the statistics of natural sounds, in our case the structure of how sounds change over time.

In the section discussing the relationship of the correlation computations to natural auditory inputs, we now cite the Sandler et al. paper mentioned above, and other papers that relate auditory processing to natural auditory statistics, including from Lewicki and Theunissen.

- It would be worth discussing how these results relate to findings with iterated ripple noise, particularly findings that cortical responses to IRN are driven by slowly varying spectro-temporal modulations instead of pitch. Is it possible you are picking up on neural signals that are not pitch-related and instead just relate to spectro-temporal modulations? Iterated ripple noise is, admittedly, a strange type of stimulus, but so are the stimuli used in this experiment. They bear some superficial similarities and I would have appreciated a deeper engagement with that literature.

Thank you for bringing up iterated ripple noise stimuli (Yost 1996). Iterated ripple noise stimuli (IRN) are generated by beginning with a noise signal $x_0(t)$, which is a raw sound waveform. This noise signal is then multiplied by a factor, or gain, before being added back to itself at with a delay, typically 2-10 ms. This can then be iterated many times: $x_i(t) = x_0(t) + gx_{i-1}(t + d)$ (there exist other variants of this, too). When the gain g is positive, these sounds tend to elicit percepts of pitch near $1/\text{delay}$; when gain is negative, it's a more complicated story and can depend on the number of iterations.

These stimuli are somewhat similar in their creation to the stimuli we used in Figure 1, which also involved adding a stochastic signal back to itself with an offset to generate correlations. There are critical differences, however, which mean that we don't think our results can be closely linked to results using IRN: (1) IRN directly adds the waveform back to itself, while our method adds together envelopes of carrier frequencies. (2) Relatedly, delays in IRN are much shorter than the envelope delays we used, which were ~ 160 ms. (3) IRN creates a waveform with specified temporal correlations, but these will not be spectrotemporally oriented or be able to generate consistently rising or falling pitch, which our stimuli were designed to do. For these reasons, we do not think that the key aspects of our psychophysics are easily related to the interesting effects associated with the IRN stimulus. Because the construction of the stimuli is similar, in the revised manuscript, we now cite the IRN literature and draw these important distinctions more clearly in the Methods section.

Addressing the neural signals portion of the question above: For the fMRI results in the paper, we used a very different stimulus — essentially drifting gratings rising or falling in frequency (Figure 4). Since the gratings differ by either increases or decreases in frequency, or both, we simply attribute the measured responses to those features of the stimuli. Since the drifting grating stimuli are also very different from IRN, it's difficult for us to see how to relate neural opponency-based responses to shared features of the two types of stimuli.

- How do the locations where you observed your opponency effects relate to classic pitch-selective voxels in the auditory cortex? (See work of Allen and Oxenham (<https://www.jneurosci.org/content/jneuro/42/3/416.full.pdf>), or Haigner, Kanwisher &

McDermott - <https://pubmed.ncbi.nlm.nih.gov/24336712/>). Likewise, work from Claire Tang and Eddie Chang using ECoG has shown voice pitch-selective neurons that, if the voxels you identified are related to speech, might be co-located. Connections to previous fMRI work on pitch was, in general, lacking.

Thank you for this apt point. We agree that we could have situated our neuroimaging results more fully in the existing literature. Functional regions of primary and non-primary auditory cortex found in our opponency experiment, and regions found in other fMRI literature on spectral and pitch processing, broadly do overlap. Given that our significant voxels include a large region of primary auditory areas (in A1 and Lbelt), it is possible that opponency computations are performed in the same primary auditory voxels already found to be sensitive to spectral content (e.g., (Allen et al. 2022)). We also saw some putative opponent-related activity in non-primary areas both posterior and anterior to HG, which have been linked to perception of pitch (Allen et al. 2022; Norman-Haignere, Kanwisher, and McDermott 2013). We further observed significant results in some more medial non-primary regions tucked into the lateral fissure (e.g., area 52 and RI cortex) which are thought to participate in higher-order auditory processing. Generally, our results showed opponent activity medial and dorsal to the STG, showing only modest overlap with typical speech regions measured in human ECoG work (e.g., Tang, Hamilton, and Chang 2017). This could simply be because we used pure tones as our stimuli and not anything resembling speech sounds.

Given the wide range of auditory cortical regions spanned by our key contrasts, a speculative interpretation here would be that opponency is a basic feature of any auditory area – primary or non-primary – that processes relative spectral information over time. A more conservative view is that given our limited sample, and use of only pure tones, we cannot make definitive anatomical conclusions about the scope of opponent spectrotemporal computations beyond primary auditory cortex and directly neighboring areas. Either way, we think that our results are consistent with previous findings related to pitch processing in the human cortex, as the reviewer notes.

In our revision, we have fleshed out our discussion of the anatomical aspects of our fMRI study to include these citations and points.

In general, the discussions of the neural basis of pitch, and the pitch literature in general, struck me as outdated and incomplete. Hopefully the references here will help.

Thank you for the comments and references above. We have noted in response to each one how we are addressing it. Overall, by addressing these comments, we have added to the introductory, results, and discussion sections of the paper and included most of the citations above, plus some additional key citations from other reviewers and our own further reading.

Other concern:

- fMRI safe headphones are generally less linear than regular headphones, causing distortions (<https://pmc.ncbi.nlm.nih.gov/articles/PMC4803580/>), could this have influenced your results? Particularly with noise cancellation? You said that participants could not distinguish the

correlated intensity stimuli – how was this determined, as the participants had no responses? Were the other stimuli equally distinguishable for participants? Could this have influenced the results?

This is a good point. We note that previously reported distortion effects tend to accompany the presence of many harmonics in the stimulus (Norman-Haignere and McDermott 2016). Our fMRI stimuli only used octave-spaced tones, so that each frequency coexists with only a limited set of harmonics at f_0 , $2*f_0$, $4*f_0$, $8*f_0$. This should in theory limit distortion products in the headphones. More importantly, it is not clear to us how these distortions would be a confound in the directional versus opponent stimuli conditions, rather than simply a subtle source of noise in the form of additional faint frequencies. Indeed, as the presented frequencies increase or decrease over time in the stimuli, the frequency of any distortion products should simply scale with them, since the distortion products are linear combinations of the frequencies present.

As for the psychophysical stimuli, we aren't quite sure if you are referring to stimuli in our behavioral experiments or the fMRI study. We'll answer with respect to both: for trying to use the behavioral stimuli in the scanner, we ran several pilots on two of the authors (SM and PV) and found that in the scanner, with our particular set-up (headphone model, non-sparse sampling fmri protocol, etc.), the behavior was hard to replicate. That is, both of us had trouble distinguishing the pitch direction compared to outside the scanner. This is why we limited our fMRI experiment to the opponency question, which was much easier to test using much simpler stimuli. We hope to address this limitation in future studies, perhaps by using sparse sampling and thus pausing the scanner noise during stimulus presentation. Second, in reference to the fMRI stimuli, these were quite easy to distinguish behaviorally as up, down, or neither (the opponent condition); thus, it is not clear how the explicit perception of direction would influence the fMRI results given that people were almost certainly at ceiling for consciously distinguishing all three stimulus conditions.

Smaller comments:

- What are the lines on the bar graphs? There are not enough to be individual participants? If this was signaled, I missed it, so it should be more clearly marked.

Thank you for this question. Yes, the lines are individual participants. Sometimes dots and lines overlapped, which is why it might look like there are fewer dots/lines than the number of subjects we reported. When there is a larger spread in the data, you can see the individual subjects much more easily. We have double checked our code and figures and confirmed that the figures all correctly represent the data.

- What were the examples that participants were shown before the experiment? The stimuli do not really sound like 'tones', so asking the participants to identify whether they sound like they are rising or falling tones seems like odd instructions. (The proof that the instructions made sense is demonstrated in the interpretable pattern of results), but I am wondering why the authors chose this set of instructions.

For each participant in experiments in Figs 1-3, we played several examples of the sounds they were going to hear in the experiment and asked them to tell the experimenter if they thought the tone was rising or falling. They were not given feedback about whether the answer was “correct”. This procedure allowed them to get used to the admittedly odd-sounding stimuli before officially reporting their judgments in the task.

Here is the approximate script for what the experimenter said to each participant:

“In this experiment, your task is to decide whether a tone is rising or falling in pitch over time. Press the ‘up’ arrow if you think that the tone is rising and press the ‘down’ arrow if you think that the tone is falling. Please do not react too quickly or too slowly. You should wait for the tone to finish before you make your decision, but you should not take more than a couple of seconds after the conclusion of the tone to make your decision. You will receive a warning on the screen if your reaction times are too fast or slow. I will now play a couple of example tones, so that you can practice deciding whether the tone is rising or falling. Please adjust the headphone volume, so that it is at a comfortable level, and make sure that you can hear the tones clearly. After we listen to a few example tones, let me know when you feel ready to begin the task.”

We have now added these details of the psychophysics procedure to the methods section of the paper.

- For all experiments, were the participants musicians? Were there differences between musicians and non-musicians?

This is a really interesting question and one we looked into somewhat. Indeed, for each participant, we asked how much musical training they had had (in years) and whether they were a native speaker of a tonal language. Roughly half of the participants had had substantial musical training and roughly half spoke tonal languages natively — and these two attributes were themselves correlated, so that tonal language speakers were more likely to have musical training. We did not see any glaring distinctions between musicians vs. non-musicians or tonal vs. non-tonal language speakers. In the triplet correlation experiment (Fig. S3), there was some hint that tonal language speakers might have been more consistent in their judgments. However, our study was not designed to test these hypotheses and was under-powered for making these sorts of comparisons between sub-populations. We do think this is an interesting direction to pursue in the future, and we have looked into Mechanical Turk-style crowd-sourced experiments in which we could get large numbers of participants with varying degrees of musical training or spoken language attributes.

Reviewer #2 (Remarks to the Author):

The manuscript describes a novel kind of auditory stimulus designed to introduce local “spectro-temporal correlations” within a sound’s spectra. The idea is directly inspired by visual stimuli that elicit spatial motion percepts, which the aim of uncovering similar functional mechanisms in the two modalities. In a series of psychophysical experiments, it is shown that listeners report upward- or downward- pitch changes associated with the nature and strength of the spectro-temporal correlations. A computational model suggests that frequency motion direction opponency can account for the behavioral results. This conclusion is further strengthened by an fMRI study, with a different set of stimuli (upward frequency sweeps, downward sweeps, and their combination). Finally, acoustic analyses of speech databases show that frequency changes induce spectro-temporal correlations, which is taken as an argument for an ecological role of such correlations.

I found the study fascinating and thought-provoking. The translation of a visual phenomenon to audition can certainly be illuminating and has the potential to highlight deep parallels across sensory modalities. The diversity of the evidence provided in the manuscript is also impressive, with psychophysical results complemented by a computational model, brain imaging data, and acoustic analyses of natural sounds.

We are pleased to read this positive evaluation of the work.

Now, for the issues I’d like to see addressed: I would like to stress that I am writing this review with the perspective of an auditory scientist. As such, I must admit it took me a few re-reads to try and recast the features of the stimuli in a language that I am more familiar with. This included putting the sound examples from the Supplemental movies (very helpful, thanks) through various kinds of auditory models, including the STRFs cited in the manuscript. Which led to the main point I’d like to raise: can the behavioral and neural results be explained by existing models of auditory processing, or do they absolutely require the novel framework developed in the manuscript?

I will detail this general question below, so the authors have a chance to refute all specific points if possible. My perspective is really that I’d like to be convinced that the novel framework is warranted, but I’d like to be reassured that it is not a rephrasing of well-established auditory models. Or if it is, that this is made clear to highlight the other contributions of this work.

Specific comments

- My main question is whether the stimulus design procedure used here introduces local frequency-shifts between spectral components, which could then be used to predict the behavioral results. An option to answer this question would be to try to fit the results with existing auditory models. For instance, cochleograms could be computed and provided to complement the schematic illustrations of the stimuli; then local frequency shifts could be extracted; then the amount of ‘up’ versus ‘down’ shifts compared. This is more or less Allik et al. 1989’s original suggestion, and also the idea behind the frequency-shift detector model (both appropriately cited in the manuscript). The extraction of frequency shifts could be achieved by

STRF models, as also cited in the manuscript. I realize that this suggestion is a big ask, and that there is no strong incentive for the authors to find an alternative, less exciting model to account for their results. But I think it would really clarify the novelty of the findings.

Thank you for this crucial question. Although we did not say so explicitly in the initial submission, we had already adopted the reviewer's proposed model! In particular, we believe this is essentially the basic model we used to think about and simulate opponency in Figure 4b and 4c. The cochlea fundamentally splits sounds into power signals arrayed by frequency over time, which is the basic input to our models in Figure 4 and our analysis in Figure 5. (In Figure 5, we used a spectrogram analysis, which has fixed bandwidth at each frequency in contrast to a more realistic cochleagram, but the basic analyses we performed examining patterns in the sounds should be robust to such filtering differences.) Then in Figure 4, we used a simple STRF linear weighting before a nonlinearity to generate signals that we could analyze for different symmetries. We think there are a few important points to make here.

First, we were unable to find in the literature any proposal for STRFs being fundamental to frequency shift detection. Instead, we found in the literature discussion of binding (i.e., into an auditory object) (Demany and Ramos 2005; Demany and Semal 2018), and other problems associated with tracking individual frequencies, which are obviated by an array of STRFs that are sensitive to appropriate correlations. There was also discussion of temporal interactions between tones, for instance a chord's components beginning synchronously or asynchronously, or preceding or succeeding the single test tone (Demany and Semal 2018), but these discussions did not specify a low-level method for determining direction. If we were mistaken and this exact proposal exists in the literature, we will gladly add in that reference to the discussion of this point. One key additional point here, though, is that we don't think that Allik's explanation for their results is compatible with these STRF models—in that paper, the authors go to some lengths to devise an explanation that is only sensitive to high-high intensity pairs and not to negative correlations per se, while we think our Figures 2 and 3 clearly demonstrate substantial sensitivity to negative correlations in the psychophysics. We show in **Figure R1** below that the simple opponent motion energy model in Figure 4c can account qualitatively for many of the psychophysical results in our paper, including the negative correlation sensitivity in Figures 2 and 3.

Second, we agree that our data do not require a fundamental rethinking of how auditory features are extracted from inputs to the ear. We view this as a positive feature of these results—there is no need to postulate new or undiscovered machinery to account for our observations. Instead, what is new in our paper is the precise description of what auditory *features* the system is sensitive to. Here, our stimuli show for the first time that the auditory system is sensitive to local spectrotemporal correlations and uses them to help discern the direction of pitch motion. We further show that *both positive and negative correlations* are used in these percepts, and moreover that both positive and negative correlation types provide information about pitch direction in sound (Figure 5).

Third, the detection of visual motion is also sensitive to correlations, and those correlations are also thought to be detected using oriented STRFs, just with the 'S' being 'spatio' rather than 'spectral'. Thus, the linear filtering proposed in the vision literature lines up very nicely

with the filtering explanations proposed in the auditory literature. What is being added here, we believe, is the emphasis on the correlational structures in sounds that these STRFs amplify, with the most ‘surprising’ result being sensitivity to the *negative* ones.

This distinction can be thought of in part in terms of the algorithmic vs. the implementational levels, in the Marrian sense (Marr and Poggio 1976). That is, correlations are theoretically and phenomenologically fundamental to visual motion detection, but have not been much considered in putative algorithms of auditory detection of pitch direction. Here, we show that an auditory algorithm in the human brain uses spectrotemporal correlation features to detect pitch motion. If they are indeed detected by units with frequency-time oriented STRFs, as the reviewer (and we) suggest, then that is how this algorithm is *implemented*. That STRFs are capable of this and have been shown to underlie directional tuning in bats makes the sensitivities measured here more compelling, we think. We believe it is largely new in this field to think about these STRFs in terms of their sensitivity to specific spectrotemporal correlations (once a nonlinearity is added after convolution). (We also note that some of the bat literature does use systems identification techniques that are quite similar to measuring responses to correlations (Andoni and Pollak 2011).)

Overall, this comment led us to realize that (1) we had not clearly stated that this linear-nonlinear model was a potential model for these effects in the discussion and (2) that we had not emphasized enough how thinking about correlations leads to a different way of viewing the processing by units with oriented frequency-time STRFs. In the revised manuscript, we have clarified both of these points in the results and discussion. We also present **Figure R1** below as part of Figure S4, emphasizing that a simple model can indeed account for the correlational sensitivity we measure as we investigate the algorithms for pitch motion detection.

Figure R1. A simple STRF followed by a quadratic nonlinearity accounts for qualitative psychophysical results in our paper. Here, we plot mean responses to all our stimuli for the opponent motion energy model in Figure 4c, a simple model with an oriented frequency-time filter and a quadratic nonlinearity. (a) When the model is presented with the ternary stimuli in Figure 1, it responds positively (negatively) to the positive (negative) correlation stimuli. (b) When the model is presented with the stimuli in Figure 2, it responds positively (negatively) to the positive (negative)

correlation stimuli. (c) When the model is presented with the stimuli in Figure 3, it responds positively to both the high-high and low-low positive correlation stimuli, and negatively to the high-low and low-high negative correlation stimuli.

One last point is that though we believe this type of STRF model can explain some previous frequency shift detection (FSD) results, there are some FSD results in which changes are detected after a silent pause of 0.5 seconds or longer (Demany, Pressnitzer, and Semal 2009). It's harder to reconcile this kind of data with the timescales we observed for correlation detection in this paper, which operate on a timescale almost an order of magnitude shorter. It therefore seems likely that there are multiple mechanisms at play (as would indeed be similar to vision (Lu and Sperling 1995)).

- L21. "Importantly, work on such spectral pattern cues typically employs sounds with persistent groupings of tones". It is not clear to me what is meant by persistent, here and elsewhere (the word is certainly not common in the auditory jargon). How long would qualify as persistent? In the work cited in the manuscript (Demany and Ramos 2005 plus follow-ups), there was a single short random chord followed by a single short tone. How is this different in terms of 'persistence' to the frame-by-frame correlation introduced here, at least for the positive correlation cases?

This is a good point, and it led us to think more carefully about our phrasing, especially whether 'persistence' of stimuli was a useful distinction between our stimuli and prior stimuli. Upon reflection, we believe this is not a useful distinction for two reasons. First, as the reviewer correctly points out, though our stimuli have shorter-range correlations than some prior stimuli, they are not notably shorter than many prior stimuli used to study frequency shift detectors. Second, short duration does not in principle preclude streaming or auditory object tracking, as this reviewer and others mention in other comments.

To address this critique in the revised manuscript, we have removed all references to 'persistence' as a distinction between our stimulus and prior ones. We now emphasize, as the reviewer hints above, that the negative correlation stimuli are the critical stimuli for assessing correlation sensitivity compared to potential short timescale tracking. In the case of negative correlations, it is parsimonious to conclude correlation sensitivity and otherwise requires awkward contortions to produce a tracking algorithm that can follow, for instance, pips from high intensity to low intensity, assess their direction, and then recognize that a sign change has occurred and invert the percept, compared to pips that don't change their intensity.

- L67. "stimuli that use increments and decrements in intensity to generate local correlations in intensity at specific offsets in frequency and time (see Methods)." This could be expanded, as unpacking the meaning of the sentence is critical to appreciate the rest of the manuscript. For instance, it could help to further specify which vector/2D matrix is correlated with what, what are the numerical ranges of correlations obtained with the stimuli used in the experiment, etc...

Thank you for this comment. In the revised manuscript, we have expanded this statement at this point of the introduction to better unpack its meaning and prepare readers for what is to come.

- L93. “We emphasize that in these negative correlation stimuli, there do not exist spectral patterns that persist between frames.”. This is, I believe, the critical claim for the whole manuscript. But I am not sure I understand why this is a priori obvious. The schematics include ‘white spots’ suggesting absence of sound, but wouldn’t the relevant correlations for perception be between the remaining ‘black spots’ or ‘gray spots’? Could it be that a side effect of the negative correlation design is to introduce local frequency-shifts between the remaining spectral components, in the opposite direction and consistent with the subjective judgments? Eyeballing the cochleograms of the stimuli and computing a few STRFs suggested that it is not out of the question. Eyeballing is no strong argument (thankfully!) but a proper quantification of such putative shifts would help tremendously to strengthen your point, I think.

Thank you for this important question. We believe there are several responses to provide here. First, it is certainly mathematically true that there exist no positive pairwise correlations in the upward direction when negative correlations are enforced downward — we show that analytically in the methods section and plot the cross-correlogram in Figure S1a. The only pairwise spectrotemporal correlations that exist in these stimuli are the ones we impose.

Second, the reviewer is absolutely correct that if we impose, for instance, a negative downward correlation, there become fewer positive downward correlations. This was essentially the argument in Allick et al. (1989). If one were only capable of sensing pairwise correlations (i.e., pure products of pairs of intensities, as computed in the mathematical argument above), then fewer positive correlations is identical to more negative correlations, and this becomes a distinction without a difference. However, it is possible that the auditory system could count solely the pairs of high intensity tones at a certain time and frequency offset, and nothing else. (Note that this selective computation itself would entail computing higher order correlations than just pairwise correlation detection.) In this case of sensitivity only to high-high pairs of intensity, with the stimulus in Figure 1, imposed negative downward correlations could in theory be detected by counting the net high-high intensity pairs in the up vs. down directions, in keeping with the reviewer’s logic (and the logic of Allick’s explanation).

However, the experiments in Figures 2 and 3 argue powerfully against this proposed explanation for negative correlation detection. In these stimuli, the tone differences from baseline are sparse in time (in contrast to the dense correlations in the correlated ternary noise stimulus used in Figure 1). This means that imposing a negative correlation in the downward orientation does still reduce the number of downward directed positive correlations, but by relatively very little as a fraction of all positive correlations.

To gain a better understanding of the size of these different effects, we have run simulations on the stimuli used in Figures 1, 2, and 3, and we show the results below (**Figure R2**). In these simulations, we counted the net number of up- vs. down-directed high-high, low-low, high-low, and low-high intensity pairs, similarly to the analysis in Figure 5. For the Figure 1 stimuli, we found that—just as the reviewer observed and predicted—counting only the net high-high intensity pairs could be used, for instance, to obtain an upward signal from downward directed negative correlation stimuli (**Figure R2a**).

Figure R2. Counting different high and low intensity patterns in stimuli used in Figures 1-3. The y-axes show the fraction of time steps during which a particular pattern appears. The patterns are high-high (HH), low-low (LL), high-low (HL) and low-high (LH) intensity (as also in Figure 5 of the paper). The pattern can either be in the same frequency-time orientation as the imposed correlation (“with”) or in the opposite orientation (“against”). The difference between with and against counts would be a net count for that pattern type, shown in the lower axes in each panel. The temporal offset for each pattern was 40 ms, which matched the pip offset for the stimuli used in the Figure 2 and Figure 3 stimulus simulations. (a) Figure 1 stimuli have the same counts of all patterns in the “against” orientation, since that orientation has random patterns. The positive correlation stimulus has an excess of ‘with’ HH and LL counts and a dearth of HL and LH ‘with’ counts, signaling that either positive correlation pairs (HH and LL) or negative correlation pairs (HL and LH) could be used to determine the stimulus direction. Importantly, the opposite is true in the negative correlation stimulus, so that HH and LL net counts can predict the percept, as the reviewer suggested. (Allick’s model was that the HH is all that is used.) (b) The same counts for the positive correlated pip stimuli in Figure 2 show that the net counts for the HH and LL patterns are roughly a factor of 10 larger than the net counts for the HL and LH patterns. When negative correlations are shown, the LH and HL patterns show net counts about ten times larger than the HH and LL patterns. This suggests that it would be very difficult to ascertain direction of the negative correlation stimulus only by examining the positive pairs (HH and LL). (c) A similar pattern holds for specific counts for the stimulus in Figure 3, in which the net counts of the imposed pattern show the largest values, roughly ten times larger than for other patterns. The results shown in panels b and c thus do not comport with the psychophysics. In the revised manuscript, the panels from Figure R2 have been distributed into Figures S2 and S3.

For the Figure 2 stimuli, it was no longer easy to use the high-high counts to discriminate direction of the negative correlation stimuli (**Figure R2b**). In correlated pip stimuli in Figure 2, when negative correlations were presented, counting high-high pairs gave a signal aligned with the perceived directional tuning, but with a signal that was a factor of ~ 8 times smaller than the negative correlation net counts and a factor of ~ 8 times smaller than the high-high counts for the positive correlation version of the stimulus. In the psychophysical experiments (Figure 2), the rising and falling categorization performance was about equal for the positive and negative correlation stimuli, which is not consistent with an 8-fold change in signal. (And our coherence experiments in Figure 1 lead us to believe we are not saturated in the psychophysics.)

For the Figure 3 experiments, the simulation results were also striking (**Figure R2c**): There, when high-low correlations are imposed, the net signal in the high-low pairs was about 10 times larger than the net signal in the high-high pairs. Again, in the Figure 3 psychophysics, we found classification performance that is consistent with approximately equal salience of positive and negative correlations, and therefore inconsistent with a factor of 10 difference in signal this type of counting model would predict. To make this fact clearer, we have computed the net directional discrimination between the 4 different pip patterns (as shown in Figure R3 below). The positive and negative patterns are about the same distance from 0, indicating similar perceptions for the stimuli. We have also added this as a panel to Figure 3 in the main paper.

Figure R3. Discrimination between upward and downward directed pip patterns from Figure 3 in the main paper. Here, we plot $P(\text{rising} \mid \text{upward directed}) - P(\text{rising} \mid \text{downward directed})$ to obtain a ‘net probability’ distinguishing the upward and downward directed pip patterns.

Overall, if we made the correlated pips even more sparse, we could strengthen further the difference between these different kinds of pip-pairs in these stimuli. We believe that the experiments in Figures 2 and 3 thus rule out the possibility that the relevant auditory algorithms are effectively counting high-high pairs. A more parsimonious explanation is that the auditory system is detecting spectrotemporal correlations for both positive and negative correlation signs. To make this clear to readers, in the revised manuscript, we have now added the panels of Figure R2 to Figures S2 and S3, added long-ish explanations in the figure captions and methods, and summarized these findings in the main text of the results.

Finally, it's also interesting to note that it is not simple to make a biological version of the high-high intensity detector, as we used for this modeling exercise. For smooth, biologically implemented mechanisms, response as a function of the stimulus can be expanded in a Volterra series, which has zeroth, first, second, and higher order terms (Clark and Fitzgerald 2024). The second order term will be sensitive to the correlation in the input, positive or negative, and many higher order terms must be added to it in order to generate a mechanism that responds, for instance, to high-high but not low-high or low-low.

- The binaural experiment is both elegant and interesting. I understand the results as showing that both the FSD/correlation detectors and the putative opponency operation must be post binaural convergence. Would you agree? In this case, it is still possible that each happen at different stages of processing. For instance, there is plenty of neural evidence for FM detectors subcortically (in the inferior colliculus for instance), but as far as I know, no evidence for opponency. Would a two-stage model still be consistent with your findings?

Yes, we agree with this interpretation. It is quite consistent with the binaural data to: (1) combine signals across ears, (2) compute correlations rising vs. falling, and (3) subtract them afterward. Steps (2) and (3) could be combined to an extent, as they appear to be in fly visual motion detectors (Badwan et al. 2019), as we discuss, or could happen sequentially, analogously to V1/MT in models of visual motion detection (Heeger et al. 1999).

- The next experiment, involving tone pips, is used to measure tuning characteristics. I am not sure I understand why it was necessary to change the stimulus design procedure for that purpose. Couldn't you adjust the parameters of the stimuli of Experiment 1?

This a very good question. In prior work in vision using the stimuli in Experiment 1 (Salazar-Gatzimas et al. 2016), we used the reviewer's proposed method: we just decreased the frame duration and then created correlations between every n th frame in order to generate a fine-grained tuning curve with correlation intervals equal to $n*dt$ with $dt = 1/180$ s. That worked to get us ~6ms resolution in the fly. However, when we tried moving to faster frame rates in the auditory experiments, we stopped being able to perceive any rising or falling nature of the sounds, regardless of the interval of the imposed correlations. We thought initially that this might mean we could only hear strong correlations at the 150 ms timescale, but when we devised the correlated pip stimulus, we realized that it must be some other aspect of that fast stimulus that interferes with perception. We switched to the correlated pips because they were sparse and allowed us to continuously vary both the pip duration and the interval. As noted above, they also are more convincing for showing that percepts are related to correlation detection rather than some kind of tracking heuristic. To address this question, in the revised manuscript, we make these advantages of the correlated pip stimulus more explicit.

- L207: The sensitivity for the correlation-detection mechanism peaks at very similar parameters than what was measured for FSDs in the literature. Isn't this a tantalizing hint that both mechanisms could in fact be one and the same thing?

We agree that the frequency different tuning we observe here is similar to that measured in (Demany, Pressnitzer, and Semal 2009). As we suggested above, we believe that some FSD

psychophysics could be explained by local correlation detection. There are also other tuning properties in that paper that the correlation sensitivity we measure cannot explain. In particular, people are able to detect shifts with silent intervals of up to 900 ms—this timescale is much longer than a simple model based on our results would predict. Thus, we believe that our results may suggest a mechanism for some aspects of FSD, which could rely on several parallel mechanisms.

To address this question, in the revised manuscript, we point out this similarity in the tuning explicitly near Figure 2 and again in the Discussion. We also emphasize in the revised manuscript, as we have here and above, that the correlation sensitivity we observed could explain aspects of FSD, and that the sensitivity itself, to both positive and negative correlations, is the novel result.

- L234. The triplet correlation experiment, again inspired by visual studies, is very succinctly described. I am not sure I understand what it brings to the core argument.

These results demonstrate that human rising and falling pitch percepts are sensitive to triplet correlations, in addition to the pairwise ones that make up the bulk of our study. This is visible not just in the non-random percepts, but also in the inversion of the percepts on inversion of the correlation — this is the third-order analog of the reverse-phi percept with pairwise correlations. Moreover, the experiments demonstrate a common pattern of responses between vision in several animals and in audition, and thus suggest similar algorithmic processing. Because these results do not fit easily into the main flow of the pairwise correlation story, but because we still consider them interesting, we placed them into a supplemental figure. In the revised manuscript, we have expanded our succinct description, especially in the figure caption of the supplemental figure but also in the main text, where we offer more details about the stimulus, analysis, and interpretation of the results. If the reviewers believe this should be excised in favor of clarity, we would do that, though it is not our first choice.

- L265. “Crucially, however, in this model, negatively correlated stimuli exhibit a different tuning from oppositely directed positive stimuli; [...]”. I agree this is crucial, but I am not sure I see it in Fig 4b, which looks quite symmetrical. It could be useful to provide the detailed predictions of a model without opponency, so the benefit of adding the second stage becomes obvious.

Thank you for this important question. The symmetry we are examining is not whether the positive and negative correlations result is symmetrical about the mean response—that is, the lines in Figure 4b are indeed symmetric when reflected about a line of $y=350$, but that is not the symmetry we are examining. Instead, we point out that in Figure 4b, negatively correlated stimuli look *different* from positively correlated stimuli *in the opposite direction*. In Figure 4b, this is equivalent to asking whether the purple curve looks like the green curve reflected about $x=0$. (It does not.) However, when we examine the opponent signals in Figure 4c, they do have this symmetry, as do the data curves in 4d and 4e. To address this issue, we have added to the text in question near former L265 to make clearer the symmetry we are discussing, and also added further text in the figure caption.

- fMRI experiment. The nice novelty of this experiment is to contrast responses of upward or downward frequency glides to responses of the same but superimposed up and down glides. While this has never been analyzed in terms of an opponency model, there has been at least one previous study using similar stimuli: Overath et al., J Neurosci, 2010. I think it could be relevant to cite, and if possible at all (given that their task was very different), compare with your results.

Thank you for pointing us to this paper. These stimuli are pleasingly similar to ours, in terms of mixing rising and falling tones. In Overath et al., the authors concentrate on the effects of coherence and changes in coherence. They find modestly smaller responses to incoherent stimuli (mixing up and down), which seems consistent with our opponent suppression. It is difficult to compare the findings however, because their incoherent stimuli seem to preserve the total number of rising and falling ‘streaks’, while our stimulus did not. Our stimulus also had far more separated tones than theirs. And the bulk of their analysis investigated how auditory cortex responds to abrupt *changes* in spectrotemporal coherence, where they saw the largest effects. We cannot make comparisons with that aspect of their analysis because we had long durations of silence between our longer blocks of rising, falling, and rising+falling stimuli. We now cite Overath et al. (2010) in our revised paper as having examined mixtures of rising and falling tones that relate to our study.

- Fig 4g, stimuli. I agree that, if one assumes an opponency model, your predictions are fully warranted. However, I wonder if the same predictions could be made without any need for an opponency mechanism. For example, current neural models of FM detection involve delayed inhibition to select for one FM direction (following the work of Suga 1965, admittedly mostly in bats). Wouldn't such a delayed inhibition mechanism also produce a reduced response for the superimposed stimulus?

Thank you for this question. We interpret this to ask whether an STRF model with delayed inhibition — a Barlow-Levick model in vision — would also produce a reduced response to the summed up and down directed tones, as in Figure 4. The short answer is that a delayed inhibition mechanism is not alone sufficient to generate opponent responses. To demonstrate that, we simulated a linear-nonlinear cascade model using an STRF with a delayed negative lobe and three distinct nonlinearities (**Figure R4**). The orientation of the STRF linear filter makes the model's preferred direction “up”, and the mean responses to upward moving gratings are larger than to downward moving gratings. However, the response of these 3 models to the combined upward + downward moving gratings is larger than to the upward gratings alone, demonstrating a lack of opponency (as we have defined it).

The longer answer is considerably more nuanced. First, in (Badwan et al. 2019), we showed that, for *any* linear-nonlinear model in which the nonlinearity is convex, it is impossible to suppress the mean response to a preferred direction drifting grating by the addition of a null direction drifting grating. That proof was for drifting sinusoids. However, as we examined this question, we have demonstrated that the result can be extended to drifting comb functions of the type we used in the auditory experiments.

The proofs for those (surprisingly general) results rely on the linearly filtered stimuli all having the same mean (up, down, and up+down). This can occur when the stimuli all have mean 0, which is a natural way to represent visual contrast stimuli, in which mean stimuli have contrast 0. It can also occur when the integral of the linear filter is 0, for instance if it senses only derivatives.

It is less obvious that auditory stimuli should be represented with mean 0. In our simulation in **Figure R4**, the stimuli did not have mean 0, and the negative lobe in the filter had twice the integral of the positive lobe, so the mean filtered stimuli were not all equal (and the mean of the added gratings was 2x the mean of the single gratings). Even though our proof does not apply in this case, we easily found cases like the one shown below (the first we tested), in which opponency does not occur. However, by varying the linear filtering (especially by employing negative lobes with much larger integrals than the positive lobes) as well as by varying the shape of the point nonlinearity, it is possible to construct cascade models that do show opponency. (In (Badwan et al. 2019), we also showed we could do this when we relaxed the convexity restriction.)

In the discussion, we had originally noted that one could generate opponent responses in single units, without subtraction. Based on this question and what we found answering it, we have now expanded that statement to say that single units could also potentially be opponent, depending on their linear and nonlinear processing.

Figure R4. Simulation of linear-nonlinear (LN) model of single auditory unit to the up- and down-moving gratings found in the Figure 4 fMRI experiments. The linear filter is shown to the left and has a large delayed negative lobe (indicated by the blue), and an almost instantaneous positive lobe (indicated by the red). After the linear filter is convolved with the stimulus, the resulting signal was sent through one of three distinct point nonlinearities, $g(\cdot)$, shown in the middle column. These nonlinearities are rectified linear units (ReLU) raised to the power of 0.5, 1, and 2. The response was then averaged over time, and responses were also averaged over all possible phases of the up and down gratings for the up+down condition. In all three cases here, the up+down condition resulted in model responses larger than the up condition alone. This is therefore an example of a model that has a negative lobe and is not opponent.

- L382. The use of an optical flow algorithm to evaluate the amount of ‘tone change’ over time is yet another example of the potential benefit of getting inspiration from visual science, very clever. Would it be interesting to apply the algorithm to the stimuli used in your experiment, and not only for the acoustic database analysis? Could this quantify the amount of local frequency-shifts in the stimuli?

This is interesting question. We applied the Horn-Schunk method to our Figure 1 stimulus, using each frame in the stimulus as a frame of a movie. We found that the flow was in the direction of imposed positive correlations but in the opposite direction of imposed negative correlations. We think this is actually expected, since the algorithm is iterative and driven by terms proportional to $(\partial I/\partial f)(\partial I/\partial t)$, where I is the intensity, f is the frequency band, and t is time. This multiplicative term is equivalent to computing a pairwise correlation (Potters and Bialek 1994), so that this algorithm is sensitive to pairwise correlations in the stimulus. (In fact, we believe this to be true of all optic flow detection algorithms that we know of — if it’s based on local intensity and minimizing squared error, then it is difficult not to be sensitive to pairwise correlations. Lucas-Kanade is another one, and the optimal algorithm in Potters & Bialek 1994 is another.) So we believe these flow algorithms can quantify local frequency shifts in our stimuli from Figure 1, but in essentially the same way that our correlation metrics do.

There are two additional points to make here. First, we don’t think this algorithm will work for stimuli in Figures 2 and 3. In those cases, the correlation is between times that can be separated by scores of milliseconds in which neither pip is present. Optic flow detection algorithms almost always work very locally in time, on just two adjacent frames, so they would not be able to detect the spectrotemporal correlations in these stimuli without some kind of modification, since there are many frames between the correlations.

Second, one might ask: is it circular to use a correlation-based method to compare with correlation patterns in the stimulus in Figure 5? We worried a bit about this, but we believe the argument is not circular for what we are trying to show. Even though the Horn-Schunk algorithm is sensitive to spectrotemporal correlations, it does not distinguish between high-high and low-low correlations for instance, or net positive high-high and net negative high-low counts (again in Figure 5). Our goal in Figures 5 and S5 is not to say that pairwise correlations are informative about motion direction, since that is well-known from the vision literature (Potters and Bialek 1994; Fitzgerald et al. 2011; Adelson and Bergen 1985). Instead, our goal in Figure 5 is to point out that both positive and negative correlations are informative about motion direction, and that, for instance, high-low pairings can also tell us about motion direction.

- The amount of ‘tone change’ is found to be correlated to the presence of spectro-temporal correlations. Is this really surprising? I wonder if it is not an inescapable consequence of having harmonic sounds, as changes in f_0 will necessarily translate to many structured changes in correlations all over the spectrum. If so, then focusing on speech databases for the analysis may be slightly misleading. The important prerequisite for the observed correlation is not that ecological sounds are used, but rather that harmonic sounds are included.

Thank you for this question. We do not believe this has to do with harmonic sounds in our database. As the reviewer says, it is true that with harmonics, there are lots of structured changes to observe at many frequencies. However, the correlations we observe will exist even with a single pure tone moving to higher or lower frequency. To see this, one may just look at the bottom of Figure 5b: the leading edge of an upward moving tone has high-high and low-low patterns, as well as low-high patterns. Meanwhile, the lagging edge contains high-high, low-low, and high-low patterns. The association of the two reverse-phi patterns with the leading and lagging edges has been pointed out before in visual stimuli (Salazar-Gatzimas et al. 2018). Once negative correlations are pointed out in natural stimuli, it is clear they are there, but our experience is that most neuroscientists think of these as being a laboratory oddity rather than a fact of natural inputs, so one point of this figure is to demonstrate the ubiquity of negative as well as positive pairwise correlations. We believe this approach of thinking about correlations has not been a large part of the auditory literature, but especially for detecting pitch motion, it should be, and it's a different framework from what we've found in the literature (and hopefully now cited more completely in our revised manuscript).

- L633. Several sound examples were given before experiment to train participants. I can imagine that the task was tricky to explain, as most pitch-shift tasks usually are. Was there any exclusion criterion? Did all recruited participants manage to pass the informal training phase? If they did, that'd be quite surprising, and worth reporting.

Thank you for this question. In response to this and other questions, we have expanded our explanation of the experimental protocol in the Methods section. Briefly, we had participants listen to the sounds and asked them to judge whether overall it was rising or falling, but did not provide feedback about their answers. We did exclude 3 participants. One reported being “tone deaf” and being unable to make any judgements whatsoever about the sounds we played (or other sounds in their life). Two additional participants were excluded after they volunteered after the experiment that they were trying to game the task and believed there was a pattern we were using in rising vs. falling tones to trick them. (!) Overall, most participants seemed to do okay on these perception tasks, which is also our impression playing these for colleagues and friends — most people discern some overall rising or falling in these sounds.

- L659. Waveforms were “scaled to have a minimum value of -1 and maximum value of $+1$.” In addition to being slightly unusual (one would usually scale rms values and not max/min amplitude), this does not sound right: such a procedure implies that all waveforms had symmetrical maximum/minimum values, or that they were scaled by introducing a non-zero DC. It is likely completely inconsequential to the results, of course, so just noted here for precision.

Thank you for pointing this out. This method described the scaling that the Matlab command ‘soundsc’ performs before playing sounds. We used this during debugging, but during the actual experiment, we do not use this command and instead used ‘PsychPortAudio’ in Psychtoolbox. We have therefore removed this text from the Methods and instead report the command used to play the sounds.

L838: the sample size for the fMRI experiment is only clearly stated here: N=5, including two of the authors. I am no fMRI expert, but it seems unusual from what I see in the literature. Is this acceptable by current standards?

This is an important point. For many traditional fMRI studies, larger samples (e.g., 20 subjects) are indeed the norm. However, for very simple psychophysics experiments with few stimuli displayed for many TRs worth of time, as in our study, smaller samples (e.g., 2-8) like ours are common (Heeger et al. 1999; Norman-Haignere et al. 2019; Heitmann et al. 2023; Pinsk et al. 2009). In essence, our study is similar to an oversampled ‘localizer’-like run used in studies of more complex perceptual processes (e.g., categorization, responses to song/music, memory effects, etc), and we believe sufficiently powered here. Most importantly, we note that we saw the observed effect in every subject (a rarity in fMRI studies of more complex tasks), and the fMRI contrasts were also significant in the same general auditory regions in each subject as well. Thus, we think the results are robust for the question our fMRI study addressed.

Reviewer #3 (Remarks to the Author):

Vaziri et al. use novel correlation stimuli in the auditory domain combined with fMRI and acoustic analysis of natural speech to make two main claims:

The first main claim is that humans can use local spectrotemporal correlations to judge whether an acoustic stimulus (“pitch” is used) is rising or falling, even if listeners are prevented from tracking spectral patterns across longer timescales (i.e. successively rising or falling local components). The use of stimuli with negative local spectrotemporal correlations and the fact that listeners perceived these stimuli in the opposing direction as their positively correlated counterparts is the main evidence used for the central claim.

The second main claim, based on fMRI experiments with upwards, downwards, and combined upwards-and-downwards spectral motion, is that spectral motion uses an opponent channel code in auditory cortex.

If true, these results would indeed be novel and interesting and potentially initiate a new line of inquiry in (human) auditory processing. However, I have several concerns that should be addressed by the authors before I can endorse the findings:

We are happy to hear this positive evaluation of the claims in the paper, and we believe we can address the concerns below.

1. It’s not clear from the data and analyses shown how different the down-negative stimuli (right-most panels of Fig 1B) are from the up-positive stimuli (left-most panels of Fig 1B). I understand how they are different theoretically, but practically speaking, the resulting down-negative stimuli appear to have several spectrotemporal sub-regions with upward spectral motion that could potentially be tracked across time as an auditory stream. Similarly, the up-negative stimuli (3rd column or Fig 1B) seem to have spectrotemporally localized downward spectral motion (like the the down-positive stimuli do). Given how critical this is to the authors’ first main claim, I think more needs to be done to convince the reader that these two pairs of classes of stimuli (pair 1: up-positive and down-negative; pair 2: down-positive and up-negative) are substantially different from each other, especially at the level of individual trials where such localized spectrotemporal motion should be characterized.

a. One suggestion would be autocorrelation types of pitch models.

b. Or maybe something here could be useful? <https://amtoolbox.org/>

Thank you for this question. We agree that this is a critical point of our paper, and is similar to Reviewer 2’s question labeled “L92” above. Our response to that question and analyses in **Figure R2** both make similar points to our response below to this question.

To begin with, the imposed correlations are necessarily averages over instantiations of the stimulus, but the average correlation exists only in the imposed direction, as we illustrate in Figure S1 and demonstrate analytically in the methods. There will be some fluctuations in correlations about this mean, but those demonstrably average to 0.

There are two possible patterns that the reviewer may be observing here. First, because eyes detect orientation at least in part by using pairwise correlations (Victor, Thengone, and Conte 2013), we perceive the same visual orientation in our two-dimensional representation of the

correlated spectrotemporal sounds as we hear in them when equivalent space-time intensity patterns are presented. So, it is natural to see a degree of upward orientation in the downward negative stimulus illustration.

The second pattern was noted by the reviewer: an imbalance in the high-high and low-low pixel pairs oriented upwards vs. downwards in, for instance, the downward negative stimulus. This occurs for the stimulus in Figure 1 because imposing negative correlations in the downward direction, for instance, means that there are necessarily fewer positive correlations in that direction. To illustrate this effect, we have counted the number of high-high intensity patterns in both the direction of the imposed correlation (“with”) and the opposite direction (“against”) for both positive and negative correlation versions of Figure 1 (results shown in **Figure R2**, above). There, we find, just as the reviewer noted, that one could examine only high-high (or low-low) intensity patterns and obtain the perceived direction of the stimulus. (Alternatively, it is equally true that one could be sensitive exclusively to high-low or low-high patterns and obtain the perceived direction of both positive and negative versions of the stimulus.)

However, these high-high and low-low “object tracking patterns” are really only usable in this way for the stimuli in Figure 1. In **Figure R2**, we show how these counts are significantly biased for the stimuli used in Figures 2 and 3. In those stimuli, the patterns matching the imposed correlation carry a signal that is 8-10 times larger than the signal carried by other patterns. For instance, when negative correlations are imposed in Figure 2, the high-high counts have a net signal about 1/8 the size of the high-low counts. A similar difference exists for the negative correlation patterns in Figure 3. Since the positive and negative correlation versions of the stimulus are found to be about equally perceptible in both Figures 2 and 3 (as measured by the fraction of percepts of rising and falling in these cases), this pattern counting heuristic is strongly disfavored compared to a correlation detector, which easily accounts for the percepts. (We have now shown how the simple correlation-sensitive model in Figure 4c accounts for the direction of percepts of stimuli in Figures 1-3 — that is shown in **Figure R1** above and is now included in Figure S4 in the paper.) Our revised manuscript also includes the figure panels in **Figure R2** in Figures S2 and S3, and includes a brief summary of the points made here.

2. I couldn't find in the manuscript whether feedback was given to listeners during the main psychoacoustic experiment. This should be reported, as it could potentially have a big impact on the behavioral findings.

Thank you for making this important point. When introducing the experiment, the experimenter presented participants with sample sounds, but did not provide any feedback about “correct” answers. In fact, the participants were wearing headphones and the wav files were unlabeled, so the experimenter could not easily have provided any unintentional feedback, either. The experiment began whenever the participants felt ready (but this would never exceed a couple of minutes of listening). Participants would also often report (during the instruction phase) that they were hearing both rising and falling throughout a single tone, so the feedback given to them was to try to make a decision about the overall pitch direction.

No feedback was given during the main task either. In the revised manuscript, we have now described this procedure in more detail so readers have this information.

3. Does the time interval scale of Fig 2C match the offset parameter mentioned in the first paragraph of the Results? That is, does 1/6 of second as mentioned in the first paragraph of the Results align with 166.67 ms on the x-axis of Fig 2C? If so, then aren't there some behavioral inconsistencies between the data shown in Figs 2 and 3? Some clarification here would be appreciated.

Thank you for this question. Figures 1 and 2 used qualitatively different stimuli. In Figure 1, each 'frame' of the sound was played for 1/6 s (=167 ms) in time, then switched to the next frame in the sound, and so forth. Thus, each frequency played was constant for 167 ms, before changing (or not changing, depending on the exact instantiation of the stimulus). In contrast, in Figure 2, we played short pips, which lasted for 50 ms each. With these, we could continuously vary the time between correlated pips at adjacent frequencies. The time scale on the x-axis of Figure 2c is this delay. The peak here occurs at 40 ms, and by 160 ms and 320 ms, there are no discernable percepts.

This result is not inconsistent with the result in Figure 1, in which we had strong responses to frames that lasted for 167 ms. This is because in those long frames in Figure 1, the signal is on for the entire duration of the frame, and thus there are correlations at many delays, ranging from almost zero ms (between the end of one frame and the beginning of the next) to 333 ms (between the beginning of one frame and the end of the next). An algorithm that integrated correlations at many delays would yield strong responses to the stimulus in Figure 1, as we saw.

To demonstrate this concretely, we have generated a simple simulation of a correlation detector, based on the same model as in Figure 4c: the opponent subtraction of two oppositely tuned motion energy models. The result of this simulation is shown in **Figure R5** below. The model we used is tuned to peak at around 40ms to the Figure 2 experiments, as shown in our main Figure 4c. Figure 4c also shows that this model has virtually no response when delays between pips are 160 ms (Figure 4c). However, when presented with the stimulus in Figure 1, it responds strongly for tone durations from around 40 ms out to 200 ms, within those bounds responding more strongly than to the correlated pip stimuli. (The units are arbitrary but matched across the two numerical experiments.) The responses to Figure 1 stimuli are stronger in part because there is much more motion energy (as defined in (Adelson and Bergen 1985)) in the stimulus than in the correlated pip stimuli in Figure 2. Normalization operations in the brain could tend to equalize such signals (Carandini and Heeger 2012). Based on the logic above and the simulations in **Figure R5**, we believe that the peak response to the pips at 40 ms is entirely consistent with strong responses to the Figure 1 stimuli with tones that last for 1/6 second.

Figure R5. Here, we plot the response of the simple model in Figure 4c to the stimulus used in Figure 1, but at a variety of frame durations. In the actual experiment, we used ‘pixels’ that lasted for ~160 ms. This plot shows that although this model has its strongest response to correlated pips with a delay of ~40 ms (see Figure 4c), the model still responds strongly to our designed chords in Figure 1 lasting about 1/6 second.

4. Related to (3), it seems that the maximum effect size happens for time intervals of ~40 ms. This is largely consistent with other lines of work showing figure popout amongst otherwise random sequences of chords (pop-out was strongest around 25 Hz or 40 ms per chord). See Teki et al. 2013 and related work from Maria Chait’s group. This again makes me wonder about auditory stream tracking across time of localized spectrotemporal motion as a possible explanation for the results.

We agree that it’s tantalizing that some of the timescales we have found here are similar to other measurements, both of frequency change detection and the auditory object detection in Teki et al. and Chait’s other work. In response to this comment and comments from other reviewers, we have inserted the Teki citation near Figure 3 and made the observation that this also matches psychophysical timescales associated with stream tracking.

We believe that the fundamental question being asked here is whether what we’re seeing is really correlation detection or just really fast tracking of auditory objects. The difference is not just semantic, since we think the two algorithms make very different predictions in the case of the *negative correlation* stimuli. In particular, we think auditory stream tracking of local spectrotemporal displacements of pips could not explain the inverted percepts when the pips change from high to low or low to high in Figures 2 and 3. To obtain such a result with a tracking algorithm, the algorithm would have to (1) bind the two nearby pips with opposite deviations from the background; (2) determine the direction of displacement of this ‘object’ that consists of one part high intensity and one part low intensity; and (3) then recognize that the pattern had a peculiar intensity structure and invert (for unknown computational reasons) the direction of the associated percept to be in the direction opposite the displacement. It is possible that a tracking algorithm does all that, but in our view the detection of pairwise correlations to detect motion seems to provide a more parsimonious explanation. And the specific observed sensitivity to pairwise correlations seems natural when one considers the structure of pitch motion (Figure 5).

We want to emphasize here, and we emphasize in the paper, that tracking and correlation detection are by no means mutually exclusive algorithms for detecting pitch motion direction. To the contrary, the auditory system likely uses both for this task in natural conditions (as is also true of visual motion detection). Our experiments have tried to identify stimuli that highlight correlational computations and minimize or preclude tracking. (Please also see **Figure R2** and related discussion of whether looking only for same-polarity patterns could explain our results. That sort of pattern detection would look a lot like object tracking, and we don't think it can easily explain our results.)

5. The contrast shown in Fig 4H and Fig 4I, which I believe is the average of the response to up-motion and down-motion vs the response to stimuli with simultaneous up- and down-motion should be more clearly indicated on the figure itself.

Thanks for noting this; in response we added some arrow symbols \uparrow and $\downarrow > \uparrow\downarrow$ for the group and individual results to the revised figure. We have also clarified in the caption that we conducted a weighted linear average comparison looking for voxels where the mean activation across up/down responses (weighted equally) exceeded the responses to simultaneous up/down stimuli.

6. The mask that was used to search for significant clusters should be shown somewhere, at least in the supplement. Additionally, it would help convince the readers regarding the significance of the clusters shown to also include a whole-brain analysis (i.e., without using a mask).

Thanks for this important point. We had *a priori* predictions only for regions of auditory cortex (primary, non-primary, and STG regions) and that is why we chose to use a mask. We have now added an image of the mask to the supplemental material (Figure S4, see below).

***A priori* Anatomical Mask**

Moreover, below we show the results of a whole-brain analysis (non-masked) with stricter corrections (FPR, $p < 0.01$); we still see some of the same voxels active in our contrast in the regions we see in the original masked results, at least in 4/5 of our subjects. We thus believe the results we found were generally robust, though of course the masking revealed more activation. We would be happy to include the whole-brain result in the supplement at the reviewer's discretion.

7. The rationale for using different stimuli for the fMRI experiment isn't clear to me. Was it mainly for simplicity of testing the opponency idea in fMRI? In any case, it would be useful for the reader to be able to hear those stimuli for themselves, as the authors have done for the stimuli used in the psychophysical and modeling studies.

Thank you for this suggestion. In the revised manuscript, we have included a supplementary video that contains the up, down, and up+down stimuli used in the fMRI experiments. We note that we initially wanted to use the same stimuli in our fMRI experiments as in Figures 1-3, but found that participants could not distinguish these (relatively weak) percepts in the scanner. They could easily distinguish up and down in the stimuli we used.

Note that the up+down stimulus does not really sound like both up and down together. We attribute this percept precisely to the opponent process itself when we listen to these stimuli. Opponency would tend to suppress detectors sensitive both to rising pitch and to falling pitch when both stimuli are simultaneously present. A similar effect occurs in vision: drifting sinusoids to the left or right are clearly visible as motion to the left or right, but when they are summed, it creates a counterphase grating, which is *not* perceived as both motion to the left and to the right. There, the perceived absence of net motion is attributed to opponency in visual area MT. (Distinct stimuli with dots moving in different direction can look like both

directions at once, in what is termed ‘transparent motion’ (Qian, Andersen, and Adelson 1994).)

8. Related to (7), and looking at Fig S4a, pitch saliency seems like it could be a potential confound in that both the single upward-sweeping complex tone and its downward-sweeping counterpart may generate pitch percepts that are more salient than the combined upwards- and downwards-sweeping tones.

This is a fair point. In some sense, one could argue that an opponency computation, by weakening the direction-specific response in spectrotemporally sensitive regions, also weakens the saliency of the stimulus to the observer, in that one potentially salient feature (direction) is now ambiguous. Thus, opponency and saliency may be two sides of the same coin, so to speak. In terms of loudness or intensity however, the stimuli were matched, in that the up/down simultaneous stimulus was the sum of the up and down stimuli. And given the additional literature the reviewer notes below, which shows opponency-like computations in other aspects of audition, we think our interpretation of opponency is parsimonious.

There are other auditory features – spatial localization, in particular - for which there is good evidence for opponency codes. This work should be cited. For example:

- Stecker et al. 2005 PLoS Biology
- Magezi and Krumbholz 2010 J Neurophys
- Ortiz-Rios et al. 2017 Neuron
- Day and Delgutte 2013 J Neurosci
- Derey et al. 2015 Cerebral Cortex
- Werner-Reiss and Groh 2008 J Neurosci
- Brand et al. 2002 Nature
- Briley et al. 2013 J Assoc Res Oto
- Stecker et al. 2015 Neuroimage
- McLaughlin et al. 2016 J Assoc Res Oto

Thank you for pointing us to this literature on opponency in spatial localization. In the revised manuscript, we now refer to several of these papers in the discussion where we discuss opponency.

As a very last note, on revision we discovered that we had reported 9 subjects in the Figure S2 experiments, when in fact there were 8 subjects. We have changed the caption to note the correct number.

References

- Adelson, EH, and JR Bergen. 1985. 'Spatiotemporal energy models for the perception of motion', *Journal of the Optical Society of America A*, 2: 284-99.
- Allen, Emily J, Juraj Mesik, Kendrick N Kay, and Andrew J Oxenham. 2022. 'Distinct representations of tonotopy and pitch in human auditory cortex', *Journal of Neuroscience*, 42: 416-34.
- Andoni, Sari, and George D Pollak. 2011. 'Selectivity for spectral motion as a neural computation for encoding natural communication signals in bat inferior colliculus', *Journal of Neuroscience*, 31: 16529-40.
- Badwan, Bara A., Matthew S Creamer, Jacob A. Zavatone-Veth, and Damon A Clark. 2019. 'Dynamic nonlinearities enable direction opponency in *Drosophila* elementary motion detectors', *Nature Neuroscience*, 22: 1318-26.
- Carandini, Matteo, and David J Heeger. 2012. 'Normalization as a canonical neural computation', *Nature Reviews Neuroscience*, 13: 51-62.
- Chi, Taishih, Powen Ru, and Shihab A Shamma. 2005. 'Multiresolution spectrotemporal analysis of complex sounds', *The Journal of the acoustical society of America*, 118: 887-906.
- Clark, Damon A, and James E Fitzgerald. 2024. 'Optimization in visual motion estimation', *Annual Review of Vision Science*, 10: 2.1-2.24.
- Demany, Laurent, Daniel Pressnitzer, and Catherine Semal. 2009. 'Tuning properties of the auditory frequency-shift detectors', *The Journal of the acoustical society of America*, 126: 1342-48.
- Demany, Laurent, and Christophe Ramos. 2005. 'On the binding of successive sounds: Perceiving shifts in nonperceived pitches', *The Journal of the acoustical society of America*, 117: 833-41.
- Demany, Laurent, and Catherine Semal. 2018. 'Automatic frequency-shift detection in the auditory system: A review of psychophysical findings', *Neuroscience*, 389: 30-40.
- Fitzgerald, J.E., A.Y. Katsov, T.R. Clandinin, and M.J. Schnitzer. 2011. 'Symmetries in stimulus statistics shape the form of visual motion estimators', *PNAS USA*, 108: 12909-14.
- Heeger, David J, Geoffrey M Boynton, Jonathan B Demb, Eyal Seidemann, and William T Newsome. 1999. 'Motion opponency in visual cortex', *Journal of Neuroscience*, 19: 7162-74.
- Heitmann, Carolin, Minye Zhan, Madita Linke, Cordula Hölig, Ramesh Kekunnaya, Rick van Hoof, Rainer Goebel, and Brigitte Röder. 2023. 'Early visual experience refines the retinotopic organization within and across visual cortical regions', *Current Biology*, 33: 4950-59. e4.

- Lu, Zhong-Lin, and George Sperling. 1995. 'The functional architecture of human visual motion perception', *Vision Research*, 35: 2697-722.
- Marr, David, and Tomaso Poggio. 1976. "From understanding computation to understanding neural circuitry." In *A.I. Memo*. Massachusetts Institute of Technology.
- Norman-Haignere, Sam, Nancy Kanwisher, and Josh H McDermott. 2013. 'Cortical pitch regions in humans respond primarily to resolved harmonics and are located in specific tonotopic regions of anterior auditory cortex', *Journal of Neuroscience*, 33: 19451-69.
- Norman-Haignere, Sam, and Josh H McDermott. 2016. 'Distortion products in auditory fMRI research: measurements and solutions', *NeuroImage*, 129: 401-13.
- Norman-Haignere, Sam V, Nancy Kanwisher, Josh H McDermott, and Bevil R Conway. 2019. 'Divergence in the functional organization of human and macaque auditory cortex revealed by fMRI responses to harmonic tones', *Nature Neuroscience*, 22: 1057-60.
- Pinsk, Mark A, Michael Arcaro, Kevin S Weiner, Jan F Kalkus, Souheil J Inati, Charles G Gross, and Sabine Kastner. 2009. 'Neural representations of faces and body parts in macaque and human cortex: a comparative FMRI study', *Journal of Neurophysiology*, 101: 2581-600.
- Potters, Marc, and William Bialek. 1994. 'Statistical mechanics and visual signal processing', *Journal de Physique I*, 4: 1755-75.
- Qian, Ning, Richard A Andersen, and Edward H Adelson. 1994. 'Transparent motion perception as detection of unbalanced motion signals. I. Psychophysics', *Journal of Neuroscience*, 14: 7357-66.
- Salazar-Gatzimas, Emilio, Margarida Agrochao, James E Fitzgerald, and Damon A Clark. 2018. 'The Neuronal Basis of an Illusory Motion Percept Is Explained by Decorrelation of Parallel Motion Pathways', *Current Biology*, 28: 3748-62.
- Salazar-Gatzimas, Emilio, Juyue Chen, Matthew S Creamer, Omer Mano, Holly B Mandel, Catherine A Matulis, Joseph Pottackal, and Damon A Clark. 2016. 'Direct measurement of correlation responses in *Drosophila* elementary motion detectors reveals fast timescale tuning', *Neuron*, 92: 227-39.
- Tang, Claire, LS Hamilton, and EF Chang. 2017. 'Intonational speech prosody encoding in the human auditory cortex', *Science*, 357: 797-801.
- Victor, Jonathan D, Daniel J Thengone, and Mary M Conte. 2013. 'Perception of second-and third-order orientation signals and their interactions', *Journal of Vision*, 13: 21-21.
- Yost, William A. 1996. 'Pitch of iterated rippled noise', *The Journal of the acoustical society of America*, 100: 511-18.

Dear editor and reviewers,

We are thankful for the additional thoughts and suggestions on our revision.

Below we address the two main remaining points:

R2 suggested softening some of our claims regarding the alternative heuristic-based algorithm. We have made the appropriate changes to the MS.

R3 had remaining concerns about the robustness of our fMRI results. We have now performed additional analyses using non-parametric statistics that correct for multiple comparisons, which replicate our original results at the group level and at the individual level (though the conservative individual correction lead to somewhat weaker activation profiles). These additional statistical analyses are now covered in the Results section and have been added to Figure S4. (Furthermore, as an additional sanity check we ran the main group analysis using a completely independent processing pipeline — and we found similar results to the analysis reported in the paper.) We have also clarified that we did indeed use an *a priori* anatomical mask of a wide area of auditory-sensitive cortical regions. Finally, we have made some additional changes to the text to make our fMRI analysis methods clearer, as we realized that our prior revision had some confusing language. We think the fMRI section of our paper is now much stronger and thank the reviewer for spurring these additional analyses and robustness checks.

We thank the reviewers again for their time and attention to detail.

Best regards,
Parisa Vaziri, Damon Clark, & Sam McDougle

Reviewer #1 (Remarks to the Author):

Thank you for thoroughly addressing my comments and engaging more thoroughly with the existing pitch perception literature.

We are happy to hear this positive evaluation of our revision.

Reviewer #2 (Remarks to the Author):

The authors have extensively engaged with all of my previous comments, and I would first like to acknowledge the amount of work put in the revision in general and in the rebuttal in particular. The detailed responses to each point were very interesting to read, and led to further insights for me. For instance, I did enjoy how you highlighted generic algorithmic principles. So, thank you for this asynchronous but fruitful scientific exchange!

We are glad to hear this positive assessment.

Overall, I am now reassured that the results cannot be easily accounted for by existing auditory models, which was my main concern. Moreover, I understand better how they could in fact contribute to such models - a point which is well made in the Discussion. There would be still a few things I'd be curious to clarify, but this is quite subjective and I don't feel it would contribute constructively to the review process, so I will keep this new review very brief.

The first thing that may deserve minor adjustments is what is concluded from the new figure panels S2c,d and S3a, b (I found these new simulations very interesting, thanks). If I understand correctly, the heuristic model always predicts correctly the subjective direction of pitch shift. However, the size of the "signal" from the heuristic model is much smaller for negative than positive correlations. I agree that this observation strengthens your alternative model based on spectrotemporal correlations, from a parsimony argument, but I don't think it rules out completely the heuristic model: one could imagine all kinds of transforms/thresholding to map the heuristic model's output to perception. So, maybe soften the claim that this rules out completely the heuristic model?

We agree that it is actually surprisingly hard to design experiments that can completely rule out a tracking-style algorithm. We have softened the claim in the paper that this rules out a heuristic model to say instead that these results favor a correlational model as the most parsimonious explanation, and that this model also has known implementation in neural circuits.

The second thing concerns the acoustic analyses. In the rebuttal, you mention that "However, the correlations we observe will exist even with a single pure tone moving to higher or lower frequency." I completely agree. But doesn't this go in the direction of my argument, which was that the main conclusion from the acoustic analysis may have been drawn from all sorts of stimuli, artificial or natural? Speech sounds, at least the voiced bits, can be thought of as a coherent bunch of pure tones (harmonic as it turns out, but you are right, it is only the coherence part that matters). Isn't this fact alone enough to predict the outcome, for English, Mandarin, or

any language that has a fair proportion of voiced sounds? All this to say that I agree that it was useful to check that the prediction holds for natural sounds, but I still believe that the outcome may have been predicted from first principles. If the authors disagree, it could be useful to spell out what alternative outcomes could have been predicted. Otherwise, maybe mention the generic interpretation somewhere?

Thank you for this comment. Indeed, we were never in doubt that our analysis would work on speech waveforms (once we had the independent metric of ‘rising’ vs. ‘falling’). This is because correlational visual algorithms work on virtually any natural motion in scenes, though with varying degrees of accuracy, dependent on scene statistics. It was less clear that all four ‘sub-correlations’ (the (+,+), (+,-), etc.) would work equally well, but as we show, this will likely depend on processing while calculating them (Figure S5). As suggested, in the revised manuscript we have now added the generic interpretation to the results section where we describe what we found.

Again, congratulations for an original, impressive, and thought-provoking piece of work.

This review process has been very productive and we think improved the paper substantially. Thank you. We appreciate your positive evaluation and comments.

Reviewer #3 (Remarks to the Author):

The authors did a good job responding to all my earlier comments. However, I still have substantial concerns.

1. I wouldn't interpret the fMRI results to be as robustly in favor of pitch direction opponency as the authors do. First, it's only 5 subjects.

This is a fair point. In the revised manuscript, we have softened the language throughout to address the reviewer's comment. We certainly do not want to oversell the robustness of our fMRI result. While we do think the results of our contrast and beta estimate analyses are sufficiently convincing to be included in this study, especially with the additional analyses we discuss below, there is certainly more work that could be done on this question in follow-up studies with either larger Ns, more variety in the auditory stimuli, or use of timed pauses in the scan sequence during stimuli presentation to increase stimulus-specific auditory signals.

Second, not all subjects show significant voxels in the *fdr*-corrected maps.

All 5 subjects showed significant effects in the contrast of interest in our original parametric analyses. But, at the more conservative *p*-threshold we used in our responses in the first revision ($p < 0.01$, two-sided), 4/5 subjects showed significant auditory cortical effects, with one subject not showing any significant bilateral effects and two subjects showing mostly unilateral effects.

However, in responding to the reviewer's points here we discovered that our language in the previous revision was not as clear as it should have been. We regret the confusion that this caused and apologize for that. In our original and first revision analyses, we performed what *nilearn* dubs an "FPR correction" (false positive rate) not "FDR correction" (false discovery rate) with cluster thresholding at $p < 0.05$ (two-sided) and no allowable clusters < 20 voxels. Thus, our analysis is better described as cluster formation thresholding but *without* multiple comparisons corrections (whereas FDR would implement voxel-wise correction for multiple comparisons). (While we did write "FPR" in the text, the nomenclature of this uncorrected function in *nilearn* is a bit obscure and we should have avoided it.) Thus, in the prior analyses, we were not correcting for multiple comparisons across voxels. Below and in the revised manuscript, we show our results while correcting for multiple comparisons at both the group analysis and individual analysis levels.

For the sake of clarity and more rigor, in the rest of our response we focus on two key points: Emphasizing that the auditory cortical mask we used was indeed a fully *a priori* anatomical mask, which motivated our statistical approach, and performing additional analyses with various corrections for multiple comparisons to test the robustness of our result.

1) First, we want to emphasize that the masked analysis was based on our focus on the effects of the contrast of interest (Directional > Opponent stimuli) in areas of human cortex known to be involved in auditory processing. Our mask was an *a priori* defined concatenation of all voxels in Heschel's gyrus and *any* region of the STG. We defined these voxels as those that pass the 50% probability threshold of being in those areas according to the Harvard-Oxford probabilistic MRI atlas. Computed this way, the mask is rather diffuse and liberal. In the revised methods section, we have endeavored to make this selection procedure clearer. To reiterate, the mask we used was a large, completely *a priori* anatomical ROI created from an atlas, not something determined from any observed functional results. We focused on auditory cortex because the opponency hypothesis makes sense there but not elsewhere in the brain — that is, our opponency hypothesis was specifically related to auditory cortex. Activity elsewhere in the brain (e.g., PFC) was not something we were prepared to interpret in this study, where our neuroimaging result was just one component of a broader suite of experiments. Thus, we think the use of a liberal, *a priori*, anatomically-defined mask was a reasonable and standard analysis choice and was also the motivation of our general statistical approach.

That said, as referenced in our response to a later point below, we do see some other activity in the brain in addition to auditory regions when we look beyond the auditory mask. Given that the directional and opponent stimulus types did have some phenomenological differences (they are easily discriminated by the listener), we can speculate that these activations may reflect some interesting aspects of these stimuli that may be unrelated to opponency (e.g., Visual imagery? Different melodic schema?), but we don't think we can say more given the targeted experiments we ran.

2) As noted above, we want to correct one point here — in the analyses we presented in the original and initial revised submissions, we did *not* apply voxel-wise FDR correction at the single-subject (or group) level. In our initial analyses, our approach to correcting for spurious false-positive clusters was to set a minimum cluster-forming threshold (of 20 contiguous voxels) but we did not do any posthoc voxel-wise corrections. We again apologize that our methods were not written clearly enough. In the revised manuscript, we have edited the Methods to make this point clear. In the revised manuscript, we make use of maximum cluster-size null distributions to test the significance of our results. This is not voxel-wise FDR correction, but is a strong form of false discovery correction, since the entire response is contracted into a single variable (max cluster size), so multiple comparisons are avoided. Since voxel-wise FDR correction makes assumptions about the independence of voxels that are highly correlated in our data, we think voxel-wise correction here is far too conservative, while the permutation-based analyses we use account for the spatiotemporal correlations in voxels during these stimuli.

New Analyses:

To increase reader (and reviewer) confidence in our result, we have now performed a non-parametric permutation-based test for the (masked) group result (implemented in *nilearn* using the `non_parametric_inference` function, which is similar to FSL's `randomise` function). This more conservative, non-parametric cluster-corrected analysis showed that the largest auditory cortical cluster in the non-permuted group test was

significant relative to the null distribution of cluster sizes, with $P=0.03$, where this value represents a form of false discovery rate correction (we note that this is the minimum p-value possible with this test on this data, i.e., $1/2^5$). We report this new result in the figure caption and provide a paragraph in the Methods section detailing this. Below, we show the result of that cluster-based non-parametric FDR correction, and we now also show it in Figure S4.

Figure R3.1: Results of our Directional > Opponent contrast, using a non-parametric permutation test to perform statistical correction (projected onto an average cortical surface).

To further bolster our own confidence in the results, we ran two additional analyses. First, it is well known that different fMRI analysis packages can sometimes yield different results due to differences in preprocessing pipelines and other statistical assumptions (Bowring, Maumet et al. 2019). We thus tried using a completely different pipeline for the analysis (FSL), from preprocessing through to second-level inference. We used the FSL-specific FLAME 1 function, known to be effective at reducing false positives in group level inference (Eklund, Nichols et al. 2016), masked, with both cluster-forming threshold and cluster-corrected alpha set at $p<0.05$. The analysis here further differs from the permutation analysis above because it is based on a parametric random-field model fitted to the data (FSL's default). In this analysis, we see clear activation in the auditory areas that also show up in the main analyses in Figures 4 and S4. This gives us even more confidence that we have a true positive result for this contrast in areas of auditory cortex.

Figure R3.2: Results of our Directional > Opponent contrast, using a standard FSL pipeline (projected onto an average cortical surface). These results largely mirror the ones in the *nilearn* pipeline in the paper.

Finally, as a third (and arguably most conservative) test we wrote a custom routine using *nilearn* to perform non-parametric inference at the individual level by randomly permuting trial-wise stimulus condition labels (i.e., up, down, and opponent tones) for each individual participant before computing the Directional > Opponent contrast. Our analysis followed an established procedure outlined for such analyses (Nichols and Holmes 2002). For each permutation, we computed clusters with $z > 3$ for the contrast. By doing 100 permutations, we created a null distribution of cluster sizes, under the hypothesis that trials are interchangeable; that is, under the null hypothesis that the trials result only in signal fluctuations uncorrelated with the stimulus label. This permutation methodology has the advantage of preserving the spatiotemporal structure of the measured responses, but decorrelating them to produce a null hypothesis contrast distribution. We then looked at subject-level true maximum cluster sizes (with correct labels) relative to this null distribution for each participant. As seen in the histograms below, 4/5 subjects showed $p < 0.1$ for the true maximum cluster size relative to their respective null distributions, 2/5 showed $p < 0.05$, and 1/5 showed $p > 0.1$. (These are all 1-tailed tests because our prior, laid out in Figure 4 and our hypothesis motivated by psychophysical measurements, is that we should test for this contrast to be positive.)

Figure R3.3: *Top:* Results of our Directional > Opponent contrast, using a non-parametric permutation test at the individual subject level to perform statistical correction (projected onto an average cortical surface). Black circles denote general locales of significant activity. Clusters shown are ones greater than the 90th percentile of the max cluster size distribution. That is, we (conservatively) used the max cluster distribution for including all clusters shown here. *Bottom:* Null distribution of the maximum cluster size (in mm³) over 100 stimulus label permutations, with the max cluster size of the true labeling indicated by the red bar. One subject (S3) did not show reliable activity for this contrast when non-parametric inference was performed.

Taken as a whole and keeping in mind the *a priori* nature of our original anatomical masking procedure, these analyses suggest that our results are sufficiently robust for the straightforward univariate-based question we are asking here. We find significance in the cluster size analysis at the group level and at varying levels of confidence among the individuals, all in roughly the same regions of the cortical auditory system.

Third, the mask used is atypical insofar as it isn't a concatenation of predefined anatomical regions based on an atlas.

We apologize that our methods were not clear on this point. As described above, our mask was an *a priori* defined concatenation of all voxels in Heschel's gyrus or any region of the STG. In the revised manuscript, we have clarified the relevant methods section.

I could be more convinced if the authors would also show an unmasked version of Fig 4H and 4I, that is, the same cluster-based correction method without the mask that is now included in Fig S4.

Below are plots of the group analysis using the same cluster forming constraint (must be >20 voxels) and statistical threshold ($\alpha = 0.05$ two-sided, no MC correction) as in our initial MS, but here without any masking. In addition to the key auditory regions we also see some activity in regions of visual cortex and in the (left) central sulcus. Because our experiments were strongly hypothesis driven and not exploratory, we have little to say about any potential activation outside of auditory regions. If the concern is more about multiple comparisons, then we think the analyses above with the masked auditory region are the appropriate tests, given our experimental design and reasoning.

Smaller points:

- Yes, please include the whole-brain FDR-corrected singles-subject contrast maps in the supplement.

We have added the relevant figures to Figure S4. Please note the stats approach we used was not voxel-wise FDR corrected in the individual subjects, but corrected using non-parametric inference via stimulus label permutations to test whether participant responses

are larger than expected due to a null hypothesis, as described above and in the revised Methods and Figure S4.

- In Figs 1-3, please make more explicit the frame duration being used (e.g. 166.67 versus 40 ms).

Thank you for pointing out that this wasn't clear. We have added text to the captions of both figures emphasizing the stimulus parameters. Note that while Figure 1 had 'frames' in the sense of all sound intensities being updated synchronously every 1/6 second, the stimuli in Figures 2 and 3 were continuous (with a resolution of 1 ms), but the pips lasted for 50 ms, had delays of 40 ms, etc.

- The rationale for using different stimuli (and not the correlated stimuli as used in the psychophysics) should be included in the Results section (the first time the new stimuli/expt are mentioned) in addition to the Methods section.

Thank you. We have added this rationale to the Results where we introduce the stimulus and experiment.

- The idea for opponency coding in the auditory domain should also be briefly mentioned in the motivation for the fMRI experiment.

Thank you for this suggestion. We have added brief prose on auditory opponent coding in azimuthal location coding when we motivate the fMRI experiment.

References

Bowring, A., C. Maumet and T. E. Nichols (2019). "Exploring the impact of analysis software on task fMRI results." Human brain mapping **40**(11): 3362-3384.

Eklund, A., T. E. Nichols and H. Knutsson (2016). "Cluster failure: Why fMRI inferences for spatial extent have inflated false-positive rates." Proceedings of the national academy of sciences **113**(28): 7900-7905.

Nichols, T. E. and A. P. Holmes (2002). "Nonparametric permutation tests for functional neuroimaging: a primer with examples." Human brain mapping **15**(1): 1-25.